# From stability of Langevin diffusion to convergence of proximal MCMC for non-log-concave sampling

**Marien Renaud**[*]
Univ. Bordeaux, CNRS, Bordeaux INP, IMB, UMR 5251
F-33400 Talence, France

**Valentin de Bortoli**
ENS, CNRS, PSL University
Paris, 75005, FRANCE

**Arthur Leclaire**
LTCI, Télécom Paris, IP Paris, France

**Nicolas Papadakis**
Univ. Bordeaux, CNRS, INRIA, Bordeaux INP, IMB, UMR 5251
F-33400 Talence, France

## Abstract

We consider the problem of sampling distributions stemming from non-convex potentials with Unadjusted Langevin Algorithm (ULA). We prove the stability of the discrete-time ULA to drift approximations under the assumption that the potential is strongly convex at infinity. In many context, e.g. imaging inverse problems, potentials are non-convex and non-smooth. Proximal Stochastic Gradient Langevin Algorithm (PSGLA) is a popular algorithm to handle such potentials. It combines the forward-backward optimization algorithm with a ULA step. Our main stability result combined with properties of the Moreau envelope allows us to derive the first proof of convergence of the PSGLA for non-convex potentials. We empirically validate our methodology on synthetic data and in the context of imaging inverse problems. In particular, we observe that PSGLA exhibits faster convergence rates than Stochastic Gradient Langevin Algorithm for posterior sampling while preserving its restoration properties. The code associated with the paper can be found in github.

## 1 Introduction

Sampling a probability distribution $\pi \propto e^{-V}$ on $\mathbb{R}^d$ is a fundamental task in machine learning [2, 10, 13, 28, 42, 51, 91]. It becomes challenging when the potential $V$ is non-convex [39, 43]. A well-known approach is to solve the Langevin stochastic differential equation [59, 79], defined by

$$\mathrm{d}x_t = -\nabla V(x_t)\mathrm{d}t + \sqrt{2}\mathrm{d}w_t, \tag{1}$$

where $(w_t)_{t\geq 0}$ is a Wiener process [53] on $\mathbb{R}^d$. This process can be seen as a continuous gradient descent on $V$ randomly perturbed by $\sqrt{2}\mathrm{d}w_t$. Under mild assumptions, equation (1) admits $\pi$ as its invariant distribution [79]. Moreover, the solution converges geometrically in law to $\pi$ when $t$ goes to infinity if $V$ is strongly convex *at infinity* [30, 23]. In Section 4 we discuss alternative methods that have been developed to sample $\pi$, such as Gibbs Sampling [17] or Diffusion Models [87].

As the function $\nabla V = -\nabla \log \pi$ is often unknown [74], it is approximated by a drift function $b : \mathbb{R}^d \to \mathbb{R}^d$ such that $b \approx \nabla V$. In practice, the Euler-Maruyama discretization of equation (1)

---

[*]corresponding author: marien.renaud@math.u-bordeaux.fr

39th Conference on Neural Information Processing Systems (NeurIPS 2025).

| | $f$ non-convex | $g$ non-convex | $\text{Prox}_g$ inexact | $\mathbf{W}_2$ metric | $\mathbf{W}_p$ metric | $TV$ metric |
|---|---|---|---|---|---|---|
| [29, 84, 85] | ✗ | ✗ | ✗ | ✓ | ✗ | ✗ |
| [33] | ✗ | ✗ | ✓ | ✓ | ✗ | ✗ |
| This paper | ✓ | ✓ | ✓ | ✓ | ✓ | ✓ |

Table 1: Existing convergence results in the literature for PSGLA (3).

leads to the inexact Unadjusted Langevin Algorithm (iULA)

$$X_{k+1} = X_k - \gamma b(X_k) + \sqrt{2\gamma} Z_{k+1}, \tag{2}$$

with a step-size $\gamma > 0$ and $Z_{k+1} \sim \mathcal{N}(0, I_d)$ i.i.d. It is called inexact because $b$ is an inexact approximation of $\nabla V$ and unadjusted because the invariant law of this Markov Chain is $\pi$ up to a discretization error (see Theorem 2 in Section 2). This Langevin Markov Chain could be adjusted by adding a Metropolis–Hastings accept-reject step [78, 70]. With ergodic MCMC, the uniform measure supported on the iterates $(X_k)_{0 \leq k \leq N}$ will be used as an estimator for $\pi$. To evaluate the quality of iULA, there is a need to quantify the shift between the invariant law of the MCMC and the target distribution $\pi$. This leads to investigate the stability of iULA invariant law to shift in the drift $b$. Existing stability results [98, 75] are limited by a discretization error. With a new proof strategy, we overcome this limitations in Theorem 1 in Section 2.

In many applications the potential has a composite form $V = f + g$. For instance in Bayesian inverse problems [73, 15, 36], the goal is to sample the posterior distribution with $f$ the negative log-likelihood, which is usually smooth and possibly non-convex, and $g$ the regularization, which might be non-smooth and non-convex. Due to the non-smoothness of $V$, iULA that relies on evaluation of $\nabla V$ can not be implemented. A popular solution is to design a forward-backward Langevin scheme [9, 7, 85, 1], with the so-called Proximal Stochastic Gradient Langevin Algorithm (PSGLA) [29, 84] defined by

$$X_{k+1} = \text{Prox}_{\gamma g} \left( X_k - \gamma \nabla f(X_k) + \sqrt{2\gamma} Z_{k+1} \right), \tag{3}$$

where the proximal operator defined by $\text{Prox}_{\gamma g}(x) = \arg\min_{y \in \mathbb{R}^d} \frac{1}{2\gamma} \|x-y\|^2 + g(y)$ can be seen as an implicit gradient step. PSGLA has recently been studied [84, 33] in the convex setting. However, to our knowledge, there is no convergence analysis in the non-convex setting (see Table 1). We fill this gap by providing a convergence proof based on our stability result for Langevin Markov Chain in the case of $f$ smooth and $g$ non-smooth and weakly convex (Theorem 3 in Section 3).

## 1.1 Contributions

**(1)** We prove the first stability result for inexact Unadjusted Langevin Algorithm (iULA defined in equation (2)) with respect to the drift without discretization errors (Theorem 1 in Section 2). **(2)** We give the discretization error of iULA in various metrics including $p$-Wasserstein and Total Variation (Theorem 2 in Section 2). **(3)** Leveraging properties of the Moreau envelope, we demonstrate the first convergence proof of Proximal Stochastic Gradient Langevin Algorithm (PSGLA defined in equation (3)) (Theorem 3 in Section 3) and Inexact PSGLA (Theorem 4 in Section 3) for non-convex potentials. **(4)** We apply our theory to posterior sampling and show empirically that the mixing time of PSGLA is significantly shorter that the iULA (Section 5). In particular, we propose a Plug-and-Play version of PSGLA for image restoration that achieves state-of-the art restoration.

## 1.2 Notations and definitions

For a distribution $\mu$ on $\mathbb{R}^d$ and a function $\phi : \mathbb{R}^d \to \mathbb{R}$, we denote $\mu(\phi) = \int_{\mathbb{R}^d} \phi d\mu$. For two distributions $\mu, \nu$ on $\mathbb{R}^d$, the Total Variation (TV) writes

$$\|\mu - \nu\|_{TV} := \sup_{\substack{\phi:\mathbb{R}^d \to \mathbb{R} \\ \|\phi\|_\infty \leq 1}} \mu(\phi) - \nu(\phi). \tag{4}$$

For a measurable function $\phi : \mathbb{R}^d \to \mathbb{R}^d$, the push-forward of a probability distribution $\mu$ by $\phi$ denoted by $\phi\#\mu$ is the distribution defined for a measurable set $A \subset \mathbb{R}^d$ by $\phi\#\mu(A) = \mu(\phi^{-1}(A))$.

The $p$-Wasserstein distance between $\mu$ and $\nu$ is defined, for $p \geq 1$, by

$$\mathbf{W}_p(\mu, \nu) = \left( \min_{\beta \in \Pi} \int_{\mathbb{R}^d \times \mathbb{R}^d} \|x - y\|^p d\beta(x, y) \right)^{\frac{1}{p}}, \tag{5}$$

with $\Pi$ the set of probability law $\beta$ on $\mathbb{R}^d \times \mathbb{R}^d$ with marginals $\mu$ and $\nu$.

A function $g$ is $\rho$-weakly convex, with $\rho > 0$, if and only if $g + \frac{\rho}{2}\|\cdot\|^2$ is convex. This notion is also called semi-convexity [18, 49]. A twice-differentiable function $g$ is $m$-strongly convex at infinity if and only if there exists $R \geq 0$ such that $\forall x \in \mathbb{R}^d$, if $\|x\| \geq R$, then $\nabla^2 g(x) \succeq \mu I_d$. Note that we define the strong convexity at infinity only for twice differentiable functions, which is sufficient for this paper. This non-trivial notion of strong convexity on non-convex sets is discussed in Appendix E.

A Markov Chain $(X_k)_{k \geq 0}$ has an *invariant law* if there exists $\mu$ a probability distribution on $\mathbb{R}^d$ such that if $X_0 \sim \mu$, then $\forall k \geq 0$, $X_k \sim \mu$. A Markov Chain $(X_k)_{k \geq 0}$ is said to be *geometrically ergodic*, if the law of $X_k$ converges geometrically to the invariant law, i.e. there exist $A \geq 0$, $\rho \in (0, 1)$, an invariant law $\mu$, such that $\mathbf{W}_1(p_{X_k}, \mu) \leq A\rho^k$ with $p_{X_k}$ the law of $X_k$.

## 2 Sampling stability and discretization error

In this section, we study the sampling stability to drift errors of the inexact Unadjusted Langevin Algorithm (iULA) defined in equation (2). The sampling stability result (Theorem 1) and the discretization error bound (Theorem 2) demonstrated in the section will be the key ingredients for the PSGLA convergence analysis in Section 3. We introduce two iULA with two arbitrary drifts as

$$X_{k+1}^i = X_k^i - \gamma b^i(X_k^i) + \sqrt{2\gamma} Z_{k+1}^i, \tag{6}$$

for $i \in \{1, 2\}$, with the step-size $\gamma > 0$, $Z_{k+1}^i \sim \mathcal{N}(0, I_d)$ i.i.d. and the drifts $b^i \in \mathcal{C}^0(\mathbb{R}^d, \mathbb{R}^d)$.

**Assumption 1.** *There exist $L, R \geq 0$ and $m > 0$ such that the drift $b$ verifies*

   *(i)  $b$ is $L$-Lipschitz, i.e. $\forall x, y \in \mathbb{R}^d$, $\|b(x) - b(y)\| \leq L\|x - y\|$*

   *(ii)  $\forall x, y \in \mathbb{R}^d$ such that $\|x - y\| \geq R$, we have $\langle b(x) - b(y), x - y \rangle \geq m\|x - y\|^2$.*

Assumption 1(i) imposes that the drift is sufficiently regular, without uncontrolled variations. In particular, if $b = \nabla V$, the $L$-Lipschitzness on $b$ involves that the potential $V$ is $L$-weakly convex If $b = \nabla V$, Assumption 1(ii) is a relaxation of the strong convexity assumption on $V$, as it only holds for couple of points sufficiently far away Assumption 1(ii) is called weak dissipativity [24, 62] or distant dissipativity [67]. Note that if the drift is $m$-strongly convex at infinity and $L$-weakly convex then it is weakly dissipative.

We define the $\ell_2(\mu)$ distance between two functions $f_1, f_2 : \mathbb{R}^d \to \mathbb{R}$ by

$$\|f_1 - f_2\|_{\ell_2(\mu)} = \left( \mathbb{E}_{Y \sim \mu} \left( \|f_1(Y) - f_2(Y)\|^2 \right) \right)^{\frac{1}{2}}. \tag{7}$$

**Theorem 1.** *Let $b^1$ and $b^2$ satisfy Assumption 1. $X_k^1$ and $X_k^2$ that satisfy equation (6) are two geometrically ergodic Markov Chains with invariant laws $\pi_\gamma^1, \pi_\gamma^2$. Then for $\gamma_0 = \frac{m}{L^2}$ and $p \in \mathbb{N}^\star$ there exist $C_p, C \geq 0$ such that $\forall \gamma \in (0, \gamma_0]$, we have*

$$\mathbf{W}_p(\pi_\gamma^1, \pi_\gamma^2) \leq C_p \|b^1 - b^2\|_{\ell_2(\pi_\gamma^1)}^{\frac{1}{p}}, \tag{8}$$

$$\|\pi_\gamma^1 - \pi_\gamma^2\|_{TV} \leq C \|b^1 - b^2\|_{\ell_2(\pi_\gamma^1)}. \tag{9}$$

The proof of Theorem 1 can be found in Appendix F.4. Theorem 1 shows that errors in the drifts do not accumulate, moreover if two drifts are close (in $\ell_2$ distance) then the corresponding sampled distributions are close. We also demonstrate a similar result in the $V$-norm, a generalization of the $TV$-distance (see Theorem 5 in Appendix F.3). More importantly, contrary to [75], *our bound does not contain any discretization error*. Theorem 1 does not only refine existing works, it also generalizes to $\mathbf{W}_p$ the bounds of [75], which were $\mathbf{W}_1(\pi_\gamma^1, \pi_\gamma^2) \leq C_1 \|b^1 - b^2\|_{\ell_2(\pi_\gamma^1)}^{\frac{1}{2}} + D_1 \gamma^{\frac{1}{8}}$ for equation (8) in the case $p = 1$ and $\|\pi_\gamma^1 - \pi_\gamma^2\|_{TV} \leq C \|b^1 - b^2\|_{\ell_2(\pi_\gamma^1)} + D\gamma^{\frac{1}{4}}$ for equation (9).

*Sketch of the Proof of Theorem 1* First, we prove that $X_k^1, X_k^2$ are geometrically ergodic. Then by using the triangle inequality and the geometric ergodicity we control the shift between the asymptotic laws by a shift between finite time laws. Finally this finite time shift is controlled by the shift between drifts by applying the Girsanov theory with the core Lemma 20 in Appendix F.2.

**Theorem 2.** *Let $b = \nabla V$ verifies Assumption 1 and $\pi \propto e^{-V}$. Then, for $\gamma_0 = \frac{m}{L^2}$ and $\gamma \in (0, \gamma_0]$, the Markov Chain defined in (2) with the drift $b = \nabla V$ has an invariant law $\pi_\gamma$ and is geometrically ergodic. Moreover for $p \in \mathbb{N}^\star$, there exist $D_p, D \geq 0$ such that $\forall \gamma \in (0, \gamma_0]$, we have*

$$\mathbf{W}_p(\pi_\gamma, \pi) \leq D_p \gamma^{\frac{1}{2p}}, \quad \|\pi_\gamma - \pi\|_{TV} \leq D\gamma^{\frac{1}{2}}.$$

Theorem 2 is proved in Appendix F.6 by following the same strategy as Theorem 1 with continuous-time Markov process instead of discrete-time Markov Chain. It provides a result on the discretization error of the Unadjusted Langevin Algorithm, which has been extensively studied in the literature for finite time [67], in the strongly convex setting [20, 8], in $TV$-distance [27] or $KL$-divergence [12]. To our knowledge, formulating this discretization error in $\mathbf{W}_p$ for all $p \in \mathbb{N}$ and in $TV$ under weak dissipativity (Assumption 1(ii)) is a novelty with respect to the literature.

We quantified the sampling shifts due to drift approximations in Theorem 1 and to the discretization error in Theorem 2 for iULA. We can now put together these results.

**Corollary 1.** *Let $b$ and $\nabla V$ verify Assumption 1 and $\pi \propto e^{-V}$. Then for $\gamma_0 = \frac{m}{L^2}$, and $\gamma \in (0, \gamma_0]$, the Markov Chain defined in equation (2) with drift $b$ is geometrically ergodic with an invariant law $\hat{\pi}_\gamma$. Moreover for $p \in \mathbb{N}^\star$, there exist $C_p, D_p, C, D \geq 0$ such that $\forall \gamma \in (0, \gamma_0]$, we have*

$$\mathbf{W}_p(\hat{\pi}_\gamma, \pi) \leq C_p \|b - \nabla V\|_{\ell_2(\hat{\pi}_\gamma)}^{\frac{1}{p}} + D_p \gamma^{\frac{1}{2p}}$$

$$\|\hat{\pi}_\gamma - \pi\|_{TV} \leq C\|b - \nabla V\|_{\ell_2(\hat{\pi}_\gamma)} + D\gamma^{\frac{1}{2}}.$$

Corollary 1 is a direct consequence of Theorem 1 combined with Theorem 2. It quantifies the precision of sampling $\pi \propto e^{-V}$ with the practical algorithm (2) instead of the continuous process defined in equation (1). Corollary 1 extends previous known results established in $\mathbf{W}_2$ in the strongly convex setting [8] to strongly convex *at infinity* setting with various metrics. The pseudo-metric $\|\cdot\|_{\ell_2(\hat{\pi}_\gamma)}$ appears to be the appropriate metric to quantify the difference between drifts.

## 3    Convergence of Proximal Stochastic Gradient Langevin Algorithm for non-convex potentials

We now apply the results of Section 2 to derive in Theorem 3 the first convergence result for PS-GLA (3), summarized in Algorithm 1, in the non-convex setting specified by Assumption 2. Then, in Theorem 4 we deduce the convergence of Inexact PSGLA (3) when approximating the proximal operator of the non-convex potential.

**Assumption 2.** *The potential $V$ is composite, i.e. $V = f + g$ where*

*(i) $f$ is $L_f$-smooth, i.e. $\nabla f$ is $L_f$-Lipschitz.*

*(ii) $g$ is $\rho$-weakly convex with $\rho > 0$, i.e. $g + \frac{\rho}{2}\|\cdot\|^2$ is convex.*

In the case of posterior sampling, which will be detailed in Section 5, $f$ is typically the negative log-likelihood and $g$ the regularization. Under Assumption 2(i), $f$ can be non-convex, as it is the case in image despeckling [25] or black-hole interferometric imaging [89]. The weak convexity of $g$ ensures that the proximal operator $\mathsf{Prox}_{\gamma g}$ is well defined as soon as $\gamma < \frac{1}{\rho}$. Assumption 2(ii) is verified for various convex regularizations as total-variation [83] or for some non-convex deep regularizers [47, 86, 40].

**On the Moreau envelope**    To analyse PSGLA under Assumption 2, we will need properties on the proximal operator and the Moreau envelope $g^\gamma$ of $g$ defined for $\gamma > 0$ and $x \in \mathbb{R}^d$ by $g^\gamma(x) = \inf_{y \in \mathbb{R}^d} \frac{1}{2\gamma}\|x - y\|^2 + g(y)$. Notice that if $g$ is $\rho$-weakly convex and $\rho\gamma < 1$, then the Moreau envelope is well defined. In the following Lemma 1 we provide important properties relative to the proximal operator and the Moreau envelope of weakly convex functions.

---

**Algorithm 1** Proximal Stochastic Gradient Langevin Algorithm (PSGLA)

---

1: **input:** $N > 0$, $y \in \mathbb{R}^m$, $X_0 \in \mathbb{R}^d$
2: **for** $k = 0, \ldots, N - 1$ **do**
3:     $Z_{k+1} \sim \mathcal{N}(0, I_d)$
4:     $X_{k+1} \leftarrow \mathsf{Prox}_{\gamma g}\left(X_k - \gamma \nabla f(X_k) + \sqrt{2\gamma} Z_{k+1}\right)$
5: **end for**
6: **output:** $(X_k)_{k \in [0, N]}$

---

**Lemma 1.** *For a $\rho$-weakly convex function $g$ and $\gamma\rho < 1$, we have the following properties:*

*(i) If $g$ is differentiable at $x$, then we have $\mathsf{Prox}_{\gamma g}\left(x + \gamma \nabla g(x)\right) = x$.*

*(ii) The proximal operator is a gradient-descent step on the Moreau envelope, i.e. $\mathsf{Prox}_{\gamma g}(x) = x - \gamma \nabla g^\gamma(x)$.*

*(iii) The proximal operator $\mathsf{Prox}_{\gamma g}$ is $\frac{1}{1-\gamma\rho}$-Lipschitz.*

*(iv) The image of the proximal operator is almost convex, $Leb(Conv(\mathsf{Prox}_{\gamma g}\left(\mathbb{R}^d\right)) \setminus \mathsf{Prox}_{\gamma g}\left(\mathbb{R}^d\right)) = 0$, with $Leb$ the Lebesgue measure and $Conv(\mathsf{Prox}_{\gamma g}\left(\mathbb{R}^d\right))$ is the convex envelop of $\mathsf{Prox}_{\gamma g}\left(\mathbb{R}^d\right)$.*

In this lemma, we adapt to the weakly convex case results known in the convex setting [65, 80, 46, 3, 41, 50]. In particular point (iv) requires a subtle adaptation with respect to the convex case. All proofs are postponed in Appendix D, where we provide a self-contained analysis of the Moreau envelope for weakly convex functions.

**ULA and PSGLA: two discretizations of the same continuous process** in the case of $g$ differentiable. First, the Euler-Maruyama discretization of equation (1) with a stepsize $\gamma > 0$ leads to the Unadjusted Langevin Algorithm (ULA) [27, 59], defined by

$$X_{k+1} = X_k - \gamma \nabla f(X_k) - \gamma \nabla g(X_k) + \sqrt{2\gamma} Z_{k+1}, \tag{10}$$

with $Z_{k+1} \sim \mathcal{N}(0, I_d)$ i.i.d. This Markov Chain has been shown to sample the target law $\pi \propto e^{-V}$ [59] for a non-convex potential $V$, up to a discretization error (see Theorem 2 in Section 2). ULA adds noise at each iteration to the iterate $X_k$ leading to noisy samples. Taking advantage of the composite nature of $V$, PSGLA (3) appears to be a good candidate for producing high quality samples. Indeed, for PSGLA the iterate $X_k$ is obtained after applying $\mathsf{Prox}_{\gamma g}$, which, in practice, often corresponds to a denoising operator [92]. However PSGLA has not yet been shown to sample the law $\pi$ for a non-convex potential.

Secondly, a semi-implicit discretization of equation (1) with $\gamma \in (0, \frac{1}{\rho})$ and Lemma 1(i) gives

$$X_{k+1} = X_k - \gamma \nabla f(X_k) - \gamma \nabla g(X_{k+1}) + \sqrt{2\gamma} Z_{k+1}$$
$$(I_d + \gamma \nabla g)(X_{k+1}) = X_k - \gamma \nabla f(X_k) + \sqrt{2\gamma} Z_{k+1}$$
$$X_{k+1} = \mathsf{Prox}_{\gamma g}\left(X_k - \gamma \nabla f(X_k) + \sqrt{2\gamma} Z_{k+1}\right).$$

Our postulate is as follows. As the ULA and PSGLA are two discretizations of the same continuous process, we expect that PSGLA also approximately samples the target distribution $\pi \propto e^{-V}$.

**Reformulation of PSGLA as a two-point algorithm** PSGLA (3) can be reformulated as

$$Y_{k+1} = X_k - \gamma \nabla f(X_k) + \sqrt{2\gamma} Z_{k+1}$$
$$X_{k+1} = \mathsf{Prox}_{\gamma g}(Y_{k+1}).$$

By Lemma 1(ii), we get that the iterates $Y_k$ verify the equation

$$Y_{k+1} = \mathsf{Prox}_{\gamma g}(Y_k) - \gamma \nabla f(\mathsf{Prox}_{\gamma g}(Y_k)) + \sqrt{2\gamma} Z_{k+1}$$
$$= Y_k - \gamma b^\gamma(Y_k) + \sqrt{2\gamma} Z_{k+1},$$

where the drift $b^\gamma$ is defined for $y \in \mathbb{R}^d$ as

$$b^\gamma(y) = \nabla f(y - \gamma \nabla g^\gamma(y)) + \nabla g^\gamma(y). \tag{11}$$

Therefore, the shadow sequence $Y_k$ verifies a Markov Chain of the form of equation (2) with a drift that depends on the step-size $\gamma > 0$. In order to apply the result of Section 2, we need to make additional technical assumptions on $g$.

**Assumption 3.**

*(i)* $\forall \gamma \in (0, \frac{1}{\rho})$, $g$ is $L_g$-smooth on $\mathsf{Prox}_{\gamma g}(\mathbb{R}^d)$.

*(ii)* $g^\gamma$ is $\mu$-strongly convex at infinity with $\mu \geq 8L_f + 4L_g$, *i.e. there exists $\gamma_1 > 0$ and $R_0 \geq 0$ such that $\forall \gamma \in (0, \gamma_1]$, $\nabla^2 g^\gamma \succeq \mu I_d$, on $\mathbb{R}^d \setminus B(0, R_0)$.*

Assumption 3(i) is verified by non-smooth functions $g$ such as the characteristic function of a convex set. This assumption is not usual for $g$ that is generally not expected to be smooth on a subset of $\mathbb{R}^d$ [84, 29, 33]. Note that $\mathsf{Prox}_{\gamma g}(\mathbb{R}^d)$ can be a non-convex set if $g$ is not finite everywhere. However, Lemma 1(iv) shows that $\mathsf{Prox}_{\gamma g}(\mathbb{R}^d)$ only differs from being convex by a negligible set.

Assumption 3(ii) is key to ensure that $Y_k$ is geometrically ergodic. Assumption 3(ii) could be removed by slightly modifying PSGLA as detailed in Appendix B. Lemma 16 in Appendix E shows that if $g$ is strongly convex at infinity and *globally* smooth, such as Gaussian mixture models, then Assumption 3(ii) is verified.

**Theorem 3.** *Under Assumptions 2-3, there exist $r \in (0, 1)$, $C_1, C_2 \in \mathbb{R}_+$ such that $\forall \gamma \in (0, \bar\gamma]$, with $\bar\gamma = \min\left(\frac{1}{2L_g}, \frac{\mu}{32(L_f+L_g)^2}, \frac{1}{2\rho}, \gamma_1\right)$, where $L_g, L_f, \rho, \gamma_1$ are defined in Assumptions 2-3, and $\forall k \in \mathbb{N}$, we have*

$$\mathbf{W}_p(p_{Y_k}, \mu_\gamma) \leq C_1 r^{k\gamma} + C_2 \gamma^{\frac{1}{2p}}, \tag{12}$$

*with $p_{Y_k}$ the distribution of $Y_k$ and $\mu_\gamma \propto e^{-f-g^\gamma}$.*

*Moreover there exist $C_3, C_4 \in \mathbb{R}_+$ such that $\forall \gamma \in (0, \bar\gamma]$ and $\forall k \in \mathbb{N}$, we have*

$$\mathbf{W}_p(p_{X_k}, \nu_\gamma) \leq C_3 r^{k\gamma} + C_4 \gamma^{\frac{1}{2p}}, \tag{13}$$

*with $p_{X_k}$ the distribution of $X_k$ and $\nu_\gamma \propto \mathsf{Prox}_{\gamma g} \# e^{-f-g^\gamma}$.*

Theorem 3 is proved in Appendix G.2. It shows the convergence of PSGLA (3) in a non-convex setting. This convergence is exponential with an asymptotic bias due to discretization errors. Theorem 3 can also be proved in $TV$-distance following the same reasoning. Note that the distribution $\mu_\gamma$ and $\nu_\gamma$ are not exactly the target distribution $\pi$, the following Proposition 1 fills this gap by showing that $\mu_\gamma$ and $\nu_\gamma$ both converge to $\pi$ when $\gamma$ goes to zero.

*Sketch of the proof of Theorem 3* First, we prove that under Assumptions 2-3, the drift $b^\gamma$ verifies Assumption 1 with constants independent of $\gamma$. So we can apply the results of Section 2. In particular, $(Y_k)_{k \geq 0}$ is geometrically ergodic, with invariant law $p_\infty$. Then we apply Theorem 1 to quantify the distance between $p_\infty$ and the invariant law of the MCMC with the drift $\nabla f + \nabla g^\gamma$, named $p_\gamma$. Finally, we apply Theorem 2 to quantify the distance between $p_\gamma$ and $\mu_\gamma \propto e^{-f-g^\gamma}$. The triangle inequality and Lemma 1(iii) conclude the proof.

**Proposition 1.** *With $\pi \propto e^{-f-g}$, $\mu_\gamma \propto e^{-f-g^\gamma}$ and $\nu_\gamma = \mathsf{Prox}_{\gamma g} \# \mu_\gamma$, for $p \geq 1$, we have*

$$\lim_{\gamma \to 0} \mathbf{W}_p(\mu_\gamma, \pi) = 0 \;\;,\;\; \lim_{\gamma \to 0} \mathbf{W}_p(\nu_\gamma, \pi) = 0\,.$$

*Moreover, if $g$ is $L$-Lipschitz, there exists $E_p \in \mathbb{R}_+$ such that $\forall \gamma \in [0, \frac{2}{L^2}]$*

$$\mathbf{W}_p(\mu_\gamma, \pi) \leq E_p(L^2\gamma)^{\frac{1}{p}} \;\;,\;\; \mathbf{W}_p(\nu_\gamma, \pi) \leq E_p(L^2\gamma)^{\frac{1}{p}} + L\gamma\,.$$

Proposition 1 is proved in Appendix G.3. We choose to separate Proposition 1 from Theorem 3 as $g$ is not $L$-Lipschitz is many applications. Moreover, Proposition 1 is only based on the Moreau envelope approximation study, independently from the Markov Chain context of this paper.

**Inexact PSGLA** In practice, many functions $g$, such as the Total-Variation regularisation [83], have no closed-form $\mathsf{Prox}_{\gamma g}$. Therefore, the proximal operator is approximated by an operator $S_\gamma \in \mathcal{C}^0(\mathbb{R}^d, \mathbb{R}^d)$ such that $S_\gamma \approx \mathsf{Prox}_{\gamma g}$. The authors of [33] studied the inexact PSGLA defined by

$$\hat{X}_{k+1} = S_\gamma \left( \hat{X}_k - \gamma \nabla f(\hat{X}_k) + \sqrt{2\gamma} Z_{k+1} \right), \tag{14}$$

with $Z_{k+1} \sim \mathcal{N}(0, I_d)$ i.i.d. We now extend the convergence result of [33, Theorem 3.14] obtained in a convex setting to the strong convexity *at infinity* assumption. To that end, we observe that the shadow sequence $\hat{Y}_k = \hat{X}_k - \gamma \nabla f(\hat{X}_k) + \sqrt{2\gamma} Z_{k+1}$ verifies equation (2) with the drift $\hat{b}^\gamma(y) = \nabla f(y - G_\gamma(y)) + G_\gamma(y)$, where $G_\gamma = \gamma^{-1}(I_d - S_\gamma)$.

**Theorem 4.** *Under Assumption 2-3, if $\hat{b}^\gamma$ verifies Assumption 1, there exist $\hat{\gamma} > 0$ and $C_5, C_6, C_7 \in \mathbb{R}_+$ such that $\forall \gamma \in (0, \hat{\gamma}]$, the shadow sequence $\hat{Y}_k$ has an invariant law $\hat{\mu}_\gamma$ and $\forall k \in \mathbb{N}$, the distribution $p_{\hat{X}_k}$ of $\hat{X}_k$ defined in equation (14) verifies*

$$\mathbf{W}_p(p_{\hat{X}_k}, \mu_\gamma) \leq C_5 r^{k\gamma} + C_6 \gamma^{\frac{1}{2p}} + \frac{C_7}{\gamma^{\frac{1}{p}}} \| S_\gamma - \mathsf{Prox}_{\gamma g} \|_{\ell_2(\hat{\mu}_\gamma)}^{\frac{1}{2p}}.$$

Theorem 4 is proved in Appendix G.4 by reformulating inexact PSGLA as a two-point algorithm and applying Theorem 3. Theorem 4 states that approximations in the proximal operator evaluation do not accumulate and imply a quantified shift in the PSGLA law $p_{\hat{X}_k}$.

## 4 Related Works

An alternative method to sample composite potential is Gibbs sampling that have been developed in [38, 88]. Gibbs sampler splits the sampling problem into two easier sampling sub-problems. More recently, the authors of [94, 17, 60] propose a Gibbs sampler that, combined with deep learning, provides accurate sampling. However their theoretical guarantees are only developed in the convex setting. A non-convex proof of a coarse Gibbs algorithm, without annealing, is proposed in [89].

For image applications, diffusion models [87] are another recent strategy to sample a law $\pi$ with convergence guarantees [11, 22]. Many works have focused on adapting diffusion models for image posterior sampling, see for instance [54, 15, 14, 82, 101]. However, to our knowledge, there is no drift stability results have been shown in this setting, therefore preventing the establishment of convergence results for posterior sampling. More details on related works are provided in Appendix A.

## 5 Application to posterior sampling

Recovering a signal $x \in \mathbb{R}^d$ from a degraded observation of this signal $y \in \mathbb{R}^m$, with typically $m < d$, is called inverse problem. In many settings, a degradation model is given, i.e. $y = \mathcal{A}(x) + n$, with $n$ some noise. Even though our approach is not limited to this setting, for the sake of simplicity, in this section we will focus on the linear inverse problem $y = Ax + n$ with $A \in \mathbb{R}^{m \times d}$ and additive Gaussian noise $n \sim \mathcal{N}(0, \sigma^2 I_d)$.

In a Bayesian paradigm, we suppose that $x$ is a realization of a random variable having the density $p(x)$, called *prior distribution*. Then, we aim to sample the distribution $p(x|y)$, called *posterior distribution*. The Bayes formula gives that $p(x|y) \propto p(y|x)p(x)$ and the physical model implies that $p(y|x) \propto e^{-\frac{1}{\sigma^2} \|Ax - y\|^2}$. By denoting $f(x) = \frac{1}{\sigma^2} \|Ax - y\|^2$ and $g(x) = -\log p(x)$, we get that

$$p(x|y) \propto e^{-V(x)},$$

with $V = f + g$ a composite potential. Thus, we can use inexact ULA (2) (with $b = \nabla f + \nabla g$) or PSGLA (3) (with $\nabla f$ and $\mathsf{Prox}_{\gamma g}$), to sample this posterior distribution.

**Plug-and-Play** Sampling the posterior distribution is a challenging task as it can be a high dimension problem (e.g. image inverse problems) as the prior distribution $p(x)$ is often unknown. Therefore, it is not possible to compute $\nabla g$ or $\mathsf{Prox}_{\gamma g}$. The authors of [92] proposed to replace $\mathsf{Prox}_{\gamma g}$ by

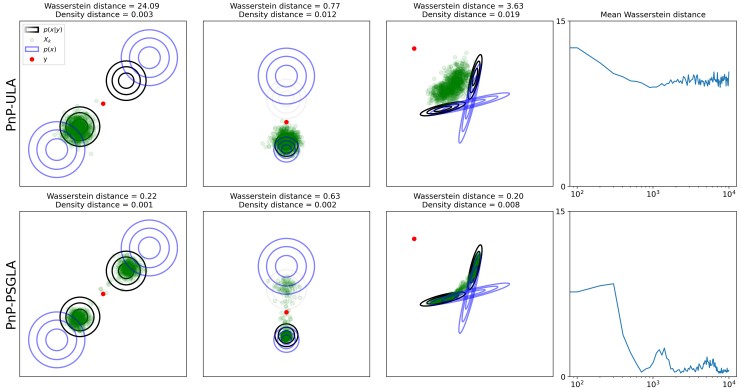

Figure 1: Posterior Sampling with PnP-ULA (top row) and PSGLA (bottow row) with three different Gaussian Mixture prior distributions (in blue) and three observations (in red) leading to three posterior distributions (in black, mode probabilities are represented by transparency). Algorithms are run with $10,000$ steps. Quantitative metrics, discrete Wasserstein distance and $L_2$ distance between the estimated density and the true density, are computed for each algorithm output. The Wasserstein distance during the iterations (right column) is averaged between the three experiments. Note that PnP-PSGLA succeeds to sample the posterior distribution even if there are various modes and the convergence is faster than PnP-ULA.

a pretrained denoiser $D_{\sqrt{\gamma}}$, learnt to remove additive Gaussian noise of standard-deviation $\sqrt{\gamma} > 0$. In fact, $\mathsf{Prox}_{\gamma g}(x) = \arg\min_{y \in \mathbb{R}^d} \frac{1}{2\gamma}\|x - y\|^2 + g(y)$ is the maximum-a-posteriori denoising with noise level $\sqrt{\gamma}$ and prior distribution $e^{-g}$. Replacing $\mathsf{Prox}_{\gamma g}$ by a pretrained denoiser $D_{\sqrt{\gamma}}$ is called *Plug-and-Play* (PnP) and $D_{\sqrt{\gamma}}$ implicitly encodes the prior knowledge.

We thus introduce the PnP-PSGLA algorithm defined by

$$X_{k+1} = D_{\sqrt{\gamma}}\left(X_k - \frac{\gamma}{\lambda}\nabla f(X_k) - \sqrt{2\gamma}Z_{k+1}\right),\tag{15}$$

with $\lambda > 0$ a regularization parameter, $D_{\sqrt{\gamma}}$ a pre-trained denoiser of noise-level $\sqrt{\gamma}$ and $Z_{k+1} \sim \mathcal{N}(0, I_d)$. This parameter $\lambda$ can be seen as a temperature parameters by replacing the prior $p$ by $p^\lambda$ which leads to sample a distribution proportional to $e^{-\frac{1}{\lambda}f - g}$ instead of $e^{-f - g}$. This parameter allows more flexibility in practice to improve PnP-PSGLA performance. Note that at each iteration, we inject $\sqrt{2}$ times more noise than we denoise; therefore, we can expect the algorithm to explore the space. If we injected as much noise as we denoise, we would expect the algorithm to behave like a stochastic optimization method rather than a sampling algorithm as detailed in [58, 76, 77].

## 6 Experiments

In this section, we present numerical validation of PnP-PSGLA (15) for posterior sampling. More details and additional experiments are provided in Appendix C.

**Gaussian Mixture in 2D** In order to test experimentally the quality of PnP-PSGLA, compared to PnP-ULA [59], we first look at a simplified case with a Gaussian Mixture prior in 2D. PnP-ULA is a Plug-and-Play version of the ULA (10). In this case, the posterior distribution, $\nabla g$ and the MMSE denoiser can be written in closed form. Therefore, we can run and compare the sampling algorithms PnP-PSGLA and PnP-ULA with the exact prior information.

As illustrated in Figure 1, PnP-PSGLA outperforms PnP-ULA both quantitatively and qualitatively on this Gaussian mixture problem. More precisely, PnP-ULA fails to discover minor modes of the posterior distribution in $N = 10^4$ steps, whereas PnP-PSGLA successfully identifies them. Our experiments suggest that the mixing time of PnP-PSGLA is shorter than for PnP-ULA. If the proximal operator is interpreted as a projection operator, PnP-PSGLA rigidly project the iterates on the "clean data" manifold, whereas PnP-ULA softly guides the iterates. This might help PnP-PSGLA to discover modes faster that PnP-ULA.

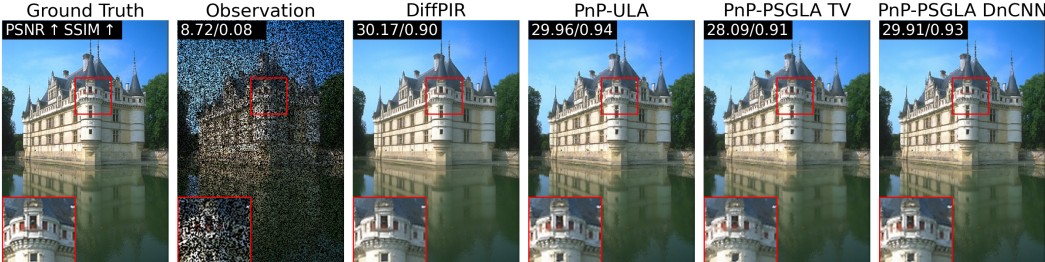

Figure 2: Qualitative result for image inpainting with 50% masked pixels and a noise level of $\sigma = 1/255$. PnP-ULA is run with 1,000,000 iterations and PSGLA with 10,000 iterations.

| Algorithm | Denoiser | PSNR ↑ | SSIM ↑ | LPIPS ↓ | N ↓ | time (s) ↓ | convergent |
|---|---|---|---|---|---|---|---|
| DiffPIR [101] | GSDRUNet | 29.99 | 0.88 | 0.06 | **20** | **1** | ✗ |
| RED [81] | DnCNN | 30.49 | 0.89 | 0.06 | 500 | 6 | ✓ |
| RED [81] | GSDRUNet | 29.26 | 0.88 | 0.12 | 500 | 20 | ✓ |
| PnP [92] | DnCNN | 30.50 | 0.91 | 0.06 | 500 | 6 | ✓ |
| PnP [92] | GSDRUNet | 30.52 | **0.92** | 0.07 | 500 | 20 | ✓ |
| PnP-ULA [59] | DnCNN | 27.89 | 0.82 | 0.12 | 100,000 | 1,200 | ✓ |
| PnP-PSGLA [33] | TV | 29.24 | 0.89 | 0.08 | 1,000 | 25 | ✓ |
| PnP-PSGLA | DnCNN | **30.81** | **0.92** | **0.05** | 10,000 | 120 | ✓ |

Table 2: Quantitative results for image inpainting with 50% masked pixels on CBSD68 dataset. The number of inference of the denoiser $N$ and the time of restoration in second on one GPU NVIDIA A100 Tensor Core are given for one image. Best and second best results are respectively displayed in bold and underlined. Note that PnP-PSGLA (15) has comparable performances with state-of-the art methods as DiffPIR and PnP and is significantly faster that PnP-ULA.

**Image restoration with Plug-and-Play approach**  We use the Denoising Convolutional Neural Network (DnCNN) denoiser proposed by the authors of [71] and trained to be firmly non-expansive This denoiser has been trained for one level of noise $\sqrt{\gamma} = 2/255$, which fixes the step-size $\gamma$ of PSGLA. We also considered the proximal deep neural network denoiser of [48], based on the DRUNet architecture [99]. Interestingly, we found that the choice of denoiser greatly influences the quality of the result. As DnCNN provided much better results than DRUnet we consistently use this architecture and report DRUNet restoration results in Appendix C.2.1.

In Figure 2, we present restorations with different methods and denoisers. DiffPIR [101] is a diffusion-inspired algorithm to restore images. It uses the denoiser with various noise levels, excluding the use of the DnCNN network [71] trained with only one level of noise. Plug-and-Play Unajusted Langevin Algorithm (PnP-ULA) [59] is a Plug-and-Play adaptation of the Unajusted Langevin Algorithm (10) that is proved to sample the posterior distribution. PnP-PSGLA is run with two denoisers. The Total Variation (TV) denoiser [83] is a classical denoising method that leads to staircasing effects and over-smoothes the denoised image. We use TV-denoiser as a baseline with limited computational time. PnP-PSGLA is run with the DnCNN denoiser proposed by [71] to improve the restoration quality. Note that PnP-PSGLA, when run with DnCNN, achieves good perceptual quality using 100 times fewer steps than PnP-ULA. We again interpret this acceleration by the ability of PnP-PSGLA to rigidly project the iterates whereas PnP-ULA softly guides them.

On Table 2, we present quantitative comparisons between various restoration methods on the CBSD68 dataset [64]. Results are presented with two distortion metrics, Peak Signal to Noise Ratio (PSNR) and Structural Similarity (SSIM) [95], and one deep perceptual metric, Learned Perceptual Image Patch Similarity (LPIPS) [100]. Regularization by Denoising (RED) [81] and Plug-and-Play (PnP) [92] are restoration algorithms that solve an optimization problem to find the most probable image knowing the degraded observation. The Gradient-Step DRUNet (GSDRUNet) [47] denoiser, based on the DRUNet architecture [99], is a deep neural network denoiser that ensures convergence for RED and PnP algorithms. For PnP-ULA and PnP-PSGLA, the mean of the iterates is taken as an estimator of the expectation of the posterior distribution. Note that PnP-PSGLA with DnCNN achieves performances that are competitive with state-of-the-art methods. Moreover,

the PnP-PSGLA achieves better performances that PnP-ULA in less iterations. PnP-PSGLA with DnCNN requires more computational resources than DiffPIR, PnP or RED. Using the TV-denoiser is an alternative to accelerate PNP-PSGLA at the expense of visual performance.

# 7 Conclusion

In this paper, we aim to sample a distribution $\pi \propto e^{-V}$ with non-convex potential $V$. We improve existing results for the stability (Theorem 1 in Section 2) and the discretization error (Theorem 2 in Section 2) of iULA (2). We use these results to derive the first convergence proof of PSGLA (3) for non-log-concave sampling (Theorem 3 in Section 3). We apply our theoretical findings to posterior sampling and show the experimental benefit, especially in computational time, of PSGLA with respect to the original iULA. We also propose a Plug-and-Play version of PSGLA (PnP-PSGLA) that achieves state-of-the art image restoration.

PnP-PSGLA is limited by its computational time, as is the case with most Markov Chain Monte Carlo (MCMC) methods. Annealing methods [89] or more efficient denoisers could offer promising directions for future works to reduce the computational time of PnP-PSGLA. From a theoretical point of view, we identify two main limitations of this work that might motivate future research. The first one is to obtain theoretical insights, such as accelerated convergence rate, of the superiority of PSGLA over ULA. The second one is to analysis the tightness of the constants involve in Theorem 1-2 that are partially implicit in the current work. Convergent posterior sampling algorithms could have a positive societal impact by providing image restoration with quantified uncertainty. Also, such a deep image restoration algorithm could also have negative societal impacts as it may hallucinate details and might be used to generate false information.

## Acknowledgements

This study has been carried out with financial support from the French Direction Générale de l'Armement and the French National Research Agency through Projects PEPR PDE-AI, ANR-19-CE40-005 MISTIC and ANR-23-CE40-0017 SOCOT. Experiments presented in this paper were carried out using the PlaFRIM experimental testbed, supported by Inria, CNRS (LABRI and IMB), Université de Bordeaux, Bordeaux INP and Conseil Régional d'Aquitaine (see https://www.plafrim.fr). We thank Andrés Almansa, Pascal Bianchi, Kaplan Desbouis, Gersende Fort, Erell Gachon, Julien Hermant, Samuel Hurault, Rémi Laumont, Éloan Rapion and Adrien Richou for their time and discussions.

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

# Appendix

In the Appendix, we provide all the proof for the theoretical results of the paper, additional details on experiments and more intuition on the theoretical objects that appears in the paper. In Appendix A, we detail related works on Langevin MCMC with a focus on Proximal Langevin methods. In Appendix B, we discuss each assumptions in details. In Appendix C, we provide additional experiments and implementation details. In Appendix D, we give a self-contained analysis of the Moreau envelope in the weakly convex setting. In Appendix E, we discuss our notion of strong convexity on non-convex sets. In Appendix F, we prove all the results of Section 2 and give intuition on Markov Chain and Markoc Process analysis. In Appendix G, we prove the convergence results of Section 3 for PSGLA.

# A   Related works

In this part, we detail related works on Langevin Diffusion with a focus on methods involving proximal steps.

**On the stability of inexact Unajusted Langevin Algorithm**   The authors of [98] study the stability of inexact Unajusted Langevin Algorithm (iULA defined in equation (2)) with respect to the KL divergence. This study can be seen as complementary to our analysis developed in $p$-Wasserstein and Total Variation distance. Note that our strategy does not generalize easily to KL divergence as it relies massively on the triangle inequality that is not verified by the KL divergence. The proof strategy of [98] relies on the log-Sobolev inequality for the potential $V$ and bounds between the inexact drift $b$ and $\nabla V$ making this stability analysis less general than our analysis. The stability of iULA has been analysed in the same setting than ours by the authors of [75]. However, their stability analysis is only provided in the 1-Wasserstein and Total Variation distances, and yields a coarser bound than Theorem 1, which in particular implies a discretization error.

**PnP-ULA**   The authors of [59] propose a Plug-and-Play strategy based on inexact ULA and the Tweedie formula [32], named PnP-ULA, and defined by

$$X_{k+1} = X_k - \gamma \nabla f(X_k) - \frac{\gamma \lambda}{\epsilon} \left( X_k - D_\epsilon(X_k) \right) - \frac{\gamma}{\alpha} \left( X_k - \Pi_{\mathbf{S}}(X_k) \right) - \sqrt{2\gamma} Z_{k+1}, \quad (16)$$

with $Z_{k+1} \sim \mathcal{N}(0, I_d)$, $D_\epsilon$ a pretrained denoiser of noise-level $\epsilon > 0$, $\Pi_{\mathbf{S}}$ the projection on the convex set $\mathbf{S} \subset \mathbb{R}^d$ and $\lambda > 0$ the regularization weight. The term $\frac{\lambda}{\epsilon} \left( X_k - D_\epsilon(X_k) \right)$ is used to encode the prior information. The projection term $\frac{1}{\alpha} \left( X_k - \Pi_{\mathbf{S}}(X_k) \right)$, controlled by the weight parameter $\alpha > 0$, is added to ensure that the sampled potential is strongly convex at infinity for convergence properties. This Markov Chain has a slow mixing time as shown on Figure 5 where it needs more than $5.10^5$ iterations to converge.

**MYULA**   The authors of [70, 28] propose the Moreau-Yoshida Unadjusted Langevin Algorithm (MYULA) defined by

$$X_{k+1} = X_k - \gamma \nabla f(X_k) - \frac{\gamma}{\lambda} \left( X_k - \mathsf{Prox}_{\lambda g}(X_k) \right) + \sqrt{2\gamma} Z_{k+1}$$

to sample $e^{-f-g^\lambda}$, with $g^\lambda$ the Moreau transform of $g$. This algorithm is based on the Moreau envelope approximation (see more details in Appendix D) to approximate $\nabla g$ using the proximal operator $\mathsf{Prox}_{\lambda g}$. It is particularly relevant if the proximal operator on $g$ can be computed easily. Contrary to PSGLA, in MYULA, the noise is added outside the proximal operator. Recently the authors of [55] consider algorithms based on MYULA to accelerate Langevin samplers. Note that if a pre-trained denoiser is plugged in place of the proximal operator, then we recover PnP-ULA.

**Gibbs sampling** The authors of [94, 17] propose another strategy to sample a composite function, named Gibbs sampling. A Gibbs sampler [38, 88] splits the sampling problem into two easier sampling sub-problems. The first sub-problem encodes the prior information and the second the degradation model, following the Plug-and-Play idea. However no convergence guarantee is given for this algorithm. The authors of [89] propose a similar algorithm with some theoretical results in the non-convex setting. However, they employ annealing techniques, and it is unclear how these affect convergence. Finally, the authors of [60] propose another Gibbs sampler and derive convergence guarantees only in the case of convex potentials.

**Proximal Langevin Monte Carlo** The authors of [5] study the Proximal Langevin Monte Carlo (PLMC) sequence defined by

$$X_{k+1} = \mathsf{Prox}_{\gamma g}(X_k) + \sqrt{2\gamma}Z_{k+1}$$

to sample $e^{-g}$ with $g$ convex everywhere. The authors of [4] study the same sequence and focus on developing convergence results beyond the Lipschitz Gradient continuity. PLMC is a particular case of MYULA in the case of $\lambda = \gamma$. A Metropolis-Hasting mechanism thas also been proposed in [72, 19] to accelerate PLMC or MYULA in the case of convex potentials.

**Projected Langevin** A particular case of PSGLA, where the function $g$ is the characteristic function of a convex set $\mathbf{K}$, leads to the projected Langevin algorithm defined by

$$X_{k+1} = \mathcal{P}_{\mathbf{K}}(X_k - \gamma\nabla f(X_k) + \sqrt{2\gamma}Z_{k+1}),$$

which was introduced over a decade ago [9, 44]. This paper studies the convergence in the log-concave setting, i.e. if the potential $f$ is convex. The author of [56] studies a generalization of this algorithm when $\nabla f$ is estimated randomly. In this paper, convergence guarantees are provided if $\mathbf{K}$ is convex and $\nabla f$ is Lipschitz and perturbed by Gaussian noise.

**Symmetrized Langevin Algorithm** The Symmetrized Langevin Algorithm (SLA) defined by $X_{k+1} = \mathsf{Prox}_{\gamma V}(X_k - \gamma\nabla V(X_k) + \sqrt{4\gamma}Z_{k+1})$ has been proposed by the authors of [97]. This algorithm does not sample from a composite potential, but instead targets the distribution proportional to $e^{-V}$. This algorithm is studied for the 2-Wasserstein metric in the case of a strongly convex potential $V$.

**Other articles on PSGLA** The PSGLA (3) was first introduced in [29], with an analysis for strongly convex potentials $V = f + g$. The authors of [85] study a generalization of PSGLA with multiple proximal operators applied in sequence. Their analysis also assume that $f$ is strongly convex and $g$ convex. Another convergence proof with $f$ strongly convex and $g$ convex was derived by making an analogy between PSGLA and the forward-backward scheme [84]. If the proximal mapping can only been approximated, the authors of [33] show that the algorithm is stable in the case of convex potentials $V$. Recently, a particle interaction algorithm based on PSGLA has been proposed [34] to improve the sampling quality. The convergence of this particle interaction algorithm have been derived in the strongly convex setting.

## B  Detailed discussion on assumptions

In this section, we discuss in details the different assumptions made in the paper.

- **Assumption 1(i)**. Assumption 1(i) is standard to obtain the existence of the strong solution of the stochastic differentiable equation (1), see [53, Theorem 2.5] for instance. It implies in particular that the drift $b$ is sub-linear, i.e. there exists $\|b(x)\| \leq \|B(0)\| + L\|x\|$. Moreover, if $b = \nabla V$, the potential $V$ is $L$-weakly convex. Assumption 1(i) is equivalent to $-LI_d \preceq \nabla b \preceq LI_d$.

- **Assumption 1(ii)**. It is called weak dissipativity [24, 62] or distant dissipativity [67] in the stochastic differentiable equation community. In order to have a geometric ergodicity of the Markov Chain, it is necessary to control the drift in the tails of the sampled distribution. Assumption 1(ii) is a relaxation of the strong convexity of $b$ that allows to control the behavior of the MCMC at infinity. In particular, having a form of strong convexity at infinity of the drift, with Assumption 1(ii), ensures that the MCMC is bounded and has all its moments bounded [23, 59]. If the drift is $m$-strongly convex at infinity and $L$-weakly convex then it is weakly dissipative.

- **Assumption 2(i)**. It is verified for convex functions $f$ such as negative log-likelihood terms associated to linear inverse problems with additive Gaussian noise $f(x) = \|Ax - y\|^2$ or non-convex $f$, as it is the case in image despeckling [25] or black-hole interferometric imaging [89]. This condition is standard in posterior sampling [59, 75, 55] or in imaging inverse problems [76, 96].

- **Assumption 2(ii)**. Many works have focused on convex regularizations [83, 71, 37]. However, it is known that the non-convexity allows to made better restoration [47]. Therefore, we only assume in this paper that the regularization is $\rho$-weakly convex. It ensures that the proximal operator $\mathsf{Prox}_{\gamma g}$ is defined for $\gamma\rho < 1$. Assumption 2(ii) is then verified for all convex regularization and some state-of-the art deep neural networks regularizers [71, 47, 86, 40].

- **Assumption 3(i)**. Assumption 3(i) can be verified by non-smooth functions $g$ such as the characteristic function of a convex set. It is not usual to assume that the function which is a proximal operator is smooth on a subset of $\mathbb{R}^d$ [84, 29, 33]. However, the Moreau envelope study leads naturally to look at the behavior of $g$ on the set $\mathsf{Prox}(\mathbb{R}^d)$ as detailed in Appendix D.7. Moreover, the deep neural network denoiser proposed in [48] has been demonstrated to verify Assumption 3(i). Assumption 3(i) ensures that the drift $b^\gamma$ is Lipschitz to verify Assumption 1(i) (based on Lemma 15 in Appendix D.7). Note that $\mathsf{Prox}_{\gamma g}\left(\mathbb{R}^d\right)$ can be a non-convex set if $g$ is not finite everywhere. However, Proposition 2 in Appendix D.7 shows that $\mathsf{Prox}_{\gamma g}\left(\mathbb{R}^d\right)$ only differs from being convex by a negligible set.

- **Assumption 3(ii)** Assumption 3(ii) implies that the potential $V$ is strongly convex at infinity, which ensures that the drift $b^\gamma$ verifies Assumption 1(ii). Lemma 16 in Appendix E shows that if $g$ is strongly convex at infinity and *globally* smooth, then Assumption 3(ii) is verified. Assumption 3(ii) is technical and hard to verify in practice but it is important to study PSGLA. An alternative to this assumption is to modify PSGLA by adding a projection term as in [59] leading to the Projected PSGLA

$$X_{k+1} = \mathsf{Prox}_{\gamma g}\left(X_k - \gamma\nabla f - \frac{\gamma}{\alpha}\left(X_k - \Pi_{\mathbf{K}}(X_k)\right) + \sqrt{2\gamma}Z_{k+1}\right),$$

with $\Pi_{\mathbf{K}}$ the projection on the convex set $\mathbf{K} \subset \mathbb{R}^d$. As detailed in [59, Appendix F.2], by choosing a small enough $\alpha > 0$, this additional projection term ensures that the drift of the shadow sequence of Projected PSGLA $\nabla f(y - \gamma\nabla g^\gamma(y)) + \nabla g^\gamma + \frac{1}{\alpha}\left(y - \gamma\nabla g^\gamma(y) - \Pi_{\mathbf{K}}\left(y - \gamma\nabla g^\gamma(y)\right)\right)$ verifies Assumption 1(ii). Therefore, it is possible to demonstrate the convergence of Projected PSGLA with a small enough parameter $\alpha > 0$ without Assumption 3(ii). Note that if $X_k$ is an image, typically in $[0, 1]^d$, then choosing $K = [-1, 2]^d$ as in [59] involves the projection term being rarely activated. However, in this paper, we chose not to modify the PSGLA, so Assumption 3(ii) is necessary.

## C  More details on experiments

### C.1  Experiments with Gaussian Mixture prior in 2D

We suppose that the prior is a Gaussian Mixture, that can be expressed as

$$p(x) = \sum_{i=1}^{p} p_i \mathcal{N}(x; m_i, \Sigma_i),$$

with $\mathcal{N}(x; m_i, \Sigma_i) = \frac{1}{(2\pi|\Sigma_i|)^{\frac{d}{2}}} e^{-\frac{1}{2}(x-m_i)^T \Sigma_i^{-1}(x-m_i)}$ and $\Sigma_i \in \mathbb{R}^{d \times d}$ a symmetric positive-definite covariance matrixcovariance matrix, $m_i \in \mathbb{R}^d$ the mean of the distribution and $p_i \in [0, 1]$ are the mode weights, such that $\sum_{i=1}^{p} p_i = 1$.

Then the gradient of the log prior is

$$\nabla \log p(x) = -\frac{\sum_{i=1}^{p} p_i \Sigma_i^{-1}(x - m_i)\mathcal{N}(x; m_i, \Sigma_i)}{\sum_{i=1}^{p} p_i \mathcal{N}(x; m_i, \Sigma_i)}.$$

Thanks to the Tweedie formula [32], the Minimum Mean Square Error (MMSE) denoiser $D_\epsilon$ can be computed by

$$D_\epsilon(x) = x - \epsilon^2 \nabla \log p_\epsilon(x),$$

with $\epsilon$ the noise level and $p_\epsilon = p \star \mathcal{N}(0, \epsilon^2 I_d)$ the convolution between $p$ and the centered Gaussian kernel $\mathcal{N}(0, \epsilon^2 I_d)$.

Due to the specific form of $p$, we know that

$$\nabla \log p_\epsilon(x) = -\frac{\sum_{i=1}^{p} p_i (\Sigma_i + \epsilon^2 I_d)^{-1}(x - m_i)\mathcal{N}(x; m_i, \Sigma_i + \epsilon^2 I_d)}{\sum_{i=1}^{p} p_i \mathcal{N}(x; m_i, \Sigma_i + \epsilon^2 I_d)}.$$

Thus the closed-form expression of the MMSE denoiser is

$$D_\epsilon(x) = x + \epsilon^2 \frac{\sum_{i=1}^{p} p_i (\Sigma_i + \epsilon^2 I_d)^{-1}(x - m_i)\mathcal{N}(x; m_i, \Sigma_i + \epsilon^2 I_d)}{\sum_{i=1}^{p} p_i \mathcal{N}(x; m_i, \Sigma_i + \epsilon^2 I_d)}.$$

For Gaussian Mixture prior, the gradient of the regularization $-\log p$ and the denoiser $D_\epsilon$ are known in closed-form. Therefore, we can compute PnP-ULA and PnP-PSGLA exactly.

**Parameters setting**  We run PnP-ULA and PnP-PSGLA for a 2D denoising problem, i.e. $d = m = 2$, $A = I_2$ and $\sigma = 1$. PnP-ULA defined in equation (16) is run with a step-size $\gamma = 0.1$, a regularization parameter $\lambda = 1.5$ and a denoiser parameter $\epsilon = 0.5$. In this case, the sampled distribution is log-concave at infinity. Therefore, we choose to set $\alpha = +\infty$, i.e. there is no projection. PSGLA is run with a step-size $\gamma = 0.3$ and a regularization parameter $\lambda = 0.67$. These parameters have been chosen to maximize the performance of each method.

**Metrics**  We compute two quantitative metric between the discrete uniform measure supported on the iterates and the exact posterior. The first one is the discrete Wasserstein distance computed using the Python Optimal Transport (POT) library [35]. The second one is the $\ell_2$ distance between the empirical density estimated using Gaussian kernels and the exact posterior density. This $\ell_2$ distance is compute on a square including most of the density ($[-8, 8]^2$ in our experiments).

### C.2  More details on image restoration experiments

In this section, we present more details on the experiments for image restoration. First we recall the definition of each compared method. Then we give the parameters setting and additional experiments.

**PnP**  We compare PSGLA into method that are mixing information from learning and the physics of degradation, named Plug-and-Play (PnP). This algorithm has been proposed for image restoration by [92] and is defined by

$$X_{k+1} = D_\epsilon \left( X_k - \frac{\gamma}{\lambda} \nabla f(X_k) \right).$$

The authors of [71] shows that with a maximal monotone DnCNN denoiser PnP converges. The convergence of this algorithm have been extensively studied, see a detailed review in [47, Part 2]. This algorithm have been design to optimize and compute an approximation of the most probable image knowing the degraded observation. Thus, PnP is not sampling the posterior distribution and there is no information about uncertainty that can be extracted from this algorithm.

| Parameters | RED GSDRUNet | RED DnCNN | PnP GSDRUNet | PnP DnCNN | PnP-ULA DnCNN | PSGLA TV | PSGLA DnCNN |
|---|---|---|---|---|---|---|---|
| $255 \times \epsilon$ | 7 | 2 | 5 | 2 | 2 | 10 | 2 |
| $\lambda$ | $70,000$ | $150,000$ | $0.5$ | $1$ | $\frac{1}{\frac{2}{\epsilon^2}+\frac{1}{\epsilon^2}}$ | 10 | 5 |
| $\gamma$ | $1.10^{-5}$ | $1.10^{-5}$ | $1.10^{-5}$ | $1.10^{-5}$ | $\frac{1}{3}\times\left(\frac{1}{\sigma^2}+\frac{1}{\lambda}+\frac{1}{\epsilon^2}\right)$ | $\epsilon^2$ | $\epsilon^2$ |

Table 3: Parameters setting for image inpainting for the different implemented methods. $\epsilon$ is the noise level of the denoiser, $\lambda$ the regularization parameter, $\gamma$ the step-size. The parameter settings for PnP-ULA are those suggested in the original paper [59].

**RED** Another optimization algorithm based mixing prior information and physical information is Regularization by Denoising (RED). This algorithm has been proposed for image restoration by [81] and is defined by

$$X_{k+1} = X_k - \gamma \nabla f(X_k) - \gamma \lambda \left( X_k - D_\epsilon(X_k) \right).$$

The authors of [47] shows that with a Gradient-Step denoiser RED converges.

**DiffPIR** DiffPIR definition is detailed in Algorithm 2. This algorithm is based on a diffusion model method applied to image restoration. To our knowledge, no theoretical analysis of this algorithm has been provided and it is not clear if this algorithm is solving an optimization problem or sampling a target law. This algorithm requires a denoiser trained with various level of noise. We use the Gradient-Step DRUNet in our experiment with the pre-trained weights provided by the authors of [47]. The precomputed parameters $(\bar{\alpha}_t)_{0<t<T}$, $(\sigma_t)_{0<t<T}$ and $(\rho_t)_{0<t<T}$ are set as proposed in [101]. In our experiments for image inpainting, we set $T = 1000$, $t_{\text{start}} = 200$, $\lambda = 0.13$, $\zeta = 0.999$. These parameters have been chosen after running a grid-search to maximize the PSNR on the 3 natural images (butterfly, leaves and starfish) from the set3c dataset [47].

---

**Algorithm 2** DiffPIR [101]

---

1: **input:** denoiser $D$, $T > 0$, $y \in \mathbb{R}^m$, $0 < t_{\text{start}} < T$, $\zeta > 0$, $(\beta_t)_{0<t<T}$, $\lambda > 0$
2: Initialize $\epsilon_{t_{\text{start}}} \sim \mathcal{N}(0, I_d)$, pre-calculate $(\bar{\alpha}_t)_{0<t<T}$, $(\sigma_t)_{0<t<T}$ and $(\rho_t)_{0<t<T}$
3: $x_{t_{\text{start}}} = \sqrt{\bar{\alpha}_{\text{start}}} y + \sqrt{1 - \bar{\alpha}_{\text{start}}} \epsilon_{t_{\text{start}}}$
4: **for** $t = t_{\text{start}}, t_{\text{start}} - 1, \ldots, 1$ **do**
5: $\quad x_0^t \leftarrow D_{\sigma_t}(x_t)$
6: $\quad \hat{x_0}^t \leftarrow \text{Prox}_{2f(\cdot)/\rho_t}(x_0^t)$
7: $\quad \hat{\epsilon} \leftarrow \left( x_t - \sqrt{\bar{\alpha}_t} \hat{x_0}^t \right) / \sqrt{1 - \bar{\alpha}_t}$
8: $\quad \epsilon_t \leftarrow \mathcal{N}(0, I_d)$
9: $\quad x_{t_1} \leftarrow \sqrt{\bar{\alpha}_t} \hat{x_0}^t + \sqrt{1 - \bar{\alpha}_t} \left( \sqrt{1 - \zeta} \hat{\epsilon} + \sqrt{\zeta} \epsilon_t \right)$
10: **end for**

---

**Parameters setting** In Table 3, we provide the parameter choices used in our experiments for the implemented methods.

**Computational time** In Table 2, we give the mean computation time to restore one image on a GPU NVIDIA A100 Tensor Core. The total computational time to generate this table is thus around 27 hours of computational time on a GPU NVIDIA A100 Tensor Core. Note that the PnP-ULA requires most of the computational resources for a poor restoration quality. The total computational time for PnP-PSGLA with DnCNN to restore the CBSD68 dataset is 136 minutes. The computational cost of the entire project, including the parameter grid search, has not been rigorously computed. We estimate it to be around 40 hours of computational time on an NVIDIA A100 Tensor Core GPU.

**On the denoiser choice** In this paper, we only use pre-trained denoisers with open-source weights. We focus on a few denoisers that are known to be stable and provide good restoration. First the DnCNN with the weights trained by [71] is a maximal monotone operator, thus it is a the proximal operator of some convex regularization $g$. Therefore our theory (Theorem 3) applies with this denoiser. We also run the methods with the Gradient Step denoiser [47], a denoiser based on the DRUNet architecture [99], that is known to provide state-of-the-art restoration for Plug-and-Play methods. As this denoiser is trained on natural images with various noise level $\epsilon \in [0, 50]/255$, we

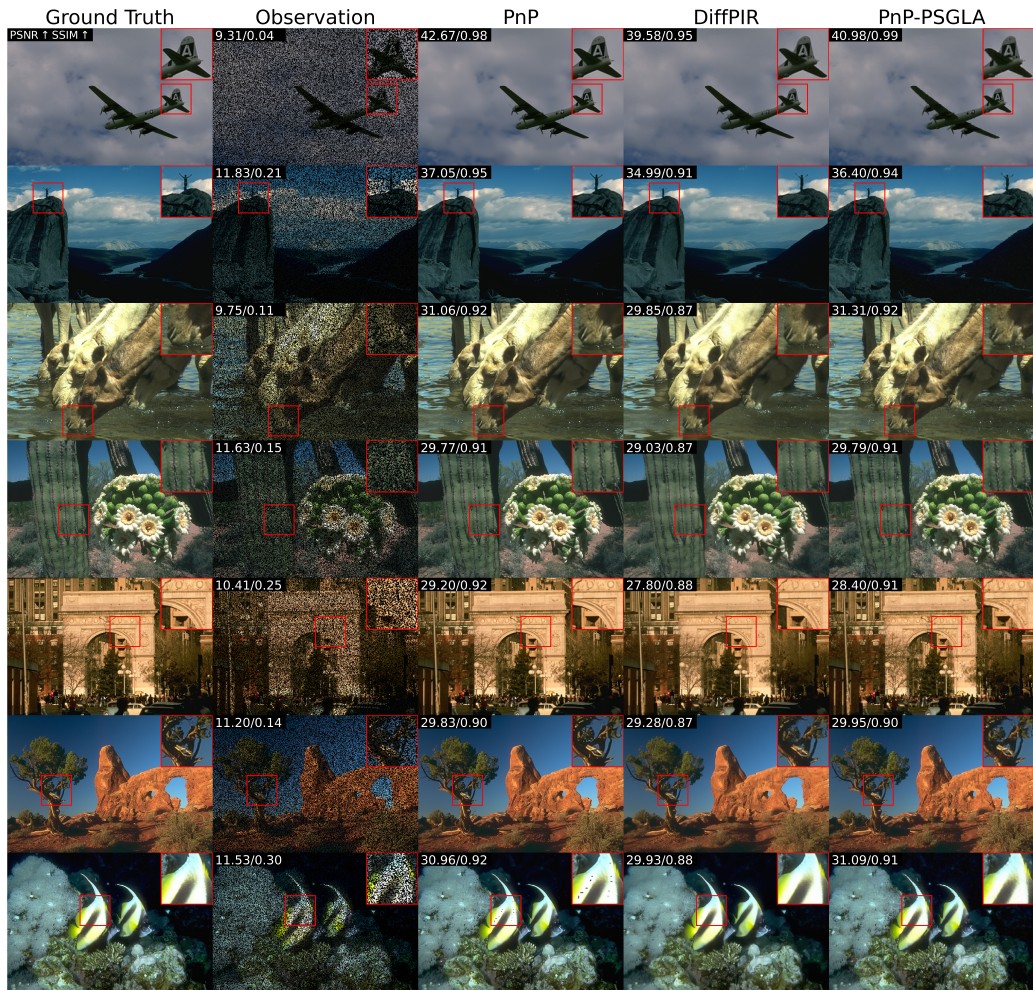

Figure 3: Qualitative result for image inpainting with 50% masked pixels and a noise level of $\sigma = 1/255$ with PnP, DiffPIR and PnP-PSGLA methods.

can use it for the DiffPIR algorithm. The open access weights of the DnCNN and the GSDRUNet used in our experiment for natural color images can be found in the library DeepInv [90] in this link. In order to reduce the computational time of PnP-PSGLA, we also use the $TV$-denoiser [83]. The Total-Variation regularization has not a closed form proximal operator so we use an iterative algorithm (implemented in the library DeepInv [90]) with 10 inner iterations, which is enough in our experiments to obtain the best restoration with $TV$-denoiser. By using such an approximation of PnP-PSGLA with the Total Variation denoiser, we recover the restoration algorithm proposed in [33].

### C.2.1 Why we do not use Prox-DRUNet with PSGLA

Based on the architecture of Gradient-Step DRUNet, the authors of [48] proposed a denoiser that is a proximal operator. This proximal operator satisfies Assumption 2(ii) and Assumption 3(i). The DRUNet architecture [99] is also known to provide a state-of-the-art denoiser. Thus Prox-DRUNet seems to be an ideal denoiser candidate for applying our theory and having empirical performances. However, we observe in our experiments some instabilities with the weight provided by the authors [48].

On Figure 4, we observe the restored image and the standard deviation of the Markov Chain run twice on the same observation with two seed in the randomness of the PSGLA. A different area of

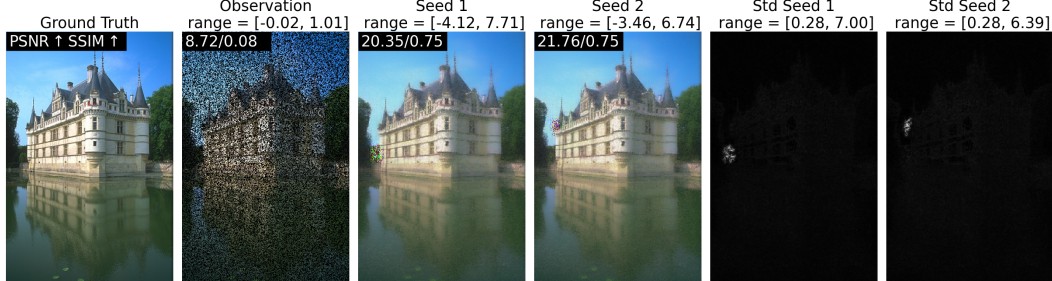

Figure 4: Restoration of PSGLA with Prox DRUNet for $50\%$ missing pixels and $\sigma = 1/255$. We chose the aggressive parameters $\epsilon = 50/255$, $\lambda = 5000$ to over-smooth the solution for more stability. PSGLA is run twice on the same observation with two seed in the randomness of the algorithm. Note that the Markov Chain explore outside the image space $[0, 1]^d$ and the instability depend of the realization of the noise. This algorithm is not stable.

the image is not restored properly and the Markov Chain explores outside the image space $[0, 1]^d$ with a large standard deviation in these areas. The mosaic motif that appears may be due to periodic colorization outside $[0, 1]^d$. The fact that the problematic areas depend of the realization of the noise indicates that the issue lies with the instability of the algorithm, not with the conditioning of the target distribution. Then, adding a projection in $[0, 1]^d$ at each iteration of $X_k$ does not prevent from these instabilities. Moreover, these instabilities appear for all the parameters setting that we test in our large grid-search. Finally, compare with DnCNN, trained with a fix level of noise, PSGLA with Prox-DRUNet has more parameters to fine-tune. For all these arguments, we chose to not use PSGLA with the Prox-DRUNet denoiser.

### C.2.2 PSGLA seems to effectively sample the posterior law

For image inverse problem, as the posterior distribution is unknown, it is not possible to evaluate if an algorithm effectively samples the posterior distribution. A standard strategy [59, 55] to have an indication of the sampling performance of a particular algorithm is to look at the standard-deviation of the Markov Chain.

In Figure 5, we present the mean and the standard deviation of the PSGLA Markov Chain for different numbers of iterations. We observe that when the Markov Chain is too short ($N = 10^4$) the mean is relevant but the standard deviation is not informative. On the other hand, when the number of iterations is large ($N = 10^6$), the standard deviation is both informative and relevant. One can thus observe that the image edges are more uncertain than uniform areas, such as the sky. The posterior distribution in image inverse problems is known to concentrate uncertainty in the edges [59, 55]. This is a good indication that the PSGLA might sample the posterior distribution in practice. In order to compute relevant uncertainty information, $N = 10^5$ seems to be a good compromise between computational efficiency and accuracy.

In Figure 6, we show the standard deviation of the MMSE estimator. We observe that the MMSE estimator has a small standard deviation compared with the standard deviation of the PSGLA Markov Chain. Moreover the standard deviation of the MMSE estimator decreases with the number of iterations. This observation support the fact the MMSE estimator is deterministic and that the PSGLA effectively solves a sampling problem.

### C.2.3 PnP-ULA is slow to converge

In Table 2, we observe that the performance of PnP-ULA is significantly lower than that of the other methods. However, the authors of [59, 75] show that PnP-ULA can provide good restoration. This apparent paradox is explained by the fact that PnP-ULA does not succeed to converge in $N = 10^5$ iterations. On Figure 5, we observe that the Markov Chain indeed converges in approximately $5.10^5$ iterations. In order to save computational resources, we choose to not run with such a large number of iterations PnP-ULA on the whole CBSD68 dataset to generate Table 2.

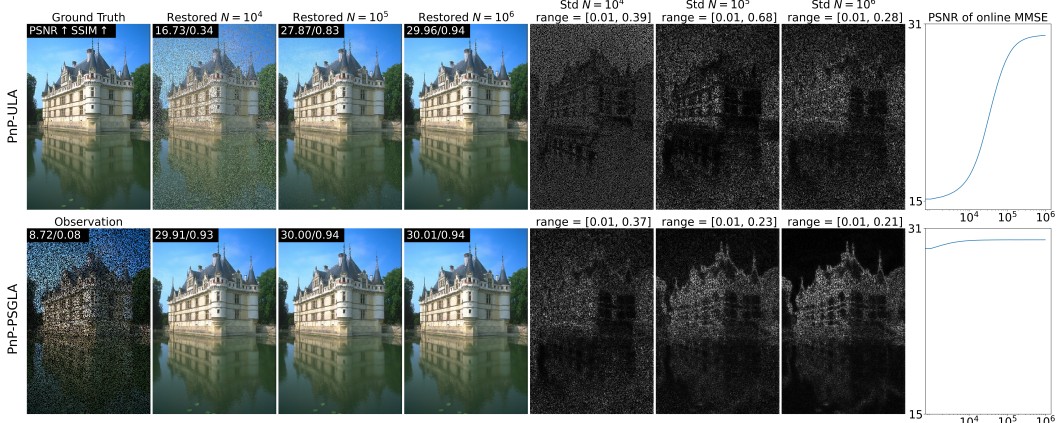

Figure 5: PnP-ULA and PSGLA with DnCNN for $50\%$ missing pixels and $\sigma = 1/255$ with various number of iterations $N \in \{10^4, 10^5, 10^6\}$. The standard deviation of the Markov Chains are shown for each number of iterations and the evolution of the PSNR for $N \in [0, 10^6]$. Note that the PnP-ULA Markov Chain is slow to converge. The effective convergence seems to occur around $5.10^5$ iterations. Note that the PSGLA Markov Chain seems to well explore the posterior distribution as the uncertainty relies on the edges and the background, as the blue uniform sky, is certain.

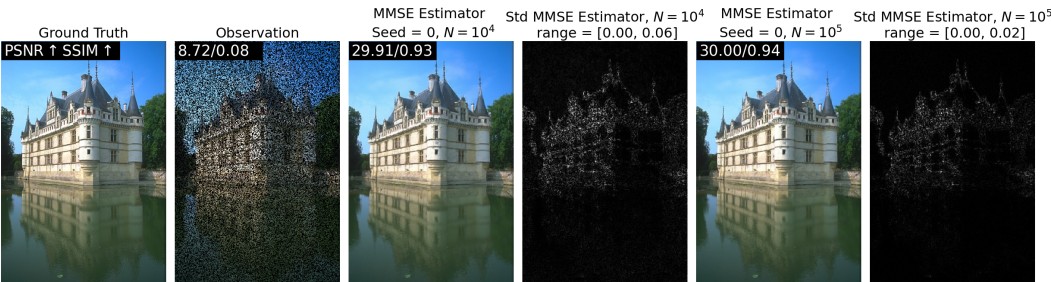

Figure 6: PSGLA with DnCNN for $50\%$ missing pixels and $\sigma = 1/255$ with two number of iterations $N \in \{10^4, 10^5\}$. We run PSGLA with 100 random seeds in the algorithm randomness for $N = 10^4$ and 10 random seeds in the algorithm randomness for $N = 10^5$ on the same observation. We can compute an estimator of the standard deviation of the MMSE estimator. This MMSE restored image is expected to be deterministic, so we expect this standard deviation to be significantly smaller than the standard deviation of the Markov Chain itself (see in Figure 5). Note that the standard deviation of the estimator is 10 times smaller than the standard deviation of PSGLA Markov Chain.

# D  On the Moreau envelope for weakly convex function

In this section, we recall and detail results on the Moreau envelope, also called Moreau-Yosida envelop, in the case of weak convexity. For clarity of this paper, we detail all the proofs. More details on the Moreau envelope of convex function can be found in [80, Part 1.G] or in [65]; and on the inf-convolution in [66].

All the following results are known by experts in the field. However, to our knowledge, there is no reference that states and proves all the results given in this appendix, especially for **weakly convex functions**. Different proofs can be found in [65, 80, 46, 3, 41, 50]. For simplicity, we choose not to introduce the Clarke sub-differential [16] which is necessary to analyse the Moreau envelope with non-differentiable functions. Therefore, some of the proofs (Lemma 11 and Lemma 13) are only given for differentiable functions, references being given for the general case.

## D.1  Definitions

We first define the notion of inf-convolution as well as the Moreau envelope.

**Definition 1.** *The inf-convolution between two functions $f, g : \mathbb{R}^d \to \mathbb{R}$ is defined for $x \in \mathbb{R}^d$ by*

$$(f \square g)(x) = \inf_{y \in \mathbb{R}^d} f(y) + g(x - y), \tag{17}$$

*with the convention $(f \square g)(x) = -\infty$ if the right-hand-side of equation (17) is not bounded from below.*

**Definition 2.** *For $\gamma > 0$, the Moreau envelope of $g : \mathbb{R}^d \to \mathbb{R}$, noted $g^\gamma$ is defined for $x \in \mathbb{R}^d$ by*

$$g^\gamma(x) = \left( g \square \frac{1}{2\gamma} \| \cdot \|^2 \right)(x) = \inf_{y \in \mathbb{R}^d} \frac{1}{2\gamma} \|x - y\|^2 + g(y). \tag{18}$$

The Moreau envelope is finite on $\mathbb{R}^d$ if $g$ is $\rho$-weakly convex, i.e. $\rho > 0$ and $g + \frac{\rho}{2} \| \cdot \|^2$ is convex, with $\gamma \rho < 1$. In fact, the function $y \mapsto \frac{1}{2\gamma} \|x - y\|^2 + g(y)$ is $\left( \frac{1}{\gamma} - \rho \right)$ strongly convex so it admits a unique minimum for $\gamma \rho < 1$. The minimal point is called the proximal point $\mathsf{Prox}_{\gamma g}(x) = \arg\min_{y \in \mathbb{R}^d} \frac{1}{2\gamma} \|x - y\|^2 + g(y)$. The Moreau envelope is an approximation of $g$ in the sense that $\forall x \in \mathbb{R}^d$, $\lim_{\gamma \to 0} g^\gamma(x) = g(x)$ (see Lemma 2) and it is a regularization as $g^\gamma$ is differentiable (see Lemma 12) even if $g$ is not.

## D.2  On the dependence of $g^\gamma$ in $\gamma > 0$

The following result shows the monotonicty of $g^\gamma$ in $\gamma$ and the fact that the Moreau envelope is an approximation of $g$ when $\gamma \to 0$.

**Lemma 2.** *If $g$ is $\rho$-weakly convex and continuous, then for $x \in \mathbb{R}^d$, $\gamma \in (0, \frac{1}{\rho}) \to g^\gamma(x)$ is decreasing and*

$$\lim_{\gamma \to 0} g^\gamma(x) = g(x). \tag{19}$$

*Moreover, $\gamma \in (0, \frac{1}{\rho}) \to g(\mathsf{Prox}_{\gamma g}(x))$ is decreasing and*

$$\lim_{\gamma \to 0} g(\mathsf{Prox}_{\gamma g}(x)) = g(x). \tag{20}$$

Lemma 2 can been generalized in the case where $g$ is lower semicontinuous instead of continuous.

*Proof.* We need $\gamma \leq \frac{1}{\rho}$ to ensure that the Moreau envelope is well defined. We obtain from the definition of the Moreau envelope (see relation (18)) that $\gamma \to g^\gamma(x)$ is decreasing and $g^\gamma(x) \leq g(x)$. The rest of the proof is a generalization of [3, Proposition 12.33] in the weakly convex case.

For $x, y \in \mathbb{R}^d$, we denote by $p = \mathsf{Prox}_{\gamma g}(x)$ and, with $\alpha \in (0, 1)$, $p_\alpha = \alpha y + (1 - \alpha)p$. By definition of the proximal operator, we have

$$g(p) + \frac{1}{2\gamma} \|x - p\|^2 \leq g(p_\alpha) + \frac{1}{2\gamma} \|x - p_\alpha\|^2.$$

By the previous inequality and the $\rho$-weak convexity of $g$, we get

$$g(p) \leq g(p_\alpha) + \frac{1}{2\gamma}\|x - p_\alpha\|^2 - \frac{1}{2\gamma}\|x - p\|^2$$

$$\leq \alpha g(y) + (1-\alpha)g(p) + \frac{\rho}{2}\alpha(1-\alpha)\|y - p\|^2 - \frac{\alpha}{\gamma}\langle x - p, y - p\rangle + \frac{\alpha^2}{2\gamma}\|y - p\|^2.$$

By rearranging the terms, we get

$$\alpha\langle x - p, y - p\rangle \leq \alpha\gamma g(y) - \alpha\gamma g(p) + \frac{\alpha}{2}(\alpha + \rho\gamma(1-\alpha))\|y - p\|^2.$$

By dividing by $\alpha$ and taking $\alpha \to 0$, we get

$$\langle x - p, y - p\rangle \leq \gamma g(y) - \gamma g(p) + \frac{\gamma\rho}{2}\|y - p\|^2.$$

For $\frac{1}{\rho} > \mu \geq \gamma$, we define $q = \mathsf{Prox}_{\mu g}(x)$. By the same reasoning, we have that $\forall y \in \mathbb{R}^d$:

$$\langle x - q, y - q\rangle \leq \gamma g(y) - \gamma g(q) + \frac{\mu\rho}{2}\|y - q\|^2.$$

Taking respectively $y = q$ and $y = p$ in the two previous inequalities, we obtain

$$\langle q - p, x - p\rangle \leq \gamma g(q) - \gamma g(p) + \frac{\gamma\rho}{2}\|q - p\|^2$$

$$\langle p - q, x - q\rangle \leq \mu g(p) - \mu g(q) + \frac{\mu\rho}{2}\|p - q\|^2.$$

By summing these two relations and rearranging the terms, we get

$$\left(1 - \frac{\rho}{2}(\mu - \gamma)\right)\|p - q\|^2 \leq (\mu - \gamma)(g(p) - g(q)).$$

Therefore, if $\gamma, \mu \in (0, \frac{1}{\rho})$, for $\mu \geq \gamma$, we get $g(\mathsf{Prox}_{\mu g}(x)) \leq g(\mathsf{Prox}_{\gamma g}(x))$. It proves that $\gamma \in (0, \frac{1}{\rho}) \to g(\mathsf{Prox}_{\gamma g}(x))$ is decreasing.

Now, we prove that $g^\gamma(x) \to g(x)$ when $\gamma \to 0$. We denote $M = \sup_{\gamma \in (0, \frac{1}{\rho})} g^\gamma(x)$, we have $\forall \gamma \in (0, \frac{1}{\rho})$,

$$M \geq g^\gamma(x) = g(\mathsf{Prox}_{\gamma g}(x)) + \frac{1}{2\gamma}\|x - \mathsf{Prox}_{\gamma g}(x)\|^2.$$

Since, $y \to g(y) + \rho\|x - y\|^2$ is $\rho$-strongly convex thus coercive and given that $\mathsf{Prox}_{\gamma g}(x)$ has a value lower than $M$, we get that $m = \sup_{\gamma \in (0, \frac{1}{2\rho})} \mathsf{Prox}_{\gamma g}(x) < +\infty$. Moreover, the function $g + \frac{\rho}{2}\|\cdot\|^2$ is convex and lower bounded by an affine function. So there exist $a_1 \in \mathbb{R}^d$ and $a_2 \in \mathbb{R}$, such that $\forall x \in \mathbb{R}^d$, $g(x) + \frac{\rho}{2}\|x\|^2 \geq \langle a_1, x\rangle + a_2$. Combining the previous facts, we get

$$M \geq g(\mathsf{Prox}_{\gamma g}(x)) + \frac{1}{2\gamma}\|x - \mathsf{Prox}_{\gamma g}(x)\|^2$$

$$\geq \langle a_1, \mathsf{Prox}_{\gamma g}(x)\rangle + a_2 - \frac{\rho}{2}\|\mathsf{Prox}_{\gamma g}(x)\|^2 + \frac{1}{2\gamma}\|x - \mathsf{Prox}_{\gamma g}(x)\|^2$$

$$\geq -\|a_1\|m + a_2 - \frac{\rho}{2}m^2 + \frac{1}{2\gamma}\|x - \mathsf{Prox}_{\gamma g}(x)\|^2.$$

So we get $\lim_{\gamma \to 0}\|x - \mathsf{Prox}_{\gamma g}(x)\| = 0$. Finally, by the continuity of $g$, we have

$$g(x) \geq \lim_{\gamma \to 0} g^\gamma(x) \geq \lim_{\gamma \to 0} g(\mathsf{Prox}_{\gamma g}(x)) = g(x),$$

which proves the desired result. $\qquad\square$

## D.3 On the convexity and weak-convexity of the Moreau envelope

In this part, we prove that the Moreau envelope preserves the convexity and weak-convexity properties.

We will first need the general result of Lemma 3.

**Lemma 3.** *If the function $(x, y) \in A \times B \to f(x, y)$, with $A, B$ convex sets, is convex on the variable $(x, y) \in A \times B$, then the function $x \in A \mapsto \inf_{y \in B} f(x, y)$ is convex.*

*Proof.* By the convexity of $f$, we get that $\forall t \in [0, 1]$, $x, x' \in A$, $y, y' \in B$,
$$f(tx + (1 - t)x', ty + (1 - t)y') \leq tf(x, y) + (1 - t)f(x', y').$$
By taking the infimum on $y, y' \in B$, we get
$$\inf_{y, y' \in B} f(tx + (1 - t)x', ty + (1 - t)y') \leq t \inf_{y \in B} f(x, y) + (1 - t) \inf_{y' \in B} f(x', y').$$
Because $B$ is convex, $\{ty + (1-t)y' | y, y' \in B\} \subset B$, so $\inf_{y,y' \in B} f(tx+(1-t)x', ty+(1-t)y') \geq \inf_{y \in B} f(tx + (1 - t)x', y)$ and we get the convexity of $x \in A \mapsto \inf_{y \in B} f(x, y)$. $\qquad \square$

**Lemma 4.** *If $g$ is convex, then for all $\gamma > 0$, $g^\gamma$ is convex.*

*Proof.* We denote $v(x, y) = \frac{1}{2\gamma}\|x - y\|^2 + g(y)$. For $x, x', y, y' \in \mathbb{R}^d$, we have
$$v(\lambda x + (1 - \lambda)x', \lambda y + (1 - \lambda)y')$$
$$= \frac{1}{2\gamma}\|\lambda(x - y) + (1 - \lambda)(x' - y')\|^2 + g(\lambda y + (1 - \lambda)y').$$
From the convexity of $\| \cdot \|^2$ and $g$, we get that $v$ is convex in $(x, y) \in \mathbb{R}^d \times \mathbb{R}^d$. Then by applying Lemma 3, we get that $g^\gamma$ is convex. $\qquad \square$

In order to study the weak convexity of the Moreau envelope, we will need the following technical lemma.

**Lemma 5.** *If $g$ is $\rho$-weakly convex, and denoting $g_\rho = g + \frac{\rho}{2}\| \cdot \|^2$ which is convex, we have*
$$g^\gamma(x) = g_\rho^{\frac{\gamma}{1 - \gamma\rho}}\left(\frac{x}{1 - \rho\gamma}\right) - \frac{\rho}{2(1 - \rho\gamma)}\|x\|^2,$$
*and*
$$\mathsf{Prox}_{\gamma g}(x) = \mathsf{Prox}_{\frac{\gamma}{1 - \gamma\rho}g_\rho}\left(\frac{x}{1 - \gamma\rho}\right).$$

*Proof.* Let us denote as $g_\rho$ the convex function $g_\rho = g + \frac{\rho}{2}\|\cdot\|^2$. One has $g^\gamma = (g_\rho - \frac{\rho}{2}\|\cdot\|^2)\square\frac{1}{2\gamma}\|\cdot\|^2$, so
$$g^\gamma(x) = \inf_y g_\rho(y) - \frac{\rho}{2}\|y\|^2 + \frac{1}{2\gamma}\|x - y\|^2 \tag{21}$$
$$= \inf_y \left(g_\rho(y) + \frac{1 - \rho\gamma}{2\gamma}\left(\|y\|^2 - \frac{2}{1 - \rho\gamma}\langle x, y \rangle + \frac{1}{(1 - \rho\gamma)^2}\|x\|^2\right)\right)$$
$$+ \frac{1}{2\gamma}\left(1 - \frac{1}{(1 - \rho\gamma)}\right)\|x\|^2$$
$$= \inf_y \left(g_\rho(y) + \frac{1 - \rho\gamma}{2\gamma}\left\|y - \frac{1}{1 - \rho\gamma}x\right\|^2\right) - \frac{\rho}{2(1 - \rho\gamma)}\|x\|^2 \tag{22}$$
$$= g_\rho^{\frac{\gamma}{1 - \gamma\rho}}\left(\frac{x}{1 - \rho\gamma}\right) - \frac{\rho}{2(1 - \rho\gamma)}\|x\|^2.$$
Then, by the definition of the proximal operator as the minimum point $y \in \mathbb{R}^d$ in the previous optimizations problems (21) and (22), we get
$$\mathsf{Prox}_{\gamma g}(x) = \mathsf{Prox}_{\frac{\gamma}{1 - \gamma\rho}g_\rho}\left(\frac{x}{1 - \gamma\rho}\right).$$
$\qquad \square$

**Lemma 6.** *If $g$ is $\rho$-weakly convex, with $\gamma\rho < 1$, then $g^\gamma$ is $\frac{\rho}{1-\gamma\rho}$-weakly convex.*

Lemma 6 generalizes Lemma 4, for $\rho > 0$. Note that the Moreau envelope of $g$ is less weakly convex that $g$ because $\frac{\rho}{1-\gamma\rho} > \rho$.

*Proof.* We give two proofs of Lemma 6. The first one is more technical and requires $g$ to be twice differentiable, based on an Hessian computation. This proof will be useful for future analysis. The second proof is more elegant and straightforward.

**Proof in the case where $g$ is twice differentiable** We introduce $v(x,y) = \frac{1}{2\gamma}\|x-y\|^2 + g(y)$. By definition, $g^\gamma(x) = \inf_{y\in\mathbb{R}^d} v(x,y)$.

For $\lambda > 0$, we study the Hessian of the function $x, y \in \mathbb{R}^d \to v(x,y) + \frac{\lambda}{2}\|x\|^2$ that writes

$$M(x,y) = \begin{pmatrix} \left(\frac{1}{\gamma} + \lambda\right) I_d & -\frac{1}{\gamma} I_d \\ -\frac{1}{\gamma} I_d & \frac{1}{\gamma} I_d + \nabla^2 g(y) \end{pmatrix}.$$

For $z, t \in \mathbb{R}^d$, by the $\rho$-weakly convexity of $g$, we have

$$\begin{pmatrix} z^T & t^T \end{pmatrix} M(x,y) \begin{pmatrix} z \\ t \end{pmatrix} = \left(\frac{1}{\gamma} + \lambda\right)\|z\|^2 - \frac{2}{\gamma} z^T t + \frac{1}{\gamma}\|t\|^2 + t^T \nabla^2 g(y) t$$

$$\geq \left(\frac{1}{\gamma} + \lambda\right)\|z\|^2 - \frac{2}{\gamma} z^T t + \left(\frac{1}{\gamma} - \rho\right)\|t\|^2.$$

Moreover, $\forall \alpha > 0$, $2z^T t \leq 2\|z\|\|t\| \leq \frac{1}{\alpha}\|z\|^2 + \alpha\|t\|^2$. By applying this inequality with $\alpha = 1 - \rho\gamma > 0$, we get

$$\begin{pmatrix} z^T & t^T \end{pmatrix} M(x,y) \begin{pmatrix} z \\ t \end{pmatrix}$$

$$\geq \left(\frac{1}{\gamma} + \lambda\right)\|z\|^2 - \frac{1}{(1-\rho\gamma)\gamma}\|z\|^2 - \frac{(1-\rho\gamma)}{\gamma}\|t\|^2 + \left(\frac{1}{\gamma} - \rho\right)\|t\|^2$$

$$\geq \left(\lambda - \frac{\rho}{1-\rho\gamma}\right)\|z\|^2.$$

So, we get that for $\lambda = \frac{\rho}{1-\rho\gamma}$, the matrix $M(x,y)$ is positive. Then the function $(x,y) \in \mathbb{R}^d \times \mathbb{R}^d \mapsto v(x,y) + \frac{\rho}{2(1-\rho\gamma)}\|x\|^2$ is convex in $(x,y) \in \mathbb{R}^d \times \mathbb{R}^d$. From Lemma 3, we get that $g^\gamma$ is $\frac{\rho}{1-\rho\gamma}$-weakly convex.

**General proof** Let us denote as $g_\rho$ the convex function $g_\rho = g + \frac{\rho}{2}\|.\|^2$. By Lemma 5, we get

$$g^\gamma(x) = g_\rho^{\frac{\gamma}{1-\gamma\rho}}\left(\frac{x}{1-\rho\gamma}\right) - \frac{\rho}{2(1-\rho\gamma)}\|x\|^2$$

Since $g_\rho$ is convex, so does $g_\rho^{\frac{\gamma}{1-\gamma\rho}}\left(\frac{x}{1-\rho\gamma}\right)$ for $\gamma\rho < 1$. Hence we get that $g^\gamma + \frac{\rho}{2(1-\gamma\rho)}\|\cdot\|^2$ is convex, so that $g^\gamma$ is $\frac{\rho}{1-\gamma\rho}$-weakly convex for $\gamma\rho < 1$. $\square$

### D.4 The intrinsic link between the Moreau envelope and the convex conjugate

In this part, we introduce the convex conjugate and shows its link with the Moreau envelope. In particular, Lemma 9 shows a duality result between the convex conjugate and the Proximal operator.

**Definition 3.** *The convex conjugate of a proper function $f : \mathbb{R}^d \to \mathbb{R}$, denoted by $f^\star$, is defined for $x \in \mathbb{R}^d$ by*

$$f^\star(x) = \sup_{y\in\mathbb{R}^d} \langle x, y\rangle - f(y).$$

The notion of convex conjugate of $f$ is related with the inf-convolution and the Moreau envelope, as shown by the following Lemmas 7 to 9.

**Lemma 7.** *For $f : \mathbb{R}^d \to \mathbb{R}$, a proper function, we have the following properties*

   *(i) $f^\star$ is convex.*

   *(ii) For $f, g : \mathbb{R}^d \to \mathbb{R}$ convex, then $(f \Box g)^\star = f^\star + g^\star$.*

   *(iii) $f : \mathbb{R}^d \to \mathbb{R}^d$ is convex if and only if $f^{\star\star} = f$.*

   *(iv) $\left(\frac{\alpha}{2}\| \cdot \|^2\right)^\star = \frac{1}{2\alpha}\| \cdot \|^2$.*

Note that, by Lemma 7(i), $f^\star$ is convex even if $f$ is non-convex. Moreover, Lemma 7(ii) shows the compatibility of the convex conjugate with the inf-convolution operation.

*Proof.* **(i)** For $x, y \in \mathbb{R}^d$ and $\lambda \in [0, 1]$, we have

$$
\begin{aligned}
f^\star(\lambda x + (1 - \lambda)y) &= \sup_{z \in \mathbb{R}^d} \langle \lambda x + (1 - \lambda)y, z \rangle - f(z) \\
&= \sup_{z \in \mathbb{R}^d} \lambda \left( \langle x, z \rangle - f(z) \right) + (1 - \lambda) \left( \langle y, z \rangle - f(z) \right) \\
&\leq \lambda \sup_{z \in \mathbb{R}^d} \left( \langle x, z \rangle - f(z) \right) + (1 - \lambda) \sup_{z \in \mathbb{R}^d} \left( \langle y, z \rangle - f(z) \right) \\
&\leq \lambda f^\star(x) + (1 - \lambda)f^\star(y),
\end{aligned}
$$

which shows the convexity of $f^\star$.

**(ii)** For $x \in \mathbb{R}^d$,

$$
\begin{aligned}
(f \Box g)^\star(x) &= \sup_{y \in \mathbb{R}^d} \langle x, y \rangle - (f \Box g)(y) \\
&= \sup_{y \in \mathbb{R}^d} \langle x, y \rangle - \inf_{z \in \mathbb{R}^d} f(z) + g(y - z) \\
&= \sup_{y \in \mathbb{R}^d} \sup_{z \in \mathbb{R}^d} \langle x, y \rangle - f(z) - g(y - z) \\
&= \sup_{z \in \mathbb{R}^d} -f(z) + \sup_{y \in \mathbb{R}^d} \langle x, y \rangle - g(y - z) \\
&= \sup_{z \in \mathbb{R}^d} -f(z) + \langle x, z \rangle + \sup_{y \in \mathbb{R}^d} \langle x, y - z \rangle - g(y - z) \\
&= \sup_{z \in \mathbb{R}^d} -f(z) + \langle x, z \rangle + g^\star(x) \\
&= f^\star(x) + g^\star(x).
\end{aligned}
$$

**(iii)** If $f^{\star\star} = f$, then by point (i), $f$ is convex.

If $f$ is convex, we have

$$
f(x) = \sup_{(a,b) \in \Sigma} \langle a, x \rangle + b,
$$

with $\Sigma = \{(a, b) \in \mathbb{R}^d \times \mathbb{R} \mid \forall x \in \mathbb{R}^d, \langle a, x \rangle + b \leq f(x)\}$. $(a, b) \in \Sigma$, if and only if $\forall x \in \mathbb{R}^d$, $\langle a, x \rangle - f(x) \leq -b$, i.e. $-b \geq f^\star(a)$. Then, we have

$$
\begin{aligned}
f(x) &= \sup_{(a,b) \in \Sigma} \langle a, x \rangle + b = \sup_{a \in \mathbb{R}^d, b \leq -f^\star(a)} \langle a, x \rangle + b \\
&= \sup_{a \in \mathbb{R}^d} \langle a, x \rangle - f^\star(a) = f^{\star\star}(x).
\end{aligned}
$$

**(iv)** For $x \in \mathbb{R}^d$, $\left(\frac{\alpha}{2}\|x\|^2\right)^\star = \sup_{y \in \mathbb{R}^d} \langle x, y \rangle - \frac{\alpha}{2}\|y\|^2$. The sup is reached at $y = \frac{x}{\alpha}$, so that $\left(\frac{\alpha}{2}\| \cdot \|^2\right)^\star = \frac{1}{2\alpha}\| \cdot \|^2$.

$\square$

**Lemma 8.** *For $f : \mathbb{R}^d \to \mathbb{R}$ a proper function, we have the following properties*

   *(i) $\forall x, y \in \mathbb{R}^d, f(x) + f^\star(y) \geq \langle x, y \rangle$.*

(ii) *For $f$ convex and differentiable and $x \in \mathbb{R}^d$, with $y = \nabla f(x)$, we have $f(x) + f^\star(y) = \langle x, y \rangle$.*

(iii) *For $f$ convex and differentiable, we have $y = \nabla f(x)$ if and only if $x = \nabla f^\star(y)$.*

(iv) *For $f$ convex and differentiable, $f$ is $\mu$-strongly convex if and only if $f^\star$ is $1/\mu$-smooth.*

Lemma 8(iii) can be reformulated as $\nabla f \circ \nabla f^\star = I_d$.

*Proof.* **(i)** For $x, y \in \mathbb{R}^d$

$$f(x) + f^\star(y) = \sup_{z \in \mathbb{R}^d} \langle y, z \rangle + f(x) - f(z) \geq \langle y, x \rangle.$$

**(ii)** If $y = \nabla f(x)$, then by the convexity of $f$, we get, $\forall z \in \mathbb{R}^d$,

$$f(z) \geq f(x) + \langle y, z - x \rangle$$
$$\langle x, y \rangle - f(x) \geq \langle y, z \rangle - f(z)$$
$$\langle x, y \rangle - f(x) \geq f^\star(y).$$

So, by definition of $f^\star$, we get $f^\star(y) = \langle x, y \rangle - f(x)$ .

**(iii)** If $y = \nabla f(x)$, by the point (ii), we have $f(x) + f^\star(y) = \langle x, y \rangle$. Then for all $t \in \mathbb{R}^d$

$$f^\star(t) = \sup_{s \in \mathbb{R}^d} \langle t, s \rangle - f(s)$$
$$\geq \langle t, x \rangle - f(x)$$
$$\geq \langle t - y, x \rangle + \langle y, x \rangle - f(x)$$
$$\geq \langle t - y, x \rangle + f^\star(y).$$

Thanks to Lemma 7(i) $f^\star$ is convex. Hence we get that $x = \nabla f^\star(y)$. The reverse is true because $f$ is convex so $f^{\star\star} = f$ by Lemma 7(iii).

**(iv)** We denote $u(x, y) = \langle x, y \rangle - f(y)$. The $\mu$-strong convexity of $f$ implies that $\forall x, y \in \mathbb{R}^d$,

$$\langle \nabla f(x) - \nabla f(y), x - y \rangle \geq \mu \|x - y\|^2. \tag{23}$$

Thus $\nabla f : \mathbb{R}^d \to \mathbb{R}^d$ is injective. As $u$ is strongly concave w.r.t. $y$, it admits a unique maximal point denoted by $y_0$. On this point the optimal condition gives $\nabla_y u(x, y_0) = 0$, so $\nabla f(y_0) = x$. Therefore, $\nabla f$ is surjective, so it is a bijective function and

$$f^\star(x) = \langle x, (\nabla f)^{-1}(x) \rangle - f((\nabla f)^{-1}(x)). \tag{24}$$

By applying Equation (23) with $(\nabla f)^{-1}$, we get $\forall x, y \in \mathbb{R}^d$,

$$\langle x - y, (\nabla f)^{-1}(x) - (\nabla f)^{-1}(y) \rangle \geq \mu \|(\nabla f)^{-1}(x) - (\nabla f)^{-1}(y)\|^2.$$

The Cauchy-Schwarz inequality then gives

$$\|(\nabla f)^{-1}(x) - (\nabla f)^{-1}(y)\| \leq \frac{1}{\mu} \|x - y\|. \tag{25}$$

Using (24), we have for $x, h \in \mathbb{R}^d$

$$f^\star(x + h) - f^\star(x)$$
$$= \langle x + h, (\nabla f)^{-1}(x + h) \rangle - f((\nabla f)^{-1}(x + h)) - \langle x, (\nabla f)^{-1}(x) \rangle$$
$$+ f((\nabla f)^{-1}(x))$$
$$= \langle x, (\nabla f)^{-1}(x + h) - (\nabla f)^{-1}(x) \rangle + \langle h, (\nabla f)^{-1}(x + h) \rangle$$
$$- f((\nabla f)^{-1}(x + h)) + f((\nabla f)^{-1}(x)).$$

By denoting $u = (\nabla f)^{-1}(x + h) - (\nabla f)^{-1}(x)$, we know that $u = \mathcal{O}(\|h\|)$ thanks to Equation (25). Thus, we get

$$f^\star(x + h) - f^\star(x)$$
$$= \langle x, u \rangle + \langle h, (\nabla f)^{-1}(x) \rangle + \langle h, u \rangle - f((\nabla f)^{-1}(x) + u) + f((\nabla f)^{-1}(x))$$
$$= \langle x, u \rangle + \langle h, (\nabla f)^{-1}(x) \rangle + \langle h, u \rangle - \langle \nabla f((\nabla f)^{-1}(x)), u \rangle + \mathcal{O}(\|h\|^2)$$
$$= \langle h, (\nabla f)^{-1}(x) \rangle + \mathcal{O}(\|h\|^2).$$

Therefore $f^\star$ is differentiable and $\nabla f^\star = (\nabla f)^{-1}$. Combining this result with equation (25) proves that $f^\star$ is $\frac{1}{\mu}$-smooth.

Conversely, if $f^\star$ is $\frac{1}{\mu}$-smooth, then $\forall x, y \in \mathbb{R}^d$,

$$\langle \nabla f^\star(x) - \nabla f^\star(y), x - y \rangle \geq \mu \|\nabla f^\star(x) - \nabla f^\star(y)\|^2. \tag{26}$$

By denoting $z = \nabla f^\star(x)$ and $t = \nabla f^\star(y)$, by Lemma 8(iii), we have $x = \nabla f(z)$ and $y = \nabla f(t)$, so

$$\langle z - t, \nabla f(z) - \nabla f(t) \rangle \geq \mu \|z - t\|^2,$$

which implies that $f$ is $\mu$-strongly convex.

$\square$

**Lemma 9** ([65])**.** *For $f$ convex and $x, y, z \in \mathbb{R}^d$, the two following statements are equivalent*

*(i)* $z = x + y$ *and* $f(x) + f^\star(y) = \langle x, y \rangle$.

*(ii)* $x = \mathsf{Prox}_f(z)$ *and* $y = \mathsf{Prox}_{f^\star}(z)$.

Lemma 9 shows the intrinsic link between the proximal operator and the convex conjugate. In particular, it shows that the inequality of Lemma 8(i) is attained only for couple that are of the form $(\mathsf{Prox}_f(z), \mathsf{Prox}_{f^\star}(z))$. Moreover, it shows also the equality $\mathsf{Prox}_f(z) + \mathsf{Prox}_{f^\star}(z) = z$.

*Proof.* **(i)** $\implies$ **(ii)** For $u \in \mathbb{R}^d$, by the convex conjugate definition, we have

$$f^\star(y) \geq \langle u, y \rangle - f(u).$$

By using that $f(x) + f^\star(y) = \langle x, y \rangle$, we get

$$\langle x, y \rangle - f(x) \geq \langle u, y \rangle - f(u).$$

Then, using the previous inequality and $z = x + y$,

$$\begin{aligned}
\frac{1}{2}\|x - z\|^2 + f(x) &= \frac{1}{2}\|x\|^2 + \frac{1}{2}\|z\|^2 - \langle x, z \rangle + f(x) \\
&\leq \frac{1}{2}\|x\|^2 + \frac{1}{2}\|z\|^2 - \langle x, z \rangle + \langle x, y \rangle + f(u) - \langle u, y \rangle \\
&\leq \frac{1}{2}\|x\|^2 + \frac{1}{2}\|z\|^2 - \|x\|^2 + f(u) - \langle u, z \rangle + \langle u, x \rangle \\
&\leq \frac{1}{2}\|u - z\|^2 + f(u) - \frac{1}{2}\|x - u\|^2.
\end{aligned}$$

Then, necessarily $x$ is the only minimum of the functional $u \mapsto \frac{1}{2}\|u - z\|^2 + f(u)$, which means that $x = \mathsf{Prox}_f(z)$. A similar computation gives that $y = \mathsf{Prox}_{f^\star}(z)$.

**(ii)** $\implies$ **(i)** We note $y' = z - x$, with $x = \mathsf{Prox}_f(z)$. By the convexity of $f$, for $t \in (0, 1)$ and $u \in \mathbb{R}^d$, we have

$$f(tu + (1 - t)x) \leq tf(u) + (1 - t)f(x).$$

Then, because $x = \mathsf{Prox}_f(z)$, we have $\forall u \in \mathbb{R}^d$ and $\forall t \in (0, 1)$,

$$\frac{1}{2}\|x - z\|^2 + f(x) \leq \frac{1}{2}\|tu + (1 - t)x - z\|^2 + f(tu + (1 - t)x).$$

By combining the two previous inequalities, we get

$$\frac{1}{2}\|x - z\|^2 + f(x) \leq \frac{1}{2}\|tu + (1 - t)x - z\|^2 + tf(u) + (1 - t)f(x)$$

$$-\frac{t^2}{2}\|u - x\|^2 + t\langle u - x, y' \rangle \leq tf(u) - tf(x).$$

By dividing by $t$ and letting $t \to 0$, we obtain that for all $u \in \mathbb{R}^d$

$$\langle u - x, y' \rangle \leq f(u) - f(x)$$

$$\langle u, y' \rangle - f(u) \leq \langle x, y' \rangle - f(x).$$

So

$$f^\star(y') = \langle x, y' \rangle - f(x),$$

which ends the proof.

$\square$

## D.5 Properties of the proximal operator

Based on the previous link between the convex conjugate and the Moreau envelope, we can now deduce some properties on the proximal operator (Corollary (2) and Lemma 11) of weakly convex functions. Moreover, we will study how the Moreau envelope preserves strong convexity (Lemma 10).

**Corollary 2.** *For $g$ $\rho$-weakly convex with $\gamma\rho < 1$ and differentiable at $x \in \mathbb{R}^d$, we have*

$$\mathsf{Prox}_{\gamma g}\left(x + \gamma\nabla g(x)\right) = x$$

Corollary 2 leads to the notation $\mathsf{Prox}_{\gamma g} = (I_d + \gamma\nabla g)^{-1}$.

*Proof.* We first demonstrate the relation for convex functions $g$ and then use it for weakly convex ones.

- For $g$ convex and $x \in \mathbb{R}^d$, by Lemma 8(ii), we have for $y = \nabla g(x)$, $g(x) + g^\star(y) = \langle x, y\rangle$. Then by Lemma 9, we get $x = \mathsf{Prox}_g\left(x + \nabla g(x)\right)$. Then, if $g$ is convex, $\gamma g$ is convex and we get $\forall \gamma > 0$,
$$\mathsf{Prox}_{\gamma g}\left(x + \gamma\nabla g(x)\right) = x. \tag{27}$$

- If $g$ is $\rho$-weakly convex with $\gamma\rho < 1$, then we introduce the convex function $g_\rho = g + \frac{\rho}{2}\|\cdot\|^2$. By Lemma 5, we have

$$\mathsf{Prox}_{\gamma g}\left(x\right) = \mathsf{Prox}_{\frac{\gamma}{1-\gamma\rho}g_\rho}\left(\frac{x}{1-\gamma\rho}\right).$$

Then

$$\begin{aligned}
\mathsf{Prox}_{\gamma g}\left(x + \gamma\nabla g(x)\right) &= \mathsf{Prox}_{\frac{\gamma}{1-\gamma\rho}g_\rho}\left(\frac{x + \gamma\nabla g(x)}{1-\gamma\rho}\right)\\
&= \mathsf{Prox}_{\frac{\gamma}{1-\gamma\rho}g_\rho}\left(\frac{x + \gamma\nabla g_\rho(x) - \gamma\rho x}{1-\gamma\rho}\right)\\
&= \mathsf{Prox}_{\frac{\gamma}{1-\gamma\rho}g_\rho}\left(x + \frac{\gamma}{1-\gamma\rho}\nabla g_\rho(x)\right)\\
&= x.
\end{aligned}$$

The last equality is obtained by applying (27) on the convex function $g_\rho$.

$\square$

**Lemma 10.** *For a function $g$ $\rho$-strongly convex, then $g^\gamma$ is $\frac{\rho}{1+\gamma\rho}$ strongly convex.*

Note that $g^\gamma$ is less strongly convex than $g$ as $\frac{\rho}{1+\gamma\rho} < \rho$.

*Proof.* Assume that $g$ is $\rho$-strongly convex, then, thanks to Lemma 7(ii) and Lemma 7(iv), $(g^\gamma)^* = (g\square\frac{1}{2\gamma}\|x\|^2)^* = g^* + \frac{\gamma}{2}\|x\|^2$ that is convex and $\frac{1}{\rho} + \gamma = \frac{1+\rho\gamma}{\rho}$-smooth thanks to Lemma 8(iv). Combining Lemma 8(iv) and Lemma 8(iii), we deduce that $(g^\gamma)^{\star\star} = g^\gamma$ is $\frac{\rho}{1+\rho\gamma}$ strongly convex.
$\square$

**Lemma 11.** *For $g$ $\rho$-weakly convex with $\gamma\rho < 1$, $\mathsf{Prox}_{\gamma g}$ is $\frac{1}{1-\gamma\rho}$ Lipschitz.*

Note that in particular, if $g$ is convex, then $\rho = 0$ and $\forall\gamma > 0$, $\mathsf{Prox}_{\gamma g}$ is 1-Lipschitz. The fact that the proximal operator is Lipschitz is important for guaranteeing its stability.

*Proof.* The general proof of this Lemma 11 can be found in [41, Proposition 2] where the authors used [65, Proposition 5.b]. We provide here a more direct proof in the case of a differentiable function $g$.

As $g$ is $\rho$-weakly convex and differentiable, then $\forall x, y \in \mathbb{R}^d$, we have by the convexity of $g + \frac{\rho}{2}\|\cdot\|^2$

$$g(x) + \frac{\rho}{2}\|x\|^2 + \langle \nabla g(x) + \rho x, y - x \rangle \leq g(y) + \frac{\rho}{2}\|y\|^2$$

$$\langle \nabla g(x), y - x \rangle \leq g(y) - g(x) + \frac{\rho}{2}\|x - y\|^2.$$

For $x, x' \in \mathbb{R}^d$, we denote $z = \mathsf{Prox}_{\gamma g}(x)$ and $z' = \mathsf{Prox}_{\gamma g}(x')$, we get

$$\langle \nabla g(z), z' - z \rangle \leq g(z') - g(z) + \frac{\rho}{2}\|z - z'\|^2$$

$$\langle \nabla g(z'), z - z' \rangle \leq g(z) - g(z') + \frac{\rho}{2}\|z - z'\|^2.$$

By the optimal condition of the proximal operator, we also have

$$\frac{1}{\gamma}(z - x) + \nabla g(z) = 0$$

$$\frac{1}{\gamma}(z' - x') + \nabla g(z') = 0,$$

Combining the two previous sets of equations, we obtain

$$\frac{1}{\gamma}\langle x - z, z' - z \rangle \leq g(z') - g(z) + \frac{\rho}{2}\|z - z'\|^2$$

$$\frac{1}{\gamma}\langle x' - z', z - z' \rangle \leq g(z) - g(z') + \frac{\rho}{2}\|z - z'\|^2.$$

By summing the two previous inequalities and using the Cauchy-Schwarz inequality, we get

$$-\rho\|z - z'\|^2 \leq \frac{1}{\gamma}\langle z - z' - x + x', z' - z \rangle$$

$$(1 - \gamma\rho)\|z - z'\|^2 \leq \langle x' - x, z' - z \rangle$$

$$(1 - \gamma\rho)\|z - z'\|^2 \leq \|x - x'\|\|z - z'\|.$$

Therefore, we get that, $\forall x, x' \in \mathbb{R}^d$

$$\|\mathsf{Prox}_{\gamma g}(x) - \mathsf{Prox}_{\gamma g}(x')\| \leq \frac{1}{1 - \gamma\rho}\|x - x'\|,$$

which proves that $\mathsf{Prox}_{\gamma g}$ is $\frac{1}{1-\gamma\rho}$-Lipschitz. $\qquad\square$

### D.6 Proximal operator as a gradient step

Now, we can prove the main results of this appendix, that allows to interpret the proximal operator as a gradient descent step on the Moreau envelope. Note that Lemma 12 also proves that the Moreau envelope is differentiable even if $g$ is not.

**Lemma 12.** *For a function $g$ $\rho$-weakly convex, for $\gamma\rho < 1$, we have*

$$\nabla g^\gamma(x) = \frac{1}{\gamma}(x - \mathsf{Prox}_{\gamma g}(x))$$

*Proof.* Our proof is based on the strategy detailed in [80, Theorem 2.26] for $g$ convex, that we generalize for $g$ $\rho$-weakly convex. This Lemma is formulated in the weakly-convex setting in [6, Lemma 2.2] and in [21, Lemma 2.2] without proofs.

For $x \in \mathbb{R}^d$, we introduce $v = \mathsf{Prox}_{\gamma g}(x)$, $w = \frac{1}{\gamma}(x - v)$ and $h(u) = g^\gamma(x + u) - g^\gamma(x) - \langle w, u \rangle$. We aim to prove that $g^\gamma$ is differentiable at $x$ with $\nabla g^\gamma = w$, which is equivalent to $h$ differentiable at 0 and $\nabla h(0) = 0$.

By definition of the Moreau envelope, since $v = \mathsf{Prox}_{\gamma g}(x)$, we have $g^\gamma(x) = \frac{1}{2\gamma}\|x - v\|^2 + g(v)$ and $g^\gamma(x + u) \leq \frac{1}{2\gamma}\|x + u - v\|^2 + g(v)$. So, we get

$$
\begin{aligned}
h(u) &\leq \frac{1}{2\gamma}\|x + u - v\|^2 - \frac{1}{2\gamma}\|x - v\|^2 - \langle w, u\rangle \\
&\leq \frac{1}{2\gamma}\|u\|^2 + \frac{1}{\gamma}\langle x - v, u\rangle - \langle w, u\rangle \\
&\leq \frac{1}{2\gamma}\|u\|^2.
\end{aligned}
$$

Moreover, as $g^\gamma(x + u)$ is $\frac{\rho}{1-\rho\gamma}$-weakly convex by Lemma 6, then $h(u) + \frac{\rho}{2(1-\rho\gamma)}\|u\|^2 = g^\gamma(x + u) + \frac{\rho}{2(1-\rho\gamma)}\|u\|^2 - g^\gamma(x) - \langle w, u\rangle$ is convex in $u$. Therefore

$$
\frac{1}{2}\left(h(u) + \frac{\rho}{2(1 - \rho\gamma)}\|u\|^2 + h(-u) + \frac{\rho}{2(1 - \rho\gamma)}\|u\|^2\right) \geq h(0) = 0
$$

$$
h(u) \geq -\frac{\rho}{(1 - \rho\gamma)}\|u\|^2 - h(-u) \geq \left(-\frac{\rho}{1 - \rho\gamma} - \frac{1}{2\gamma}\right)\|u\|^2.
$$

So, we get

$$
\left(-\frac{\rho}{1 - \rho\gamma} - \frac{1}{2\gamma}\right)\|u\|^2 \leq h(u) \leq \frac{1}{2\gamma}\|u\|^2.
$$

Therefore $h$ is differentiable at 0 and $\nabla h(0) = 0$. This concludes the proof. $\qquad\square$

We can deduce that the Moreau envelope is smooth, i.e. $\nabla g^\gamma$ is Lipschitz.

**Lemma 13.** *If $g$ is $\rho$-weakly convex, with $\gamma\rho < 1$, then $g^\gamma$ is $\max\left(\frac{1}{\gamma}, \frac{\rho}{1-\gamma\rho}\right)$ smooth.*

*Proof.* By Lemma 6, $g^\gamma$ is $\frac{\rho}{1-\gamma\rho}$-weakly convex. Moreover, for $f$ a convex function, by [3, Proposition 23.8], $x \mapsto x - \mathsf{Prox}_{\gamma f}(x)$ is 1-Lipschitz. We prove this fact for $f$ convex *and differentiable*. For $x, y \in \mathbb{R}^d$ and $u = \mathsf{Prox}_{\gamma f}(x)$, $v = \mathsf{Prox}_{\gamma f}(y)$, we get by the optimal condition of the proximal operator

$$
\langle(x - u) - (y - v), u - v\rangle = \frac{1}{\gamma}\langle\nabla f(u) - \nabla f(v), u - v\rangle \geq 0,
$$

by convexity of $f$. Then

$$
\begin{aligned}
\|x - y\|^2 &= \|(x - u) - (y - v) + u - v\|^2 \\
&= \|(x - u) - (y - v)\|^2 + \|u - v\|^2 + 2\langle(x - u) - (y - v), u - v\rangle \\
&\geq \|(x - u) - (y - v)\|^2 + \|u - v\|^2.
\end{aligned}
$$

So $\|(x - u) - (y - v)\| \leq \|x - y\|$, which means that $x \mapsto x - \mathsf{Prox}_{\gamma f}(x)$ is 1-Lipschitz.

Combining the previous property with Lemma 12, we get that $\nabla f^\gamma$ is $\frac{1}{\gamma}$-Lipschitz. So, we have the inequality,

$$
\langle\nabla f^\gamma(x) - \nabla f^\gamma(y), x - y\rangle \leq \frac{1}{\gamma}\|x - y\|^2. \tag{28}
$$

However, Lemma 5 gives that

$$
g^\gamma = g_\rho^{\frac{\gamma}{1-\rho\gamma}}\left(\frac{x}{1 - \rho\gamma}\right) - \frac{\rho}{2(1 - \rho\gamma)}\|x\|^2.
$$

As $g_\rho$ is convex, we can apply equation (28) and obtain

$$\langle \nabla g^\gamma(x) - \nabla g^\gamma(y), x - y \rangle$$

$$= \frac{1}{1 - \rho\gamma} \langle \nabla g_\rho^{\frac{\gamma}{1-\gamma\rho}}(\frac{x}{1-\gamma\rho}) - \nabla g_\rho^{\frac{\gamma}{1-\gamma\rho}}(\frac{y}{1-\gamma\rho}), x - y \rangle - \frac{\rho}{1 - \gamma\rho} \|x - y\|^2$$

$$\leq \frac{1}{1 - \rho\gamma} \frac{1}{\gamma} \|x - y\|^2 - \frac{\rho}{1 - \gamma\rho} \|x - y\|^2$$

$$\leq \frac{1}{\gamma} \|x - y\|^2.$$

With the weak convexity of $g^\gamma$ (Lemma 6), we get

$$-\frac{\rho}{1 - \gamma\rho} \|x - y\|^2 \leq \langle \nabla g^\gamma(x) - \nabla g^\gamma(y), x - y \rangle \leq \frac{1}{\gamma} \|x - y\|^2.$$

So, we get that $\nabla g^\gamma$ is $\max\left(\frac{1}{\gamma}, \frac{\rho}{1-\gamma\rho}\right)$ smooth. $\qquad\square$

### D.7 Second derivative of the Moreau envelope and convexity of the image of the proximal operator

We can now prove that the Moreau envelope is twice differentiable if $g$ is twice differentiable and Lipschitz on the image of the proximal operator. Moreover, we will also show that the border of the the convex envelop of $\mathsf{Prox}_{\gamma g}\left(\mathbb{R}^d\right)$ is of measure zero (Proposition 2). In other words, the image of the proximity operator of a weakly convex function is almost convex.

**Lemma 14.** *Let $g$ be a $\rho$-weakly convex function. Assume that $g$ that is $\mathcal{C}^2$ and $L_g$-smooth on $\mathsf{Prox}_{\gamma g}\left(\mathbb{R}^d\right)$ with $\rho\gamma < 1$ and $L_g\gamma < 1$. Then we get, for $x \in \mathbb{R}^d$*

$$\nabla \mathsf{Prox}_{\gamma g}(x) = \left(I_d + \gamma \nabla^2 g(\mathsf{Prox}_{\gamma g}(x))\right)^{-1}$$

$$\nabla^2 g^\gamma(x) = \frac{1}{\gamma}\left(I_d - \left(I_d + \gamma \nabla^2 g(\mathsf{Prox}_{\gamma g}(x))\right)^{-1}\right).$$

*Proof.* This proof is an adaptation of [69, Theorem 3.4] for weakly convex functions.

By the optimal condition of the proximal operator, we get that

$$\frac{1}{\gamma}\left(\mathsf{Prox}_{\gamma g}(x) - x\right) + \nabla g(\mathsf{Prox}_{\gamma g}(x)) = 0.$$

Combined with Lemma 12, we get that

$$\nabla g^\gamma(x) = \nabla g(\mathsf{Prox}_{\gamma g}(x)). \tag{29}$$

Then for all $x, h \in \mathbb{R}^d$,

$$\nabla g^\gamma(x + h) - \nabla g^\gamma(x) = \nabla g(\mathsf{Prox}_{\gamma g}(x + h)) - \nabla g(\mathsf{Prox}_{\gamma g}(x))$$

$$= \nabla^2 g(\mathsf{Prox}_{\gamma g}(x))\left(\mathsf{Prox}_{\gamma g}(x + h) - \mathsf{Prox}_{\gamma g}(x)\right) + o(\|\mathsf{Prox}_{\gamma g}(x + h) - \mathsf{Prox}_{\gamma g}(x)\|).$$

However, by Lemma 11, $\mathsf{Prox}_{\gamma g}$ is $\frac{1}{1-\gamma\rho}$-Lipschitz. Therefore $o(\|\mathsf{Prox}_{\gamma g}(x + h) - \mathsf{Prox}_{\gamma g}(x)\|) = o(\|h\|)$. Then, we get

$$\nabla g^\gamma(x + h) - \nabla g^\gamma(x) = \nabla^2 g(\mathsf{Prox}_{\gamma g}(x))\left(\mathsf{Prox}_{\gamma g}(x + h) - \mathsf{Prox}_{\gamma g}(x)\right) + o(\|h\|).$$

Injecting the characterization from Lemma 12 in the left part of the previous equation, we get

$$\frac{1}{\gamma}h - \frac{1}{\gamma}\left(\mathsf{Prox}_{\gamma g}(x + h) - \mathsf{Prox}_{\gamma g}(x)\right)$$

$$= \nabla^2 g(\mathsf{Prox}_{\gamma g}(x))\left(\mathsf{Prox}_{\gamma g}(x + h) - \mathsf{Prox}_{\gamma g}(x)\right) + o(\|h\|)$$

$$\frac{1}{\gamma}h = \left(\frac{1}{\gamma}I_d + \nabla^2 g(\mathsf{Prox}_{\gamma g}(x))\right)\left(\mathsf{Prox}_{\gamma g}(x + h) - \mathsf{Prox}_{\gamma g}(x)\right) + o(\|h\|)$$

$$\mathsf{Prox}_{\gamma g}(x + h) - \mathsf{Prox}_{\gamma g}(x) = \frac{1}{\gamma}\left(\frac{1}{\gamma}I_d + \nabla^2 g(\mathsf{Prox}_{\gamma g}(x))\right)^{-1}(h) + o(\|h\|),$$

where $\frac{1}{\gamma}I_d + \nabla^2 g(\mathsf{Prox}_{\gamma g}(x))$ is invertible because $g$ is $L_g$-smooth at the point $\mathsf{Prox}_{\gamma g}(x) \in \mathsf{Prox}_{\gamma g}(\mathbb{R}^d)$ with $\gamma L_g < 1$. The last equation shows that $\mathsf{Prox}_{\gamma g}$ is differentiable and its gradient is

$$\nabla\mathsf{Prox}_{\gamma g} = \left(I_d + \gamma\nabla^2 g \circ \mathsf{Prox}_{\gamma g}\right)^{-1}.$$

Then, we differentiate Equation (29) to get

$$\begin{aligned}
\nabla^2 g^\gamma(x) &= \nabla^2 g(\mathsf{Prox}_{\gamma g}(x))\nabla\mathsf{Prox}_{\gamma g}(x) \\
&= \nabla^2 g(\mathsf{Prox}_{\gamma g}(x))\left(I_d + \gamma\nabla^2 g(\mathsf{Prox}_{\gamma g}(x))\right)^{-1} \\
&= \frac{1}{\gamma}\left(I_d - \left(I_d + \gamma\nabla^2 g(\mathsf{Prox}_{\gamma g}(x))\right)^{-1}\right),
\end{aligned}$$

which shows the second part of Lemma 14. $\qquad\square$

We can now deduce that for $\gamma$ sufficiently small the Moreau envelope is smooth (with a constant independent of $\gamma$) on all $\mathbb{R}^d$ if the function $g$ is smooth on $\mathsf{Prox}_{\gamma g}(\mathbb{R}^d)$.

**Lemma 15.** *Let $g$ be a $\rho$-weakly convex function. Assume that $g$ is $\mathcal{C}^2$ and $L_g$-smooth on $\mathsf{Prox}_{\gamma g}(\mathbb{R}^d)$ with $L_g\gamma \leq \frac{1}{2}$ and $\rho\gamma < 1$. Then $g^\gamma$ is $2L_g$-smooth on $\mathbb{R}^d$.*

*Proof.* By Lemma 14, we have

$$\nabla^2 g^\gamma(x) = \frac{1}{\gamma}\left(I_d - \left(I_d + \gamma\nabla^2 g(\mathsf{Prox}_{\gamma g}(x))\right)^{-1}\right).$$

As we assume that $-L_g I_d \preceq \nabla^2 g(\mathsf{Prox}_{\gamma g}(x)) \preceq L_g I_d$, we get

$$\frac{1}{\gamma}\left(1 - \frac{1}{1-L_g\gamma}\right)I_d \preceq \nabla^2 g^\gamma(x) \preceq \frac{1}{\gamma}\left(1 - \frac{1}{1+L_g\gamma}\right)I_d$$

$$\underbrace{-\frac{L_g}{1-L_g\gamma}}_{:=u(\gamma)}I_d \preceq \nabla^2 g^\gamma(x) \preceq \underbrace{\frac{L_g}{1+L_g\gamma}}_{:=v(\gamma)}I_d$$

Since $u'(\gamma) = -\frac{L_g^2}{(1-L_g\gamma)^2} \leq 0$, $u$ is decreasing. As $u(\frac{1}{2L_g}) = -2L_g$, we obtain that $u(\gamma) \geq -2L_g$, for all $\gamma \in [0, \frac{1}{2L_g}]$. In the same way, $v'(\gamma) = -\frac{L_g^2}{(1+L_g\gamma)^2} \leq 0$, so $v$ is decreasing. Since $v(0) = L_g$, for $\gamma \geq 0$, we have $v(\gamma) \leq L_g$. Finally, for $\gamma \in [0, \frac{1}{2L_g}]$, we get

$$-2L_g I_d \preceq \nabla^2 g^\gamma(x) \preceq L_g I_d.$$

So $g^\gamma$ is $2L_g$-smooth on $\mathbb{R}^d$. $\qquad\square$

**Proposition 2.** *Let $g$ be a $\rho$-weakly convex function with $\rho\gamma < 1$. We have*

$$Leb(dom(g) \setminus \mathsf{Prox}_{\gamma g}(\mathbb{R}^d)) = 0, \tag{30}$$

*with Leb the Lebesgue measure, $dom(g) = \{x \in \mathbb{R}^d \,|\, g(x) < +\infty\}$. Moreover, we have that*

$$Leb(Conv(\mathsf{Prox}_{\gamma g}(\mathbb{R}^d)) \setminus \mathsf{Prox}_{\gamma g}(\mathbb{R}^d)) = 0, \tag{31}$$

*where $Conv(\mathsf{Prox}_{\gamma g}(\mathbb{R}^d))$ is the convex envelop of $\mathsf{Prox}_{\gamma g}(\mathbb{R}^d)$.*

*Proof.* We denote by $dom(g)$ the set where $g < +\infty$. By the weak convexity of $g$, we know that $dom(g)$ is convex. By [57, Theorem 1], because $dom(g)$ is convex, we have that $Leb(dom(g) \setminus int(dom(g))) = 0$.

By [45, Theorem 4.2.3], a convex function is differentiable almost everywhere on the interior of its domain. As $g + \frac{\rho}{2}\|\cdot\|^2$ is convex, it is thus differentiable almost everywhere on the interior of its domain. As a consequence $g$ is differentiable almost everywhere on the interior of its domain,

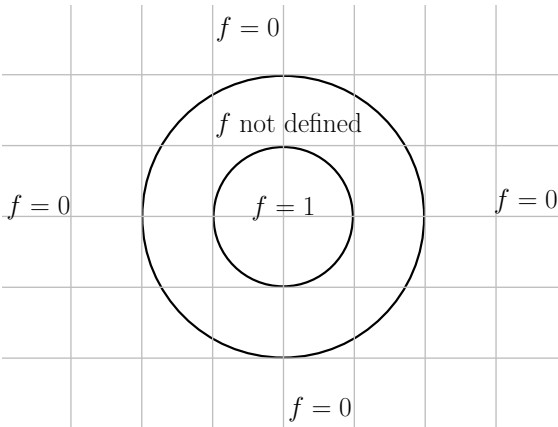

Figure 7: Representation of the function $f$ defined in Equation (32).

$\text{int}(\text{dom}(g))$. Denoting as $\Gamma \subset \text{int}(\text{dom}(g))$ the subset on which $g$ is differentiable, we thus have $\text{Leb}(\text{int}(\text{dom}(g)) \setminus \Gamma) = 0$.

Moreover, for $x \in \Gamma$, $g$ is differentiable at $x$, so by Corollary 2, we have $x = \text{Prox}_{\gamma g}(x + \gamma \nabla g(x)) \in \text{Prox}_{\gamma g}(\mathbb{R}^d)$. So we get $\Gamma \subset \text{Prox}_{\gamma g}(\mathbb{R}^d)$. Therefore, we can deduce that $\text{Leb}(\text{int}(\text{dom}(g)) \setminus \text{Prox}_{\gamma g}(\mathbb{R}^d)) = 0$.

Combining the fact that $\text{Leb}(\text{dom}(g) \setminus \text{int}(\text{dom}(g))) = 0$ and $\text{Leb}(\text{int}(\text{dom}(g)) \setminus \text{Prox}_{\gamma g}(\mathbb{R}^d)) = 0$, we get that $\text{Leb}(\text{dom}(g) \setminus \text{Prox}_{\gamma g}(\mathbb{R}^d)) = 0$. It shows equation (30).

Moreover for $x \in \mathbb{R}^d$ and $y \in \text{dom}(g)$, by definition of the proximal operator, we get $g(\text{Prox}_{\gamma g}(x)) + \frac{1}{2\gamma}\|x - \text{Prox}_{\gamma g}(x)\|^2 \le g(y) + \frac{1}{2\gamma}\|x - y\|^2$. So $g(\text{Prox}_{\gamma g}(x)) < +\infty$, which gives that $\text{Prox}_{\gamma g}(\mathbb{R}^d) \subset \text{dom}(g)$. Because $\text{dom}(g)$ is convex, we get that $\text{Conv}(\text{Prox}_{\gamma g}(\mathbb{R}^d)) \subset \text{dom}(g)$. Combining with equation (30), it gives equation (31). $\qquad\square$

## E   On the strong convexity on non-convex sets

**Definition 4.** *For an open subset $\mathbf{S} \subset \mathbb{R}^d$ and $\mu \ge 0$, a function $f \in \mathcal{C}^2(\mathbf{S}, \mathbb{R})$ is said to be $\mu$-strongly convex on $\mathbf{S}$ if and only if $\forall x \in \mathbf{S}$, $\nabla^2 f \succeq \mu I_d$.*

A function that is $\mu$-strongly convex on $\mathbf{S}$ with $\mu = 0$ is called convex on $\mathbf{S}$.

**Proposition 3.** *For a convex open set $\mathbf{S}$ and a function $f \in \mathcal{C}^2(\mathbf{S}, \mathbb{R})$, $f$ is convex on $\mathbf{S}$ if and only if $\forall t \in [0,1]$, $\forall x, y \in \mathbf{S}$, $f(tx + (1-t)y) \le tf(x) + (1-t)f(y)$.*

Proposition 3 is standard and we let the proof to the reader.

**Remark 1.** *If $\mathbf{S}$ is non-convex, then even if $f$ is convex on $\mathbf{S}$, the inequality $\forall t \in [0,1]$, $\forall x, y \in \mathbf{S}$, $f(tx + (1-t)y) \le tf(x) + (1-t)f(y)$ is not necessarily verified.*

*This is for instance the case with $\mathbf{S} = \{z \in \mathbb{R}^d | \|z\| < 1\} \cup \{z \in \mathbb{R}^d | \|z\| > 2\}$ and $f$ defined by*

$$f(x) = 1 \text{ if } \|z\| < 1 \tag{32}$$
$$f(x) = 0 \text{ if } \|z\| > 2, \tag{33}$$

*see Figure 7 for a representation of $f$. Then, $\forall x \in \mathbf{S}$, $\nabla^2 f = 0$ so $f$ is convex on $\mathbf{S}$. However, for $u \in \mathbb{R}^d$ such that $\|u\| = 1$, we have*

$$1 = f(0) = f(\frac{1}{2}3u + \frac{1}{2}(-3u)) > \frac{1}{2}f(3u) + \frac{1}{2}f(-3u) = 0.$$

**Lemma 16.** *If $g$ is $L_g$-smooth on $\mathbb{R}^d$ with $\gamma L_g < 1$ and there exists $R > 0$, such that $g$ is $\mu$-strongly convex on $\mathbb{R}^d \setminus B(0, R)$, i.e. $\nabla^2 g(x) \succeq \mu I_d$, then there exists $R_0 > 0$, such that $\forall \gamma \in [0, \frac{1}{2L_g}]$, $g^\gamma$ is $\frac{\mu}{1+\mu\gamma}$-strongly convex on $\mathbb{R}^d \setminus B(0, R_0)$.*

*Proof.* As $g$ is strongly convex on $\mathbb{R}^d \setminus B(0, R)$, it is coercive, so $g$ is lower bounded on $\mathbb{R}^d$, i.e. there exists $g_{inf} = \min_{x \in \mathbb{R}^d} g(x) > -\infty$ such that $\forall x \in \mathbb{R}^d$, $g(x) \geq g_{inf}$. We recall that the Moreau envelope of $g$ is defined as

$$g^\gamma(x) = \inf_{y \in \mathbb{R}^d} \frac{1}{2\gamma} \|x - y\|^2 + g(y). \tag{34}$$

For $x \in \mathbb{R}^d$, such that $\|x\| \geq R + 1$, for $y \in \mathbb{R}^d$, such that $\|y - x\| \geq \|x\| - R - 1$, we have

$$\frac{1}{2\gamma} \|x - y\|^2 + g(y) \geq \frac{1}{2\gamma} (\|x\| - R - 1)^2 + g_{inf}. \tag{35}$$

Since $\nabla g$ is $L_g$-Lipschitz, we get that

$$g(x) \leq g(0) + \langle \nabla g(0), x \rangle + \frac{L_g}{2} \|x\|^2. \tag{36}$$

Due to $L_g < \frac{1}{\gamma}$, we get that asymptotically $\frac{1}{2\gamma} (\|x\| - R - 1)^2 + g_{inf}$ is larger than $g(0) + \langle \nabla g(0), x \rangle + \frac{L_g}{2} \|x\|^2$. More formally, we study the quadratic form

$$\frac{1}{2\gamma} (\|x\| - R - 1)^2 + g_{inf} - \left( g(0) + \langle \nabla g(0), x \rangle + \frac{L_g}{2} \|x\|^2 \right)$$

$$= \frac{1}{2} \left( \frac{1}{\gamma} - L_g \right) \|x\|^2 - \frac{R+1}{\gamma} \|x\| - \langle \nabla g(0), x \rangle + g_{inf} - g(0) + \frac{(R+1)^2}{2\gamma}$$

$$\geq \frac{1}{2} \left( \frac{1}{\gamma} - L_g \right) \|x\|^2 - \left( \frac{R+1}{\gamma} + \|\nabla g(0)\| \right) \|x\| + g_{inf} - g(0) + \frac{(R+1)^2}{2\gamma}$$

$$\geq \left( \frac{1}{4} \left( \frac{1}{\gamma} - L_g \right) \|x\| - \left( \frac{R+1}{\gamma} + \|\nabla g(0)\| \right) \right) \|x\|$$

$$+ \frac{1}{4} \left( \frac{1}{\gamma} - L_g \right) \|x\|^2 + g_{inf} - g(0) + \frac{(R+1)^2}{2\gamma}.$$

Then, the previous quadratic form is non negative if $x \in \mathbb{R}^d$ verifies

$$\|x\| \geq 4 \frac{\frac{R+1}{\gamma} + \|\nabla g(0)\|}{\frac{1}{\gamma} - L_g}$$

$$\text{and } \|x\| \geq 2 \sqrt{\frac{g_{inf} - g(0) + \frac{(R+1)^2}{2\gamma}}{\frac{1}{\gamma} - L_g}}.$$

By defining

$$r_\gamma = \max \left( 4 \frac{\frac{R+1}{\gamma} + \|\nabla g(0)\|}{\frac{1}{\gamma} - L_g}, 2 \sqrt{\frac{g_{inf} - g(0) + \frac{(R+1)^2}{2\gamma}}{\frac{1}{\gamma} - L_g}} \right),$$

we get that $\forall x \in \mathbb{R}^d \setminus B(0, r_\gamma)$,

$$g(0) + \langle \nabla g(0), x \rangle + \frac{L_g}{2} \|x\|^2 \leq \frac{1}{2\gamma} (\|x\| - R - 1)^2 + g_{inf}.$$

Moreover, for $\gamma \leq \frac{1}{2L_g}$, $\frac{1}{\gamma} - L_g \geq L_g$ and $1 - \gamma L_g \geq \frac{1}{2}$. So we get that

$$r_\gamma \leq R_0 = \max \left( 8(R+1) + \frac{4\|\nabla g(0)\|}{L_g}, 2 \sqrt{\frac{g_{inf} - g(0)}{L_g} + (R+1)^2} \right).$$

Therefore, we get from (36) that $\forall \gamma \in [0, \frac{1}{2L_g}]$ and $\forall x \in \mathbb{R}^d \setminus B(0, R_0)$,

$$g(x) \leq g(0) + \langle \nabla g(0), x \rangle + \frac{L_g}{2} \|x\|^2 \leq \frac{1}{2\gamma} (\|x\| - R - 1)^2 + g_{inf}.$$

By combining the previous inequality and Equation (35), we get that for $x \in \mathbb{R}^d \setminus B(0, R_0)$ and for $y$ such that $\|y - x\| \geq \|x\| - R - 1$, we have

$$g(x) \leq \frac{1}{2\gamma} \|x - y\|^2 + g(y).$$

Hence for $x \in \mathbb{R}^d \setminus B(0, R_0)$, the infimum in the computation of $g^\gamma$ (see relation 34) is reached inside the ball $B(x, \|x\| - R - 1)$, i.e.

$$g^\gamma(x) = \inf_{y \in \mathbb{R}^d, \|y - x\| \leq \|x\| - R - 1} \frac{1}{2\gamma} \|x - y\|^2 + g(y).$$

Moreover, $B(x, \|x\| - R - 1) \subset \mathbb{R}^d \setminus B(0, R)$ and $B(x, \|x\| - R - 1)$ is a convex set. As we assumed that $g$ is $\mu$-strongly convex on $\mathbb{R}^d \setminus B(0, R)$, then $(x, y) \in \mathbb{R}^d \times B(x, \|x\| - R - 1) \mapsto \frac{1}{2\gamma} \|x - y\|^2 + g(y) - \frac{\mu}{2(1+\gamma\mu)} \|x\|^2$ is convex. Then, by Lemma 3, $g^\gamma - \frac{\mu}{2(1+\gamma\mu)} \|x\|^2$ is convex on $\mathbb{R}^d \setminus B(0, R_0)$ which ends the proof. $\qquad \square$

# F  Proof of Section 2

## F.1  Introduction to Markov Process and Markov Chain: definition and concepts

In this part, we introduce the tools of Markov Chain and Markov processes that will be useful for our analysis. We refer to the book of [26] for a detailed introduction.

We define a continuous-time Markov stochastic differential equation by

$$\mathrm{d}X_t = b(X_t)\mathrm{d}t + \sqrt{2}\mathrm{d}w_t \tag{37}$$
$$X_0 \sim \mu, \tag{38}$$

where $\mathrm{d}w_t$ is a Wiener process and $\mu$ a probability law on $\mathbb{R}^d$. The deterministic part of the equation $b(X_t)$ is called the drift. If the drift $b$ is Lipschitz, then the previous equation has a strong solution, i.e. there is a unique solution $(X_t)_{t \geq 0}$ up to negligible fluctuation. In this paper, we will always be in the case $b$ Lipschitz which ensures that the previous equation is well defined and defines a unique $X_t$.

The constant standard deviation of $\sqrt{2}$ ensures that if $b(X_t) = -\nabla f(X_t)$ and $b$ is Lipschitz, then the process converges in law to $\pi \propto e^{-f}$ [79]. The following tools are not restricted to the case of this constant standard deviation case. In the case of $\mu = \delta_x$, we say that the process $X_t$ starts on $x$.

We will study the law of $X_t$ for each time $t \geq 0$. To explore these laws, we introduce the Markov semi group $P_t$ defined for $x \in \mathbb{R}^d$ and $A$ a measurable set by

$$P_t(x, A) = \mathbb{P}[X_t \in A | X_0 = x] \tag{39}$$

$P_t(x, A)$ quantifies the probability that the process starting at the point $x$ and following the stochastic equation (37) is in the set $A$ at time $t$. It is called a semi-group because if we define $(P_t P_s)(x, A) = \int_{\mathbb{R}^d} P_t(y, A) P_s(x, dy)$, then we have $\forall t, s \geq 0$, $P_t P_s = P_{t+s}$. We introduce two operations with the object $P_t$. For a distribution $\mu$ on $\mathbb{R}^d$, we define $\mu P_t(A) := \int_{\mathbb{R}^d} P_t(x, A) d\mu(x)$. Thus $\mu P_t$ is the distribution $X_t$ on $\mathbb{R}^d$ knowing that $X_0$ follow $\mu$. In particular $P_0(x, A) = \delta_x(A)$ and $\mu P_0 = \mu$. For a function $\phi : \mathbb{R}^d \to \mathbb{R}$, we define $P_t \phi(x) := \int_{\mathbb{R}^d} \phi(y) P_t(x, dy) = \mathbb{E}(\phi(X_t) | X_0 = x)$. Thus $P_t \phi$ is a function of $\mathbb{R}^d$ to $\mathbb{R}$. In particular $P_0 \phi = \phi$. The two previous operations can be combined together to get $\mu P_t \phi = \mathbb{E}(\phi(X_t) | X_0 \sim \mu)$.

With a step-size $\gamma > 0$, the Markov process (37) can be discretized with the Euler-Maruyama scheme leading to

$$X_{k+1} = X_k + \gamma b(X_k) + \sqrt{2\gamma} Z_{k+1} \tag{40}$$
$$X_0 \sim \mu, \tag{41}$$

with $Z_{k+1} \sim \mathcal{N}(0, I_d)$. This process is a Markov Chain. The deterministic part $b(X_k)$ is called the drift. Similarly to the Markov semi-group, we introduce the Markov kernel $R_\gamma$ defined for $x \in \mathbb{R}^d$ and $A \subset \mathbb{R}^d$ by $R_\gamma(x, A) = \mathbb{P}[X_1 \in A | X_0 = x]$. We also define

$$R_\gamma^k(x, A) = \mathbb{P}[X_k \in A | X_0 = x]. \tag{42}$$

This Markov kernel $R_\gamma^k(x, A)$ quantifies the probability of being in a set $A$ at step $k$ for the Markov Chain starting at point $x$. We have a semi-group property in the sense that $\forall n, m \geq 0$, $R_\gamma^n R_\gamma^m = R_\gamma^{n+m}$. It can be seen as a transition matrix if the state space is not $\mathbb{R}^d$ but a finite space. In fact, for a probability distribution $\mu$, we define $\mu R_\gamma^k(A) = \int_{\mathbb{R}^d} R_\gamma^k(x, A) d\mu(x)$ the probability distribution of the discrete-time process $X_k$ at time $k$ knowing that $X_0 \sim \mu$. In particular $R_\gamma^0(x, A) = \delta_x(A)$ and $\mu R_\gamma^0 = \mu$.

We can also define the application of the Markov kernel on a function $\phi : \mathbb{R}^d \to \mathbb{R}$ by $R_\gamma^k \phi(x) = \int_{\mathbb{R}^d} \phi(y) R_\gamma^k(x, dy)$. In particular $R_\gamma^0 \phi = \phi$.

The Markov kernel will be the core object to study the evolution of the law of the iterates $X_k$.

For a function $V : \mathbb{R}^d \to [1, +\infty)$, we introduce the $V$-norm between two distributions $\mu, \nu$ defined on $\mathbb{R}^d$ by

$$\|\mu - \nu\|_V := \sup_{\substack{\phi:\mathbb{R}^d \to \mathbb{R} \\ |\phi| \leq V}} \int_{\mathbb{R}^d} \phi(d\mu - d\nu), \tag{43}$$

The supremum is taken on $f$ measurable. Note that the function $V$ that defines the $V$-norm has nothing to do with the potential $V$ of the law $\pi \propto e^{-V}$ that we aim to sample. We keep the notation $V$-norm as it is standard in the literature [26, 59]. In particular for $V = 1$, it defines the total variation norm defined by

$$\|\mu - \nu\|_{TV} := \sup_{\substack{\phi:\mathbb{R}^d \to \mathbb{R} \\ |\phi| \leq 1}} \int_{\mathbb{R}^d} \phi(d\mu - d\nu). \tag{44}$$

Note that, for any $V \geq 1$, we have $\|\mu - \nu\|_{TV} \leq \|\mu - \nu\|_V$.

### F.2 Technical lemmas

In this section we present technical lemmas that link the $V$-norm to the $\mathbf{W}_p$ distance or the $TV$-distance and an application of Girsanov's theorem that will be useful for our analysis.

**Lemma 17.** *[26, Theorem 19.1.7] For $V : \mathbb{R}^d \to [1, +\infty)$, we have*

$$\|\mu - \nu\|_V = \inf_{(X,Y)\sim\beta\in\Pi} \mathbb{E}\left((V(X) + V(Y))\mathbb{1}_{X\neq Y}\right),$$

*with the infimum taken on the set $\Pi$ of coupling $\beta$ between $\mu$ and $\nu$. A coupling between $\mu$ and $\nu$ is a probability law $\beta$ on $\mathbb{R}^d \times \mathbb{R}^d$ with marginals $\mu$ and $\nu$.*

**Lemma 18.** *For $p \geq 1$ and $\mu, \nu$ two probability measures defined on $(\mathbb{R}^d, \mathcal{B}(\mathbb{R}^d))$, we have*

$$\mathbf{W}_p^p(\mu, \nu) \leq 2^{p-1}\|\mu - \nu\|_{V_p},$$

*with $V_p = 1 + \|\cdot\|^p$.*

*Proof.* By definition of the Wasserstein distance, we have

$$\mathbf{W}_p^p(\mu, \nu) = \inf_{(X,Y)\sim\beta} \mathbb{E}\left(\|X - Y\|^p\right),$$

where the inf is taken on the set of coupling $\beta$ between $\mu$ and $\nu$. Using that the function $x \mapsto x^p$ is convex, we have that $(x + y)^p \leq 2^{p-1}(x^p + y^p)$ and Lemma 17. This leads to

$$\begin{aligned}
\mathbf{W}_p^p(\mu, \nu) &= \inf_{(X,Y)\sim\beta} \mathbb{E}\left((\|X - Y\|^p)\mathbb{1}_{X\neq Y}\right) \\
&\leq 2^{p-1} \inf_{(X,Y)\sim\beta} \mathbb{E}\left((\|X\|^p + \|Y\|^p)\mathbb{1}_{X\neq Y}\right) \\
&\leq 2^{p-1} \inf_{(X,Y)\sim\beta} \mathbb{E}\left((1 + \|X\|^p + 1 + \|Y\|^p)\mathbb{1}_{X\neq Y}\right) \\
&= 2^{p-1}\|\mu - \nu\|_{V_p},
\end{aligned}$$

with $V_p = 1 + \|\cdot\|^p$. $\qquad\square$

**Lemma 19.** *For $\mu, \nu$ two distributions on $\mathbb{R}^d$, we have*

$$\|\mu - \nu\|_V \leq \left(\mu(V^2) + \nu(V^2)\right)^{\frac{1}{2}} \|\mu - \nu\|_{TV}^{\frac{1}{2}}.$$

*Proof.* By Lemma 17 and the Cauchy-Schwarz inequality, we get

$$\begin{aligned}
\|\mu - \nu\|_V &= \inf_{(X,Y)\sim\beta} \mathbb{E}\left((V(X) + V(Y))\mathbb{1}_{X\neq Y}\right) \\
&\leq \inf_{(X,Y)\sim\beta} \mathbb{E}\left((V(X) + V(Y))^2\right)^{\frac{1}{2}} \mathbb{E}\left(\mathbb{1}_{X\neq Y}\right)^{\frac{1}{2}} \\
&\leq \inf_{(X,Y)\sim\beta} \mathbb{E}\left(2(V^2(X) + V^2(Y))\right)^{\frac{1}{2}} \mathbb{E}\left(\mathbb{1}_{X\neq Y}\right)^{\frac{1}{2}} \\
&\leq \left((\mu(V^2) + \nu(V^2))\right)^{\frac{1}{2}} \|\mu - \nu\|_{TV}^{\frac{1}{2}},
\end{aligned}$$

using that $\|\mu - \nu\|_{TV} = 2\inf_{(X,Y)\sim\beta} \mathbb{E}\left(\mathbb{1}_{X\neq Y}\right)$. $\qquad\square$

**Lemma 20.** *For $T > 0$, let $b^1, b^2 : \mathcal{C}([0,T], \mathbb{R}^d) \times [0,T] \to \mathbb{R}^d$ be two drifts such that $\forall i \in \{1,2\}$, $\mathrm{d}x_t^i = b^i((x_u^i)_{u\in[0,T]}, t)\mathrm{d}t + \sqrt{2}\mathrm{d}w_t$ admits a strong solution with $x_0^i \sim \mu$ with the Markov semigroup $(P_t^i)_{t\in\mathbb{R}_+}$ (see definition in equation (39)) and $(w_t)_{t\geq 0}$ a Wiener process. Moreover, assume that $\mathbb{P}\left[\int_0^T \|b^i((x_u^i)_{u\in[0,T]}, t)\|^2 + \|b^i((w_u)_{u\in[0,T]}, t)\|^2\mathrm{d}t < +\infty\right] = 1$. Let $V : \mathbb{R}^d \to [1, +\infty)$ measurable, then we have*

$$\|\mu P_T^1 - \mu P_T^2\|_V \leq \left(\mu P_T^1(V^2) + \mu P_T^2(V^2)\right)^{\frac{1}{2}} \left(\int_0^T \mathbb{E}\left(\|b^1((x_u^1)_{u\in[0,T]}, t) - b^2((x_u^1)_{u\in[0,T]}, t)\|^2\right)\mathrm{d}t\right)^{\frac{1}{2}}.$$

Lemma 20 is a generalization of [59, Lemma 19] for any distribution $\mu$ on $\mathbb{R}^d$ instead of Dirac distributions $\delta_x$.

*Proof.* We follow the sketch of the proof of [59, Lemma 19]. For $i \in \{1, 2\}$, we denote by $\mu^i$ the distribution of $(X_t^i)_{t\in[0,T]}$ on the Wiener space $(\mathcal{C}([0,T], \mathbb{R}), \mathcal{B}(\mathcal{C}([0,T], \mathbb{R})))$, with $x_0^i \sim \mu$ and $\mathrm{d}x_t^i = b^i((x_u^i)_{u\in[0,T]}, t)\mathrm{d}t + \sqrt{2}\mathrm{d}w_t$. We denote by $\mu_W$ the distribution of $(w_t)_{t\in[0,T]}$, the Wiener process.

By the generalized Pinsker's inequality [27, Lemma 24], we have

$$\|\mu P_T^1 - \mu P_T^2\|_V \leq \sqrt{2}\left(\mu P_T^1(V^2) + \mu P_T^2(V^2)\right)^{\frac{1}{2}} \mathsf{KL}^{\frac{1}{2}}(\mu^1|\mu^2). \tag{45}$$

We assumed $\mathbb{P}\left[\int_0^T \|b^i((x_u^i)_{u\in[0,T]}, t)\|^2 + \|b^i((w_u)_{u\in[0,T]}, t)\|^2\mathrm{d}t < +\infty\right] = 1$ for any $i \in \{1, 2\}$. Hence we can apply Girsanov's Theorem [61, Theorem 7.7] and get $\mu_W$-almost surely that

$$\frac{\mathrm{d}\mu^1}{\mathrm{d}\mu_W}((w_u)_{u\in[0,T]}, t) = \exp\left[\frac{1}{2}\int_0^T \langle b^1((w_u)_{u\in[0,T]}, t), \mathrm{d}w_t\rangle - \frac{1}{4}\int_0^T \|b^1((w_u)_{u\in[0,T]}, t)\|^2\mathrm{d}t\right]$$

$$\frac{\mathrm{d}\mu_W}{\mathrm{d}\mu^2}((w_u)_{u\in[0,T]}, t) = \exp\left[-\frac{1}{2}\int_0^T \langle b^2((w_u)_{u\in[0,T]}, t), \mathrm{d}w_t\rangle + \frac{1}{4}\int_0^T \|b^2((w_u)_{u\in[0,T]}, t)\|^2\mathrm{d}t\right],$$

where $\langle\cdot,\cdot\rangle$ is the canonic scalar product on $\mathbb{R}^d$. Note that $\int_0^T \langle b^i((w_u)_{u\in[0,T]}, t), \mathrm{d}w_t\rangle$, for $i \in \{1, 2\}$, are stochastic integrals.

Hence, we get

$$\begin{aligned}
\mathsf{KL}(\mu^1|\mu^2) &= \mathbb{E}\left(\log\frac{\mathrm{d}\mu^1}{\mathrm{d}\mu^2}((x_u^1)_{u\in[0,T]}, t)\right) \\
&= \frac{1}{4}\int_0^T \mathbb{E}\left[\|b^1((x_u^1)_{u\in[0,T]}, t) - b^2((x_u^1)_{u\in[0,T]}, t)\|^2\right]. \tag{46}
\end{aligned}$$

By putting together equation (45) and equation (46), we get the desired result.

$\qquad\square$

### F.3 Sampling stability for Markov Chain in $V_p$-norm

In this part, we demonstrate the sampling stability of discrete Markov processes in the $V_p$-norm. Theorem 5 is a significant refinement of [75, Theorem 1] as it does not involve any discretization error and it generalizes previous known results in $\mathbf{W}_1$ and $TV$-norm to the $V_p$-norms, which will imply the result in $TV$-norm and $\mathbf{W}_p$ distance for any $p \in \mathbb{N}^\star$.

We introduce two discrete-time Markov Chains $X_k^i$, $i \in \{1, 2\}$, defined for step-size $\gamma > 0$ by

$$X_{k+1}^i = X_k^i - \gamma b^i(X_k^i) + \sqrt{2\gamma} Z_{k+1}^i \tag{47}$$

with $Z_{k+1}^i \sim \mathcal{N}(0, I_d)$, two independent standard Gaussian distributions, and two drifts $b^i \in \mathcal{C}^0(\mathbb{R}^d, \mathbb{R}^d)$.

We will show a similar result than Theorem 1 in $V_p$-norm for $V_p = 1 + \|x\|^p$ with $p \in \mathbb{N}^\star$ and then deduce Theorem 1 in $TV$-norm and $\mathbf{W}_p$ (see Section F.4).

**Theorem 5.** *If $b^1$ and $b^2$ satisfy Assumption 1, the two Markov Chains $X_k^1$ and $X_k^2$ are geometrically ergodic, with invariant laws $\pi_\gamma^1, \pi_\gamma^1$. Moreover, for $\gamma_0 = \frac{m}{L^2}$, with $L$, $m$ defined in Assumption 1, $p \in \mathbb{N}^\star$ and $V_p(x) = 1 + \|x\|^p$, there exists $G_p \geq 0$ such that $\forall \gamma \in (0, \gamma_0]$, we have*

$$\|\pi_\gamma^1 - \pi_\gamma^2\|_{V_p} \leq G_p \left( \mathbb{E}_{Y \sim \pi_\gamma^1} \left( \|b^1(Y) - b^2(Y)\|^2 \right) \right)^{\frac{1}{2}}.$$

Note that the pseudo-distance between the drift $\|b^1 - b^2\|_{\ell_2(\pi_\gamma^1)} = \left( \mathbb{E}_{Y \sim \pi_\gamma^1} \left( \|b^1(Y) - b^2(Y)\|^2 \right) \right)^{\frac{1}{2}}$ is not symmetric in the drifts $b^1, b^2$ which allows to evaluate this pseudo-distance only by sampling $\pi_\gamma^1$, see [75] for empirical evaluations. It could be symmetries with $\min \left( \|b^1 - b^2\|_{\ell_2(\pi_\gamma^1)}, \|b^1 - b^2\|_{\ell_2(\pi_\gamma^2)} \right)$ instead of $\|b^1 - b^2\|_{\ell_2(\pi_\gamma^1)}$.

*Proof.* **Scheme of the Proof**

In a first part, we demonstrate that the two discrete-time Markov Chains $X_k^1, X_k^2$ are geometrically ergodic. Thus, we can define and study their invariant laws $\pi_\gamma^1$ and $\pi_\gamma^2$.

In a second part, we study the distance between these two invariant laws. The idea is to approximate them by the law of the processes at finite time $k\gamma$ starting at a point $x \in \mathbb{R}^d$, i.e. $\delta_x R_{\gamma,1}^k$ and $\delta_x R_{\gamma,2}^k$. Then the ergodicity allows us to quantify the distance between the two laws at finite time $k\gamma$ by the distance between the two laws at a bounded time ($m\gamma \approx 1$ in the proof). The Girsanov theorem (Lemma 20) gives a quantification of the two laws at a bounded time by a distance on the drift. Finally, we look at the asymptotic $k \to +\infty$ of the obtained control to get the desired result on the invariant laws.

To simplify the notations, without loss of generality, we assume that $b^1$ and $b^2$ verify Assumption 1 with the same constants $L, R \geq 0$ and $m > 0$.

An introduction to Markov continuous-time processes and Markov discrete-time processes is given in Appendix F.1 and technical lemmas proofs can be founded in Appendix F.2.

### Proof that $X_k^1, X_k^2$ are geometrically ergodic

First, by [23, Corollary 2] and under Assumption 1 with $\gamma_0 = \frac{m}{L^2}$ (see [23, Proposition 12] for the expression of $\gamma_0$), there exist $\rho \in (0, 1)$ and $D \geq 0$ such that $\forall \gamma \in (0, \gamma_0]$, $\forall i \in \{1, 2\}$, we have

$$\mathbf{W}_1(\delta_x R_{\gamma,i}^k, \delta_y R_{\gamma,i}^k) \leq D\rho^{k\gamma}\|x - y\|, \tag{48}$$

with $\delta_x$ the dirac distribution on $x \in \mathbb{R}^d$ and $R_{\gamma,i}$ the Markov kernel of the Markov Chain $i$ defined in equation (47) as introduced in Appendix F.1. Note that $\delta_x R_{\gamma,i}^k$ represents the law of the Markov Chain $X_k^i$ at step $k$ knowing that $X_0^i = x$. We want to generalize the previous inequality for any distributions $\mu, \nu$ on $\mathbb{R}^d$ instead of $\delta_x, \delta_y$. We denote $\beta^\star$ the optimal transport plan from $\mu$ to $\nu$, by definition of the $\mathbf{W}_1$ distance, we have

$$\mathbf{W}_1(\mu, \nu) = \int_{x_1, x_2 \in \mathbb{R}^d \times \mathbb{R}^d} \|x_1 - x_2\| \beta^\star(dx_1, dx_2). \tag{49}$$

By the Kantorovitch-Rubinstein Theorem [31, Theorem 4.1] [52], the dual formulation of the $\mathbf{W}_1$ distance writes

$$\mathbf{W}_1(\mu, \nu) = \sup_{\substack{\phi:\mathbb{R}^d \to \mathbb{R} \\ \mathrm{Lip}(\phi) \leq 1}} \int_{\mathbb{R}^d} \phi(d\mu - d\nu),$$

with $\mathrm{Lip}(\phi)$ the Lipschitz constant of $\phi$.

Then, by equation (48), equation (49) and the dual formulation of $\mathbf{W}_1$ and the fact that $\delta_{x_1} R^k_{\gamma,i}(dx) = R^k_{\gamma,i}(x_1, dx)$ thanks to the Markov kernel definition (42), we get

$$D\rho^{k\gamma}\mathbf{W}_1(\mu, \nu) = \int_{\mathbb{R}^d \times \mathbb{R}^d} D\rho^{k\gamma}\|x_1 - x_2\|\beta^\star(dx_1, dx_2)$$

$$\geq \int_{\mathbb{R}^d \times \mathbb{R}^d} \mathbf{W}_1(\delta_{x_1} R^k_{\gamma,i}, \delta_{x_2} R^k_{\gamma,i})\beta^\star(dx_1, dx_2)$$

$$= \int_{\mathbb{R}^d \times \mathbb{R}^d} \sup_{\substack{\phi:\mathbb{R}^d \to \mathbb{R} \\ \mathrm{Lip}(\phi) \leq 1}} \int_{\mathbb{R}^d} \phi(x)\left((\delta_{x_1} R^k_{\gamma,i})(dx) - (\delta_{x_2} R^k_{\gamma,i})(dx)\right)\beta^\star(dx_1, dx_2)$$

$$\geq \sup_{\substack{f:\mathbb{R}^d \to \mathbb{R} \\ \mathrm{Lip}(\phi) \leq 1}} \int_{\mathbb{R}^d} \phi(x)\int_{\mathbb{R}^d \times \mathbb{R}^d} \left((\delta_{x_1} R^k_{\gamma,i})(dx) - (\delta_{x_2} R^k_{\gamma,i})(dx)\right)\beta^\star(dx_1, dx_2)$$

$$= \sup_{\substack{\phi:\mathbb{R}^d \to \mathbb{R} \\ \mathrm{Lip}(\phi) \leq 1}} \int_{\mathbb{R}^d} \phi(x)\int_{\mathbb{R}^d \times \mathbb{R}^d} \left(R^k_{\gamma,i}(x_1, dx) - R^k_{\gamma,i}(x_2, dx)\right)\beta^\star(dx_1, dx_2)$$

$$= \sup_{\substack{\phi:\mathbb{R}^d \to \mathbb{R} \\ \mathrm{Lip}(\phi) \leq 1}} \int_{\mathbb{R}^d} \phi(x)\left(\left(\mu R^k_{\gamma,i}\right)(dx) - \left(\nu R^k_{\gamma,i}\right)(dx)\right)$$

$$= \mathbf{W}_1\left(\mu R^k_{\gamma,i}, \nu R^k_{\gamma,i}\right).$$

So we get for any distributions $\mu, \nu$ on $\mathbb{R}^d$,

$$\mathbf{W}_1\left(\mu R^k_{\gamma,i}, \nu R^k_{\gamma,i}\right) \leq D\rho^{k\gamma}\mathbf{W}_1(\mu, \nu).$$

This proves that for $k$ large enough, the Markov kernel $R^k_{\delta,i}$ is contractive with respect to the 1-Wasserstein distance. Moreover as $(\mathcal{P}_1(\mathbb{R}^d), \mathbf{W}_1)$ is a complete space [93, Theorem 6.18], the Picard fixed point theorem shows that there exists a unique $\pi^i_\gamma$, such that for $\forall k \geq 0$, $\pi^i_\gamma R^k_{\gamma,i} = \pi^i_\gamma$. This probability law $\pi^i_\gamma$ is called the invariant law of the Markov Chain.

**Approximation of the invariant laws by finite time laws**

We proved that $\pi^1_\gamma$ and $\pi^2_\gamma$ are well defined, we can now tackle the problem of quantifying the distance between these two distributions in $V_p$-norm. By the triangle inequality, for $k \in \mathbb{N}$ and

$$m = \left\lfloor \frac{1}{\gamma} \right\rfloor \tag{50}$$

we have

$$\|\pi^1_\gamma - \pi^2_\gamma\|_{V_p} \leq \|\pi^1_\gamma - \delta_x R^{km}_{\gamma,1}\|_{V_p} + \|\delta_x R^{km}_{\gamma,1} - \delta_x R^{km}_{\gamma,2}\|_{V_p} + \|\delta_x R^{km}_{\gamma,2} - \pi^2_\gamma\|_{V_p} \tag{51}$$

We need to control two type of term in the previous equation, the contractiveness in $V_p$-norm for the first and the third terms and the shift between the two law at time $k\gamma$ for the second term.

**Contractiveness in $V_p$-norm**

We study the contractiveness of the Markov kernel in $V_p$-norm. By Lemma 19 and [23, Corollary 2], there exist $\rho_p \in (0, 1)$ and $E_p \geq 0$ such that $\forall x, y \in \mathbb{R}^d$ and $\forall i \in \{1, 2\}$, we have

$$\|\delta_x R^k_{\gamma,i} - \delta_y R^k_{\gamma,i}\|_{V_p} \leq \left(\delta_x R^k_{\gamma,i}(V_p^2) + \delta_y R^k_{\gamma,i}(V_p^2)\right)^{\frac{1}{2}} \|\delta_x R^k_{\gamma,i} - \delta_y R^k_{\gamma,i}\|_{TV}^{\frac{1}{2}}$$

$$\leq E_p \left(\delta_x R^k_{\gamma,i}(V_p^2) + \delta_y R^k_{\gamma,i}(V_p^2)\right)^{\frac{1}{2}} \rho_p^{k\gamma}(1 + \|x - y\|)^{\frac{1}{2}}. \tag{52}$$

By [59, Lemma 16] and [59, Lemma 17], under Assumption 1, for any $m \in \mathbb{N}$ and $V_{2m}(x) = 1 + \|x\|^{2m}$ there exists $M_m \geq 0$ such that $\forall k \in \mathbb{N}$ and $\forall i \in \{1, 2\}$, we have

$$R_{\gamma,i}^k V_{2m}(x) \leq M_{2m} V_{2m}(x). \tag{53}$$

Thus, given that $V_p^2 \leq 2V_{2p}$ and $\delta_x(V_p^2) = V_p^2(x)$, we have $\forall x \in \mathbb{R}^d$

$$\delta_x R_{\gamma,i}^k V_p^2 \leq 2M_{2p} V_{2p}(x). \tag{54}$$

By injecting equation (54) into equation (52), using that $\sqrt{a+b} \leq \sqrt{a} + \sqrt{b}$ for $a, b \geq 0$ and that $\sqrt{V_{2p}} \leq V_p$, we get that for all $x \in \mathbb{R}^d$

$$\|\delta_x R_{\gamma,i}^k - \delta_y R_{\gamma,i}^k\|_{V_p} \leq \sqrt{2} E_p \sqrt{M_{2p}} \sqrt{V_{2p}(x) + V_{2p}(y)} \rho_p^{k\gamma} (1 + \|x - y\|)^{\frac{1}{2}}$$
$$\leq \sqrt{2} E_p \sqrt{M_{2p}} \left( V_p(x) + V_p(y) \right) \rho_p^{k\gamma} (1 + \|x\| + \|y\|)^{\frac{1}{2}}.$$

Using the inequality $\sqrt{1 + \|x\| + \|y\|} \leq 1 + \|x\| + \|y\| \leq 1 + \|x\| + 1 + \|y\| \leq 2^{1-\frac{1}{p}}(1 + \|x\|^p)^{\frac{1}{p}} + 2^{1-\frac{1}{p}}(1 + \|y\|^p)^{\frac{1}{p}} \leq 2(V_p(x) + V_p(y))$, we get

$$\|\delta_x R_{\gamma,i}^k - \delta_y R_{\gamma,i}^k\|_{V_p} \leq D_p \rho_p^{k\gamma} \left( V_p^2(x) + V_p^2(y) \right), \tag{55}$$

with $D_p := 4\sqrt{2} E_p \sqrt{M_{2p}}$. This inequality generalizes [59, Proposition 5] that was demonstrated only for $V_1$.

By integrating equation (55) for $x \sim \mu$ and $y \sim \nu$ and the triangle inequality of the $V$-norm, we get for any probability measures $\mu, \nu$

$$\|\mu R_{\gamma,i}^k - \nu R_{\gamma,i}^k\|_{V_p} \leq D_p \rho_p^{k\gamma} \left( \mu(V_p^2) + \nu(V_p^2) \right). \tag{56}$$

This proves in particular that with $\nu = \pi_\gamma^i$, which satisfies $\pi_\gamma^i R_{\gamma,i}^k = \pi_\gamma^i$, the Markov Chain $X_k^i$ is geometrically ergodic, i.e. the empirical measure of the Markov Chain at time $k \geq 0$ of one realization converges in $V_p$-norm geometrically to the invariant law of the process.

$$\|\pi_\gamma^1 - \delta_x R_{\gamma,1}^{km}\|_{V_p} + \|\delta_x R_{\gamma,2}^{km} - \pi_\gamma^2\|_{V_p} \leq D_p(x) \rho_p^{km\gamma} \tag{57}$$

with $D_p(x) := D_p \left( 2V_p^2(x) + \pi_\gamma^1(V_p^2) + \pi_\gamma^2(V_p^2) \right)$, thanks to $\delta_x(V_p^2) = V_p^2(x)$. This is a control for the first and the third terms of equation (51)

**Control of the shift of the two laws at time $k\gamma$**

The distribution $\delta_x R_{\gamma,i}^k$ is the distribution at time $t = k\gamma$ of the discrete-time Markov Chain $X_k^i$ starting at $x$. The term $\|\delta_x R_{\gamma,1}^{km} - \delta_x R_{\gamma,2}^{km}\|_{V_p}$ quantifies the deformation at time $t = k\gamma$ between the laws of the two processes starting at the same point run with two different drifts.

By the triangle inequality, we get

$$\|\delta_x R_{\gamma,1}^{km} - \delta_x R_{\gamma,2}^{km}\|_{V_p} = \| \sum_{j=0}^{k-1} \delta_x R_{\gamma,1}^{(k-j)m} R_{\gamma,2}^{jm} - \delta_x R_{\gamma,1}^{(k-j-1)m} R_{\gamma,2}^{(j+1)m}\|_{V_p}$$

$$\leq \sum_{j=0}^{k-1} \|\delta_x R_{\gamma,1}^{(k-j)m} R_{\gamma,2}^{jm} - \delta_x R_{\gamma,1}^{(k-j-1)m} R_{\gamma,2}^{(j+1)m}\|_{V_p}$$

$$= \sum_{j=0}^{k-1} \|\delta_x R_{\gamma,1}^{(k-j-1)m} R_{\gamma,1}^m R_{\gamma,2}^{jm} - \delta_x R_{\gamma,1}^{(k-j-1)m} R_{\gamma,2}^m R_{\gamma,2}^{jm}\|_{V_p}. \tag{58}$$

For $\phi : \mathbb{R}^d \to \mathbb{R}$ such that $\|\phi\| \leq V_p$ and due to $\pi_\gamma^2(\phi) \in \mathbb{R}$, we get

$$(\delta_x R_{\gamma,1}^{(k-j-1)m} R_{\gamma,1}^m R_{\gamma,2}^{jm} - \delta_x R_{\gamma,1}^{(k-j-1)m} R_{\gamma,2}^m R_{\gamma,2}^{jm})(\phi)$$
$$= \left( \delta_x R_{\gamma,1}^{(k-j-1)m} R_{\gamma,1}^m - \delta_x R_{\gamma,1}^{(k-j-1)m} R_{\gamma,2}^m \right) R_{\gamma,2}^{jm}(\phi)$$
$$= \left( \delta_x R_{\gamma,1}^{(k-j-1)m} R_{\gamma,1}^m - \delta_x R_{\gamma,1}^{(k-j-1)m} R_{\gamma,2}^m \right) \left( R_{\gamma,2}^{jm}(\phi) - \pi_\gamma^2(\phi) \right). \tag{59}$$

By equation (56) applied with $\mu = \delta_x$, $\nu = \pi_\gamma^2$, $i = 2$ and $k = jm$, thanks to $V_p(x) \geq 1$, we have

$$\|\delta_x R_{\gamma,2}^{jm} - \pi_\gamma^2\|_{V_p} \leq B \rho_p^{jm\gamma} V_p^2(x),$$

with $B := D_p(1 + \pi_\gamma^2(V_p^2))$.

By the definition of the $V_p$-distance (43), we have

$$\left| \delta_x R_{\gamma,2}^{jm}(\phi) - \pi_\gamma^2(\phi) \right| \leq B \rho_p^{jm\gamma} V_p^2(x).$$

Using that $\delta_x R_{\gamma,2}^{jm}(\phi) = R_{\gamma,2}^{jm}(\phi)(x)$ and that $V_p^2 \leq 2V_{2p}$, we deduce that

$$\left| \frac{R_{\gamma,2}^{jm}(f)(x) - \pi_\gamma^2(f)}{2B\rho_p^{jm\gamma}} \right| \leq V_{2p}(x). \tag{60}$$

Going back to equation (59), thanks to equation (60), we get

$$(\delta_x R_{\gamma,1}^{(k-j-1)m} R_{\gamma,1}^m R_{\gamma,2}^{jm} - \delta_x R_{\gamma,1}^{(k-j-1)m} R_{\gamma,2}^m R_{\gamma,2}^{jm})(\phi)$$
$$\leq \left| \delta_x R_{\gamma,1}^{(k-j-1)m} R_{\gamma,1}^m - \delta_x R_{\gamma,1}^{(k-j-1)m} R_{\gamma,2}^m \right| \left| R_{\gamma,2}^{jm}(\phi) - \pi_\gamma^2(\phi) \right|$$
$$\leq 2B\rho_p^{jm\gamma} \left| \delta_x R_{\gamma,1}^{(k-j-1)m} R_{\gamma,1}^m - \delta_x R_{\gamma,1}^{(k-j-1)m} R_{\gamma,2}^m \right| \left| \frac{R_{\gamma,2}^{jm}(\phi) - \pi_\gamma^2(\phi)}{2B\rho^{jm\gamma}} \right|$$
$$\leq 2B\rho_p^{jm\gamma} \|\delta_x R_{\gamma,1}^{(k-j-1)m} R_{\gamma,1}^m - \delta_x R_{\gamma,1}^{(k-j-1)m} R_{\gamma,2}^m\|_{V_{2p}}.$$

Taking the supremum on $f$ in the last inequality, we get

$$\|\delta_x R_{\gamma,1}^{(k-j-1)m} R_{\gamma,1}^m R_{\gamma,2}^{jm} - \delta_x R_{\gamma,1}^{(k-j-1)m} R_{\gamma,2}^m R_{\gamma,2}^{jm}\|_{V_p}$$
$$\leq 2B\rho_p^{jm\gamma} \|\delta_x R_{\gamma,1}^{(k-j-1)m} R_{\gamma,1}^m - \delta_x R_{\gamma,1}^{(k-j-1)m} R_{\gamma,2}^m\|_{V_{2p}}. \tag{61}$$

By injecting the previous inequality into equation (58), we get

$$\|\delta_x R_{\gamma,1}^{km} - \delta_x R_{\gamma,2}^{km}\|_{V_p} \leq 2B \sum_{j=0}^{k-1} \rho_p^{jm\gamma} \|\delta_x R_{\gamma,1}^{(k-j-1)m} R_{\gamma,1}^m - \delta_x R_{\gamma,1}^{(k-j-1)m} R_{\gamma,2}^m\|_{V_{2p}}. \tag{62}$$

### Application of the Girsanov Theorem to get control at bounded time $m\gamma$

The previous inequality shows that we can bound $\|\delta_x R_{\gamma,1}^{km} - \delta_x R_{\gamma,2}^{km}\|_{V_p}$ with quantities of the form $\|\mu R_{\gamma,1}^m - \mu R_{\gamma,2}^m\|_{V_p}$. This last quantity has a temporal duration of $m\gamma \in (1-\gamma, 1]$ due to $m = \lfloor \frac{1}{\gamma} \rfloor$, which allows us to apply Girsanov's theory to bound it by the drift difference. To do so, we need to introduce a continuous version of the Markov Chain:

$$\mathrm{d}x_t^i = -b_\gamma^i((x_u^i)_{u \in [t-\gamma, t]}, t)\mathrm{d}t + \sqrt{2}\mathrm{d}w_t^i, \tag{63}$$

with $(w_t^i)_{t \geq 0}$ a Wiener process and $b_\gamma^i((x_u)_{u \in [t-\gamma, t]}, t) = b^i(x_{\lfloor \frac{t}{\gamma} \rfloor \gamma})$, with $b^i$ defined in equation (47). Here the drift $b_\gamma^i$ is non-homogeneous, i.e. it depends on the time $t$, and it is path-dependent. This process is a continuous counterpart of the Markov Chain (47). In fact, by integrating equation (63) for $t \in [k\gamma, (k+1)\gamma)$, we get

$$\int_{k\gamma}^{(k+1)\gamma} \mathrm{d}x_t^i = - \int_{k\gamma}^{(k+1)\gamma} b^i(x_{k\gamma}^i)\mathrm{d}t + \sqrt{2} \int_{k\gamma}^{(k+1)\gamma} \mathrm{d}w_t^i$$
$$x_{(k+1)\gamma}^i - x_{k\gamma}^i = -\gamma b^i(x_{k\gamma}^i) + \sqrt{2\gamma}Z_{k+1},$$

with $Z_{k+1} \sim \mathcal{N}(0, I_d)$ because $\int_{k\gamma}^{(k+1)\gamma} \mathrm{d}w_t^i \sim \mathcal{N}(0, \gamma I_d)$. Therefore, $x_{k\gamma}^i$ initialized with the probability law $\mu$ has the same law than $X_k^i$ defined in equation (47) initialized with the law $\mu$.

Under Assumption 1, the drift is Lipschitz with respect to the infinite norm, i.e. for $\phi, \psi \in \mathcal{C}([0, \gamma], \mathbb{R}^d)$

$$\left\| b_\gamma^i((\phi(u))_{u\in[0,\gamma]}, t) - b_\gamma^i((\psi(u))_{u\in[0,\gamma]}, t) \right\| = \left\| b^i(\phi(\lfloor \frac{t}{\gamma} \rfloor \gamma)) - b^i(\psi(\lfloor \frac{t}{\gamma} \rfloor \gamma)) \right\|$$

$$\leq L \left\| \phi(\lfloor \frac{t}{\gamma} \rfloor \gamma) - \psi(\lfloor \frac{t}{\gamma} \rfloor \gamma) \right\| \leq L \|\phi - \psi\|_\infty.$$

By [63, Theorem 5.2.2], equation (63) has a unique strong solution $x_t^i$ and

$$\mathbb{E}(\|b_\gamma^i((x_u^i)_{u\in[t-\gamma,t]}, t)\|) < +\infty.$$

So assumptions of Lemma 20 are verified, and we get that for $T > 0$

$$\|\mu P_T^1 - \mu P_T^2\|_{V_p} \leq \left(\mu P_T^1(V_p^2) + \mu P_T^2(V_p^2)\right)^{\frac{1}{2}} \left(\int_0^T \mathbb{E}\left(\|b^1(x_t^1, t) - b^2(x_t^1, t)\|^2\right) dt\right)^{\frac{1}{2}}, \quad (64)$$

with $V_p = 1 + \|\cdot\|^p$, $P_t^i$ for $i \in \{1, 2\}$ the Markov semi-group at time $t$ (define in equation (39)) of the process following equation (63) and $(x_t^1)_{t\geq 0}$ the stochastic process following equation (63) with the initial law $x_0^1 \sim \mu$. For $T = m\gamma$, we have that $P_{m\gamma}^i = R_{\gamma,i}^m$ so

$$\|\mu R_{\gamma,1}^m - \mu R_{\gamma,2}^m\|_{V_{2p}}$$
$$\leq \left(\mu R_{\gamma,1}^m(V_{2p}^2) + \mu R_{\gamma,2}^m(V_{2p}^2)\right)^{\frac{1}{2}} \left(\int_0^{m\gamma} \mathbb{E}\left(\|b^1(x_t^1, t) - b^2(x_t^1, t)\|^2\right) dt\right)^{\frac{1}{2}}. \quad (65)$$

Now, we can apply inequality (65) with $\mu = \delta_x R_{\gamma,1}^{(k-j-1)m}$ to get

$$\|\delta_x R_{\gamma,1}^{(k-j-1)m} R_{\gamma,1}^m - \delta_x R_{\gamma,1}^{(k-j-1)m} R_{\gamma,2}^m\|_{V_{2p}}$$
$$\leq \left(\delta_x R_{\gamma,1}^{(k-j-1)m} R_{\gamma,1}^m(V_{2p}^2) + \delta_x R_{\gamma,1}^{(k-j-1)m} R_{\gamma,2}^m(V_{2p}^2)\right)^{\frac{1}{2}}$$
$$\times \left(\int_0^{m\gamma} \mathbb{E}\left(\|b^1(x_{t,j}^1, t) - b^2(x_{t,j}^1, t)\|^2\right) dt\right)^{\frac{1}{2}}, \quad (66)$$

with $x_{t,j}^1$ the process starting with the law $\delta_x R_{\gamma,1}^{(k-j-1)m}$ at time $t = 0$ and verifying equation (63).

Using equation (53) we have $R_{\gamma,1}^{(k-j-1)m} R_{\gamma,1}^m(V_{2p}) \leq M_{2p} V_{2p}$. Recalling that $\delta_x(V_p^2) = V_p^2(x)$ and $V_p^2 \leq 2V_{2p}$, we obtain a uniform bound

$$\delta_x R_{\gamma,1}^{(k-j-1)m} R_{\gamma,1}^m(V_{2p}^2) \leq 2M_{4p} V_{4p}(x).$$

We also have that $R_{\gamma,2}^m(V_p^2) \leq 2M_{4p} V_{4p}$ so, using again (53), we get

$$\delta_x R_{\gamma,1}^{(k-j-1)m} R_{\gamma,2}^m(V_{2p}^2) \leq 2M_{4p} \delta_x R_{\gamma,1}^{(k-j-1)m} V_{2p} \leq 2M_{4p}^2 V_{4p}(x).$$

Then we obtain

$$\left(\delta_x R_{\gamma,1}^{(k-j-1)m} R_{\gamma,1}^m(V_{2p}^2) + \delta_x R_{\gamma,1}^{(k-j-1)m} R_{\gamma,2}^m(V_{2p}^2)\right)^{\frac{1}{2}} \leq \left(2M_{4p} V_{4p}(x) + 2M_{4p}^2 V_{4p}(x)\right)^{\frac{1}{2}}.$$

By injecting the previous moment bounds in equation (66), we get

$$\|\delta_x R_{\gamma,1}^{(k-j-1)m} R_{\gamma,1}^m - \delta_x R_{\gamma,1}^{(k-j-1)m} R_{\gamma,2}^m\|_{V_{2p}}$$
$$\leq A_1(x) \left(\int_0^{m\gamma} \mathbb{E}\left(\|b^1(x_{t,j}^1, t) - b^2(x_{t,j}^1, t)\|^2\right) dt\right)^{\frac{1}{2}}$$
$$= A_1(x) \left(\gamma \sum_{l=0}^{m-1} \mathbb{E}\left(\|b^1(x_{l\gamma,j}^1) - b^2(x_{l\gamma,j}^1)\|^2\right)\right)^{\frac{1}{2}},$$

with $A_1(x) = \left(2M_{4p}V_{4p}(x) + 2M_{4p}^2 V_{4p}(x)\right)^{\frac{1}{2}}$ a constant independent of $j$ and $k$. Moreover, $x_{l\gamma,j}^1$ has the same law that $X_{(k-j-1)m+l}^1$.

Hence we get that

$$\|\delta_x R_{\gamma,1}^{(k-j-1)m} R_{\gamma,1}^m - \delta_x R_{\gamma,1}^{(k-j-1)m} R_{\gamma,2}^m\|_{V_{2p}}$$
$$\leq A_1(x)\left(\gamma \sum_{l=0}^{m-1} \mathbb{E}\left(\|b^1(X_{(k-j-1)m+l}^1) - b^2(X_{(k-j-1)m+l}^1)\|^2\right)\right)^{\frac{1}{2}}, \tag{67}$$

**From bounded time $m\gamma$ to any finite time $k\gamma$ with ergodicity**

We will massively use in this paragraph the ergodicity of the discrete-time Markov Chain to deduce from the control at time $m\gamma$ in equation (67) a control for any time $k$ on $\|\delta_x R_{\gamma,1}^{km} - \delta_x R_{\gamma,2}^{km}\|_{V_p}$.

We define $Y_k$ the Markov Chain following equation (47) and starting with the invariant law $\pi_\gamma^1$ at time $k = 0$. Then the law of the Markov Chain $Y_k$ is $\pi_\gamma^1$ at each time $k \geq 0$. Next, by Assumption 1, $b^1$ and $b^2$ are $L$-Lipschitz and we have

$$\|b^1(x) - b^2(x)\|^2 \leq \left(\|b^1(x)\| + \|b^2(x)\|\right)^2 \leq \left(\|b^1(0)\| + L\|x\| + \|b^2(0)\| + L\|x\|\right)^2$$
$$\leq 2\left(\|b^1(0)\| + \|b^2(0)\|\right)^2 + 2L^2\|x\|^2$$
$$\leq 2\max\left(\left(\|b^1(0)\| + \|b^2(0)\|\right)^2, L^2\right)\left(1 + \|x\|^2\right).$$

Thus, we have that

$$\phi(x) := \frac{\|b^1(x) - b^2(x)\|^2}{\phi_0} \leq V_2(x) = 1 + \|x\|^2,$$

with $\phi_0 = 2\max\left(\left(\|b^1(0)\| + \|b^2(0)\|\right)^2, L^2\right)$. By definition of the $V_2$-norm (equation (43)), we get that

$$\mathbb{E}(\phi(X_{(k-j-1)m+l}^1)) - \mathbb{E}(\phi(Y_k)) \leq \|\delta_x R_{\gamma,1}^{(k-j-1)m+l} - \pi_\gamma^1\|_{V_2}.$$

The previous equation can be reformulate into

$$\mathbb{E}\left(\|b^1(X_{(k-j-1)m+l}^1) - b^2(X_{(k-j-1)m+l}^1)\|^2\right) - \mathbb{E}\left(\|b^1(Y_k) - b^2(Y_k)\|^2\right)$$
$$\leq \phi_0 \|\delta_x R_{\gamma,1}^{(k-j-1)m+l} - \pi_\gamma^1\|_{V_2} \tag{68}$$

However by Lemma 19, we have that

$$\|\delta_x R_{\gamma,1}^{(k-j-1)m+l} - \pi_\gamma^1\|_{V_2} \leq \sqrt{\delta_x R_{\gamma,1}^{(k-j-1)m+l}(V_2) + \pi_\gamma^1(V_2)}\|R_{\gamma,1}^{(k-j-1)m+l} - \pi_\gamma^1\|_{TV}^{\frac{1}{2}}.$$

By injecting in the previous equation the moment bound in equation (53), we get

$$\|\delta_x R_{\gamma,1}^{(k-j-1)m+l} - \pi_\gamma^1\|_{V_2} \leq \sqrt{2M_2 V_2(x)}\|R_{\gamma,1}^{(k-j-1)m+l} - \pi_\gamma^1\|_{TV}^{\frac{1}{2}}. \tag{69}$$

By [23, Corollary 2], under Assumption 1, there exist $A_2(x) \geq 0$ and $\tilde{\rho} \in (0,1)$ such that $\forall k \geq 0$, $\forall i \in \{1, 2\}$

$$\|\delta_x R_{\gamma,i}^k - \pi_\gamma^i\|_{TV} \leq A_2(x)\tilde{\rho}^{k\gamma}.$$

Then, by combining the previous equation with equation (68) and equation (69), we get

$$\mathbb{E}\left(\|b^1(X_{(k-j-1)m+l}^1) - b^2(X_{(k-j-1)m+l}^1)\|^2\right)$$
$$\leq \mathbb{E}\left(\|b^1(Y_k) - b^2(Y_k)\|^2\right) + \phi_0\sqrt{2M_2 V_2(x)A_2(x)}\tilde{\rho}^{\frac{((k-j-1)m+l)\gamma}{2}} \tag{70}$$

By summing the previous inequality for $l \in [0, m-1]$, we get

$$\sum_{l=0}^{m-1} \mathbb{E} \left( \|b^1(X^1_{(k-j-1)m+l}) - b^2(X^1_{(k-j-1)m+l})\|^2 \right)$$

$$\leq \phi_0 \sqrt{2M_2 V_2(x) A_2(x)} \sum_{l=0}^{m-1} \tilde{\rho}^{\frac{((k-j-1)m+l)\gamma}{2}} + m \mathbb{E}_{Y \sim \pi^1_\gamma} \left( \|b^1(Y) - b^2(Y)\|^2 \right)$$

$$\leq A_3(x) \tilde{\rho}^{\frac{(k-j-1)m\gamma}{2}} + m \mathbb{E}_{Y \sim \pi^1_\gamma} \left( \|b^1(Y) - b^2(Y)\|^2 \right),$$

with $A_3(x) = \phi_0 \sqrt{2M_2 V_2(x) A_2(x)} \frac{1}{1 - \tilde{\rho}^{\frac{\gamma}{2}}}$ a constant independent of $k$ and $j$. By injecting the previous equation in equation (67) and given that $\gamma m \leq 1$ thanks to $m$ definition in equation (50), we get

$$\|\delta_x R^{(k-j-1)m}_{\gamma,1} R^m_{\gamma,1} - \delta_x R^{(k-j-1)m}_{\gamma,1} R^m_{\gamma,2}\|_{V_{2p}}$$

$$\leq A_1(x) \left( A_3(x) \gamma \tilde{\rho}^{\frac{(k-j-1)m\gamma}{2}} + \gamma m \mathbb{E}_{Y \sim \pi^1_\gamma} \left( \|b^1(Y) - b^2(Y)\|^2 \right) \right)^{\frac{1}{2}}$$

$$\leq A_1(x) A_3^{\frac{1}{2}}(x) \gamma^{\frac{1}{2}} \tilde{\rho}^{\frac{(k-j-1)m\gamma}{4}} + A_1(x) \left( \mathbb{E}_{Y \sim \pi^1_\gamma} \left( \|b^1(Y) - b^2(Y)\|^2 \right) \right)^{\frac{1}{2}}.$$

We now combine the previous inequality with equation (62) to obtain

$$\|\delta_x R^{km}_{\gamma,1} - \delta_x R^{km}_{\gamma,2}\|_{V_p}$$

$$\leq 2B A_1(x) \sum_{j=0}^{k-1} \rho_p^{jm\gamma} \left( A_3^{\frac{1}{2}}(x) \gamma^{\frac{1}{2}} \tilde{\rho}^{\frac{(k-j-1)m\gamma}{4}} + \left( \mathbb{E}_{Y \sim \pi^1_\gamma} \left( \|b^1(Y) - b^2(Y)\|^2 \right) \right)^{\frac{1}{2}} \right)$$

$$\leq 2B A_1(x) A_3^{\frac{1}{2}}(x) \gamma^{\frac{1}{2}} \frac{1}{\left| 1 - \frac{\rho_p^{m\gamma}}{\tilde{\rho}^{\frac{m\gamma}{4}}} \right|} \tilde{\rho}^{\frac{(k-1)m\gamma}{4}}$$

$$+ 2B A_1(x) \frac{1}{1 - \rho_p^{m\gamma}} \left( \mathbb{E}_{Y \sim \pi^1_\gamma} \left( \|b^1(Y) - b^2(Y)\|^2 \right) \right)^{\frac{1}{2}},$$

where, at the cost of slightly increasing $\tilde{\rho}$, we assume that $\frac{\rho_p^{m\gamma}}{\tilde{\rho}^{\frac{m\gamma}{4}}} \neq 1$. Denoting $A_4(x) = 2B A_1(x) A_3^{\frac{1}{2}}(x) \gamma^{\frac{1}{2}} \frac{\tilde{\rho}^{-\frac{m\gamma}{4}}}{\left| 1 - \frac{\rho_p^{m\gamma}}{\tilde{\rho}^{\frac{m\gamma}{4}}} \right|}$ and $A_5(x) = 2B A_1(x) \frac{1}{1 - \rho_p^{m\gamma}}$, we get

$$\|\delta_x R^{km}_{\gamma,1} - \delta_x R^{km}_{\gamma,2}\|_{V_p} \leq A_4(x) \tilde{\rho}^{\frac{km\gamma}{4}} + A_5(x) \left( \mathbb{E}_{Y \sim \pi^1_\gamma} \left( \|b^1(Y) - b^2(Y)\|^2 \right) \right)^{\frac{1}{2}}. \tag{71}$$

**Putting all together and asymptotic computations**

We combine equation (51), equation (57) and equation (71) to obtain

$$\|\pi^1_\gamma - \pi^2_\gamma\|_{V_p} \leq A_4(x) \tilde{\rho}^{\frac{km\gamma}{4}} + A_5(x) \left( \mathbb{E}_{Y \sim \pi^1_\gamma} \left( \|b^1(Y) - b^2(Y)\|^2 \right) \right)^{\frac{1}{2}}$$

$$+ D_p(x) \rho_p^{km\gamma}.$$

By taking $k \to +\infty$, we get

$$\|\pi^1_\gamma - \pi^2_\gamma\|_{V_p} \leq A_5(x) \left( \mathbb{E}_{Y \sim \pi^1_\gamma} \left( \|b^1(Y) - b^2(Y)\|^2 \right) \right)^{\frac{1}{2}},$$

with $A_5(x) = 2B A_1(x) \frac{1}{1 - \rho_p^{m\gamma}}$, with $m\gamma \geq 1 - \gamma \geq 1 - \gamma_0$, since we defined $m = \left\lfloor \frac{1}{\gamma} \right\rfloor$ in relation (50) and we supposed that $\gamma \in (0, \gamma_0)$ in the theorem's assumptions. Note that $x \in \mathbb{R}^d$ is free. We choose $x = 0$, then $B = D_p(1 + \pi^2_\gamma(V^2_p))$ which is defined after equation (59) and $A_1(0) = \left( 2M_{4p} + 2M^2_{4p} \right)^{\frac{1}{2}}$ which is defined before equation (67). By taking $k \to +\infty$ in equation (54), we

get that $\forall x \in \mathbb{R}^d$, $\pi_\gamma^2(V_p^2) \leq 2M_{2p}V_{2p}(x)$. In particular for $x = 0$, we get $\pi_\gamma^2(V_p^2) \leq 2M_{2p}$, thus $B \leq D_p(1 + 2M_{2p})$.

This proves Theorem 1 in $p$-Wasserstein distance with the constant

$$G_p = \frac{2}{1 - \rho_p^{1-\gamma_0}} D_p(1 + 2M_{2p}) \left(2M_{4p} + 2M_{4p}^2\right)^{\frac{1}{2}},$$

where the constants $D_p$ and $\rho_p$ are defined by the geometric convergence in equation (55), $M_p$ by the moment bound in (53), $\gamma_0 = \frac{m}{L^2}$ with $m, L$ defined in Assumption 1 and $V_p = 1 + \|\cdot\|^p$. $\quad\square$

### F.4 Proof of Theorem 1

In this part, we prove Theorem 1. First, we recall its statement.

**Theorem 6.** *Let $b^1$ and $b^2$ satisfy Assumption 1. $X_k^1$ and $X_k^2$ are two geometrically ergodic Markov Chains with invariant laws $\pi_\gamma^1, \pi_\gamma^2$. Then for $\gamma_0 = \frac{m}{L^2}$ and $p \in \mathbb{N}^\star$ there exist $C_p, C \geq 0$ such that $\forall \gamma \in (0, \gamma_0]$, we have*

$$\mathbf{W}_p(\pi_\gamma^1, \pi_\gamma^2) \leq C_p \|b^1 - b^2\|_{\ell_2(\pi_\gamma^1)}^{\frac{1}{p}}$$

$$\|\pi_\gamma^1 - \pi_\gamma^2\|_{TV} \leq \tilde{C} \|b^1 - b^2\|_{\ell_2(\pi_\gamma^1)}.$$

Theorem 1 is demonstrated for the $\mathbf{W}_p$ and $TV$ metrics. In [29, 68], the authors analyse the convergence of the Langevin dynamics using the Kullback-Leibler divergence. However, because our approach depends heavily on the triangle inequality property of the metric, it does not readily extend to the Kullback-Leibler divergence, which does not satisfy this inequality.

*Proof.* By Theorem 5, under Assumption 1, we have that $X_k^1$ and $X_k^2$ are geometrically ergodic, with invariant laws $\pi_\gamma^1, \pi_\gamma^2$. Moreover, with $\gamma_0 = \frac{m}{L^2}$, for $p \in \mathbb{N}^\star$, there exists $G_p \geq 0$ such that $\forall \gamma \in (0, \gamma_0]$, we have

$$\|\pi_\gamma^1 - \pi_\gamma^2\|_{V_p} \leq G_p \left(\mathbb{E}_{Y \sim \pi_\gamma^1}\left(\|b^1(Y) - b^2(Y)\|^2\right)\right)^{\frac{1}{2}},$$

with $V_p = 1 + \|\cdot\|^p$.

However, by definition of the $V$-norm in equation (43), we have

$$\|\pi_\gamma^1 - \pi_\gamma^2\|_{TV} \leq \|\pi_\gamma^1 - \pi_\gamma^2\|_{V_1} \leq G_1 \left(\mathbb{E}_{Y \sim \pi_\gamma^1}\left(\|b^1(Y) - b^2(Y)\|^2\right)\right)^{\frac{1}{2}},$$

which proves the first part of the Theorem relative to the $TV$-distance.

Next, by Lemma 18, we have for $p \in \mathbb{N}^\star$

$$\mathbf{W}_p\left(\pi_\gamma^1, \pi_\gamma^2\right) \leq 2^{1-\frac{1}{p}} \|\pi_\gamma^1 - \pi_\gamma^2\|_{V_p}^{\frac{1}{p}}$$

$$\leq 2^{1-\frac{1}{p}} G_p^{\frac{1}{p}} \left(\mathbb{E}_{Y \sim \pi_\gamma^1}\left(\|b^1(Y) - b^2(Y)\|^2\right)\right)^{\frac{1}{2p}},$$

which proves the second part for the $\mathbf{W}_p$ distance. $\quad\square$

### F.5 On the constant $C_p, C$ of Theorem 1

Note that $C = C_1$, so we only need to study $C_p$ for $p \in \mathbb{N}$. The constant $C_p$ of Theorem 1 is defined by

$$C_p = 2^{1-\frac{1}{p}} \left(\frac{2}{1 - \rho_p^{1-\gamma_0}} D_p(1 + 2M_{2p}) \left(2M_{4p} + 2M_{4p}^2\right)^{\frac{1}{2}}\right)^{\frac{1}{p}}, \tag{72}$$

where the constants $D_p$ and $\rho_p$ are defined by the geometric convergence in equation (55), $M_p$ by the moment bound in (53), $\gamma_0 = \frac{m}{L^2}$ with $m, L$ defined in Assumption 1 and $V_p = 1 + \|\cdot\|^p$.

$C_p$ is independent of the step-size $\gamma$ and of the drifts $b^1, b^2$.

The authors of [23] proved that the constant $\rho_p$ is independent of the dimension $d$ and that $D_p$ is polynomial in the dimension. Therefore, $C_p$ has a polynomial dependence in the dimension $d$ with a dependence on the moment of the target law.

The dependency on $p$ is unclear as it depends highly on the evolution of the moment bounds $M_{2p}$ with $p$. Therefore, the dependence of $C_p$ in $p \in \mathbb{N}$ is unknown in general.

The constant $D_p$ dependent of the non-convexity radius by a factor of $\sqrt{\frac{1}{R}}$ for $R \leq 1$ according to [23, Corollary 2]. $\rho_p$ is independent of $R$. Therefore the constant $C_p$ explode when $R \to 0$ with a dependency of $\left(\frac{1}{R}\right)^{\frac{1}{2p}}$. Based on [23, equation (10)], we have that the dependency is exponential in $R$ when $R \to +\infty$ with $C_p = \mathcal{O}(e^{\frac{LR^2}{4}})$.

### F.6 Quantification of the discretization Error - Proof of Theorem 2

Based on the proof of Theorem 1 we can recover the discretization error for discrete Unadjusted Langevin Algorithm (ULA). For that purpose, we introduce the ULA Markov Chain

$$X_{k+1} = X_k - \gamma \nabla V(X_k) + \sqrt{2\gamma} Z_{k+1}, \tag{73}$$

with $Z_{k+1}$ a normalized Gaussian random variable, as well as the continuous Langevin Markov Process

$$dx_t^1 = -\gamma \nabla V(x_t^1) + \sqrt{2} dw_t^1, \tag{74}$$

with $dw_t^1$ a Wiener process. Under Assumption 1, Theorem 1 shows that $X_k$ has an invariant law $\pi_\gamma$ and is geometrically ergodic. Moreover, if $-\nabla V$ verifies Assumption 1, [23, Corollary 22] proves that $x_t^1$ has $\pi \propto e^{-V}$ as an invariant law and is geometrically ergodic.

**Theorem 7.** *If the function $\nabla V$ verifies Assumption 1 and $\pi \propto e^{-V}$. Then, for $\gamma_0 = \frac{m}{L^2}$, for $p \in \mathbb{N}^\star$ there exist $C_p, D_p, \tilde{C} \geq 0$ such that $\forall \gamma \in (0, \gamma_0]$, the Markov Chain is geometrically ergodic with an invariant law $\pi_\gamma$ and the discretization error is bounded by*

$$\mathbf{W}_p(\pi_\gamma, \pi) \leq C_p \gamma^{\frac{1}{2p}}$$
$$\|\pi_\gamma - \pi\|_{TV} \leq \tilde{C} \gamma^{\frac{1}{2}}$$
$$\|\pi_\gamma - \pi\|_{V_p} \leq D_p \gamma^{\frac{1}{2}},$$

*with $\pi = e^{-V}$ and $V_p = 1 + \|\cdot\|^p$.*

*Proof.* This proof is based on the same reasoning that the proof of Theorem 5 with continuous Langevin processes instead of discrete-time Markov Chain. The key point is to write the discrete-time ULA Markov Chain as a continuous Langevin process in equation (75). Then, we verify that the step of the reasoning of the proof of Theorem 5 are still true in this continuous context.

We define the ULA continuized continuous process by

$$dx_t^2 = -b_\gamma((x_t^2)_{t \in \mathbb{R}_+}, t)dt + \sqrt{2} dw_t^2, \tag{75}$$

with $b_\gamma((x_t)_{t \in \mathbb{R}^+}, t) = \sum_{k=0}^{+\infty} \mathbb{1}_{[k\gamma, (k+1)\gamma)}(t) \nabla V(x_{k\gamma}) = \nabla V(x_{\lfloor \frac{t}{\gamma} \rfloor \gamma})$. Following the same reasoning as the one given after equation (63), the process $x_t^2$ is a continuous counterpart of the Markov Chain (73). More precisely, $x_{k\gamma}^2$ initialized with the probability law $\delta_{X_0}$ has the same law than $X_k$, the ULA Markov Chain defined in equation (73).

We denote by $P_t$ the semi-group (see definition in equation (39)) of the stochastic equation (74) and $R_\gamma$ the Markov kernel (see definition in equation (42)) of the Markov Chain defined in equation (73).

We follow a similar sketch of the proof of Theorem 5. We first demonstrate a relation similar to equation (56), by replacing [23, Corollary 2] with [23, Corollary 2] for the process $x_t^1$. We thus obtain that for $x \in \mathbb{R}^d$ and $p \in \mathbb{N}^\star$, there exist $A_p \geq 0$ and $\rho_p \in (0, 1)$ such that, for $\mu$ a probability

distribution on $\mathbb{R}^d$, we have

$$\|\pi_\gamma - \mu R_\gamma^{km}\|_{V_p} \le A_p \mu(V_p^2) \rho_p^{km\gamma}$$
$$\|\mu P_{km\gamma} - \pi\|_{V_p} \le A_p \mu(V_p^2) \rho_p^{km\gamma}. \tag{76}$$

For $x \in \mathbb{R}^d$ and $k \in \mathbb{N}$, $m = \lfloor \frac{1}{\gamma} \rfloor$, by the triangle inequality and equation (76) applied for $\mu = \delta_x$, we have

$$\|\pi_\gamma - \pi\|_{V_p}$$
$$\le \|\pi_\gamma - \delta_x R_\gamma^{km}\|_{V_p} + \|\delta_x R_\gamma^{km} - \delta_x P_{km\gamma}\|_{V_p} + \|\delta_x P_{km\gamma} - \pi\|_{V_p}$$
$$\le 2A_p V_p^2(x) \rho_p^{km\gamma} + \|\delta_x R_\gamma^{km} - \delta_x P_{km\gamma}\|_{V_p}$$
$$\le 2A_p V_p^2(x) \rho_p^{km\gamma} + \sum_{j=0}^{k-1} \|\delta_x R_\gamma^{(k-j-1)m} R_\gamma^m P_{jm\gamma} - \delta_x R_\gamma^{(k-j-1)m} P_{m\gamma} P_{jm\gamma}\|_{V_p}.$$

Similarly to the reasoning that led us to equation (61), we get

$$\|\delta_x R_\gamma^{(k-j-1)m} R_\gamma^m P_{jm\gamma} - \delta_x R_\gamma^{(k-j-1)m} P_{m\gamma} P_{jm\gamma}\|_{V_p}$$
$$\le 2A_p \rho_p^{jm\gamma} \|\delta_x R_\gamma^{(k-j-1)m} R_\gamma^m - \delta_x R_\gamma^{(k-j-1)m} P_{m\gamma}\|_{V_{2p}}.$$

By combining the two previous equations, we obtain

$$\|\pi_\gamma - \pi\|_{V_p}$$
$$\le 2A_p V_p^2(x) \rho_p^{km\gamma} + 2A_p \sum_{j=0}^{k-1} \rho_p^{jm\gamma} \|\delta_x R_\gamma^{(k-j-1)m} R_\gamma^m - \delta_x R_\gamma^{(k-j-1)m} P_{m\gamma}\|_{V_{2p}}. \tag{77}$$

Then, by Lemma 20 and the fact that the drift of $x_t^1$ is $\nabla V$ and the drift of $x_t^2$ is $b_\gamma$, we get

$$\|\delta_x R_\gamma^{(k-j-1)m} R_\gamma^m - \delta_x R_\gamma^{(k-j-1)m} P_{m\gamma}\|_{V_{2p}}$$
$$\le \left( \delta_x R_\gamma^{(k-j-1)m}(V_{2p}^2) + \delta_x R_\gamma^{(k-j-1)m}(V_{2p}^2) \right)^{\frac{1}{2}}$$
$$\times \left( \int_0^{m\gamma} \mathbb{E} \left( \|\nabla V(y_{t,j}) - b_\gamma((y_{t,j})_{t \in \mathbb{R}_+}, t)\|^2 \right) dt \right)^{\frac{1}{2}},$$

with $b_\gamma$ defined in equation (75) and $y_{t,j}$, the continuous process starting with the law $\delta_x R_\gamma^{(k-j-1)m}$ and following equation (74).

Using the moment bound given by [59, Lemma 17], we get that there exists $F_p(x) \ge 0$ such that $\forall k \in \mathbb{N}^\star, j \in \{0, \dots, k-1\}$,

$$\left( \delta_x R_\gamma^{(k-j-1)m}(V_{2p}^2) + \delta_x R_\gamma^{(k-j-1)m}(V_{2p}^2) \right)^{\frac{1}{2}} \le F_p(x).$$

Combining previous equations thus gives

$$\|\delta_x R_\gamma^{(k-j-1)m} R_\gamma^m - \delta_x R_\gamma^{(k-j-1)m} P_{m\gamma}\|_{V_{2p}} \tag{78}$$
$$\le F_p(x) \left( \int_0^{m\gamma} \mathbb{E} \left( \|\nabla V(y_{t,j}) - b_\gamma((y_{t,j})_{t \in \mathbb{R}_+}, t)\|^2 \right) dt \right)^{\frac{1}{2}}. \tag{79}$$

Since $\nabla V$ and $b_\gamma$ are $L$-Lipschitz from Assumption 1, we get, with $m = \lfloor \frac{1}{\gamma} \rfloor$, that

$$\int_0^{m\gamma} \mathbb{E} \left( \|\nabla V(y_{t,j}) - b_\gamma((y_{t,j})_{t \in \mathbb{R}_+}, t)\|^2 \right) dt$$
$$\le L \left( \sum_{k=0}^{m-1} \int_{k\gamma}^{(k+1)\gamma} \mathbb{E} \left( \|y_{t,j} - y_{k\gamma,j}\|^2 \right) dt \right) \tag{80}$$

Because, $y_{t,j}$ is following equation (74), we have $y_{t,j} - y_{k\gamma,j} = \int_{k\gamma}^t -\nabla V(y_{u,j})du + \sqrt{2}\int_{k\gamma}^t dw_u^1$. Combined with the Itô isometry, the fact that $\|\nabla V(x)\| \leq \|\nabla V(0)\| + L\|x\|$, the Cauchy-Schwarz inequality, the moment bound in equation (53) and the fact that $t - k\gamma \leq \gamma$, we get

$$\mathbb{E}\left(\|y_{t,j} - y_{k\gamma,j}\|^2\right) = \mathbb{E}\left(\|\int_{k\gamma}^t -\nabla V(y_{u,j})du + \sqrt{2}\int_{k\gamma}^t dw_u^1\|^2\right) \tag{81}$$

$$\leq 2\mathbb{E}\left(\|\int_{k\gamma}^t -\nabla V(y_{u,j})du\|^2\right) + 4\mathbb{E}\left(\|\int_{k\gamma}^t dw_u^1\|^2\right)$$

$$\leq 2\mathbb{E}\left(\left(\gamma\|\nabla V(0)\| + L\int_{k\gamma}^t \|y_{u,j}\|du\right)^2\right) + 4\gamma d$$

$$\leq 4\gamma^2\|\nabla V(0)\|^2 + 4L^2\mathbb{E}\left(\left(\int_{k\gamma}^t \|y_{u,j}\|du\right)^2\right) + 4\gamma d$$

$$\leq 4\gamma^2\|\nabla V(0)\|^2 + 4L^2\gamma\int_{k\gamma}^t \mathbb{E}\left(\|y_{u,j}\|^2\right)du + 4\gamma d$$

$$\leq 4\gamma^2\|\nabla V(0)\|^2 + 4L^2\gamma^2 M_2 V_2(x) + 4\gamma d$$

$$\leq \left(4\gamma_0\|\nabla V(0)\|^2 + 4L^2\gamma_0 M_2 V_2(x) + 4d\right)\gamma. \tag{82}$$

Combining equations (82), (80) and (79), with $B_6 = 4\gamma_0\|\nabla V(0)\|^2 + 4L^2\gamma_0 M_2 V_2(x) + 4d$ and the fact that $m\gamma \leq 1$, we obtain

$$\|\delta_x R_\gamma^{(k-j-1)m} R_\gamma^m - \delta_x R_\gamma^{(k-j-1)m} P_{m\gamma}\|_{V_{2p}} \leq F_p(x) L^{\frac{1}{2}} (B_6 m\gamma)^{\frac{1}{2}} \gamma^{\frac{1}{2}} \leq F_p(x) L^{\frac{1}{2}} (B_6)^{\frac{1}{2}} \gamma^{\frac{1}{2}},$$

since $m = \lfloor\frac{1}{\gamma}\rfloor$ so that $m\gamma \leq 1$. By injecting the previous inequality into the equation (77), we get

$$\|\pi_\gamma - \pi\|_{V_p} \leq 2A_p V_p^2(x)\rho_p^{km\gamma} + 2A_p \sum_{j=0}^{k-1} \rho_p^{jm\gamma} F_p(x)(LB_6)^{\frac{1}{2}}\gamma^{\frac{1}{2}}$$

$$\leq 2A_p V_p^2(x)\rho_p^{km\gamma} + 2A_p \frac{1}{1-\rho_p^{m\gamma}}(LB_6)^{\frac{1}{2}}\gamma^{\frac{1}{2}}.$$

Taking $k \to +\infty$ in the previous inequality, we get the desired result in $V_p$-norm. We get the result in $\mathbf{W}_p$ and $TV$ norms following the same reasoning as in the proof of Theorem 1 in section F.4. $\quad\square$

# G    Proofs for PSGLA

In this part, we provide proofs for the analysis of the PSGLA algorithm defined in equation (3). First, we demonstrate regularity results on the drift $b^\gamma$ defined in equation (11) , then based on our IULA Markov Chain (definition in equation (2)) analysis of section 2, we derive the convergence theory for the PSGLA algorithm.

### G.1    The drift of PSGLA is verifying Assumption 1

**Lemma 21.** *Under Assumptions 2-3, for all $\gamma \in \left(0, \min\left(\frac{1}{2L_g}, \frac{1}{\rho}, \gamma_1\right)\right]$, $b^\gamma$ verifies*

    *(i) $b^\gamma$ is L-smooth on $\mathbb{R}^d$, with $L = 2L_f + 2L_g$.*

    *(ii) $\forall x, y \in \mathbb{R}^d$ such that $\|x - y\| \geq 4R_0$, we have*

$$\langle b(x) - b(y), x - y\rangle \geq \frac{\mu}{4}\|x - y\|^2,$$

*with the constants $L_f, L_g, \rho, \gamma_1, R_0$ and $\mu$ defined in Assumptions 2-3.*

*Then, for $\gamma \in \left(0, \min\left(\frac{1}{2L_g}, \frac{1}{\rho}, \gamma_1\right)\right]$, $b^\gamma$ verifies Assumption 1 with $L = 2L_f + 2L_g$, $R = 4R_0$ and $m = \frac{\mu}{4}$.*

Note that the constant that appears in Lemma 21 are independent of $\gamma$.

*Proof.* We will demonstrate each point separately for the drift $b^\gamma(x) = \nabla f(x - \gamma \nabla g^\gamma(x)) + \nabla g^\gamma(x)$ defined in equation (11). We need that $\gamma \leq \frac{1}{\rho}$ for the Moreau envelope $g^\gamma$ of $g$ to be well defined.

**(i)** Under Assumption 3(i), Lemma 15 of Appendix D.7 gives that for $\gamma \leq \frac{1}{2L_g}$, $\nabla g^\gamma$ is $2L_g$-Lipschitz. Combined with Assumption 2(i), we get, for $x, y \in \mathbb{R}^d$

$$
\begin{aligned}
&\|b^\gamma(x) - b^\gamma(y)\| \\
&\leq \|\nabla f(x - \gamma \nabla g^\gamma(x)) - \nabla f(y - \gamma \nabla g^\gamma(y))\| + \|\nabla g^\gamma(x) - \nabla g^\gamma(y)\| \\
&\leq L_f \|x - y - \gamma \nabla g^\gamma(x) + \gamma \nabla g^\gamma(y)\| + 2L_g \|x - y\| \\
&\leq L_f \left(\|x - y\| + \gamma 2L_g \|x - y\|\right) + 2L_g \|x - y\| \\
&\leq (2L_f + 2L_g) \|x - y\|,
\end{aligned}
$$

which proves point (i) of Lemma 21.

**(ii)** We want to have a lower bound of the quantity $\langle b^\gamma(x) - b^\gamma(y), x - y \rangle$, for $x, y \in \mathbb{R}^d$, when $\|x - y\|$ is sufficiently large.

For $0 < \gamma \leq \frac{1}{2L_g}$ and $x, y \in \mathbb{R}^d$, we have

$$
\langle b^\gamma(x) - b^\gamma(y), x - y \rangle \geq -2L_f \|x - y\|^2 + \langle \nabla g^\gamma(x) - \nabla g^\gamma(y), x - y \rangle,
$$

where we used that $\nabla f(x - \gamma \nabla g^\gamma(x))$ if $2L_f$-Lipschitz thanks to the computation of point (i).

We suppose that $\|x - y\| \geq 4R_0$, then there is at most one point between $x$ and $y$ that belongs to the ball $B(0, R_0)$.

We distinguish three cases.

- $[x, y] \cap B(0, R_0) = \emptyset$. From Assumption 3(ii), if $\gamma \leq \gamma_1$, $\nabla^2 g^\gamma \succeq \mu I_d$ on $\mathbb{R}^d \setminus B(0, R_0)$. Then we immediately have

$$
\langle b^\gamma(x) - b^\gamma(y), x - y \rangle \geq (\mu - 2L_f) \|x - y\|^2.
$$

- If $x \in B(0, R_0)$, then $\|y\| = \|y - x + x\| \geq \|y - x\| - \|x\| \geq 3R_0$. We take $z \in [x, y]$, such that $\|z\| = R_0$, then $]z, y] \subset \mathbb{R}^d \setminus B(0, R_0)$. We define $t_0 \in [0, 1]$, such that $z = x + t_0(y - x)$. It holds that $\|z - x\| \leq 2R_0$ and $\|y - z\| = \|y - x + x - z\| \geq \|y - x\| - \|x - z\| \geq 2R_0$ and $t_0 = \frac{\|z - x\|}{\|y - x\|} \leq \frac{1}{2}$.

  Under Assumption 3(i), by Lemma 15 of Appendix D.7, $g^\gamma$ is $2L_g$-smooth, we get

$$
\begin{aligned}
\langle \nabla g^\gamma(y) - \nabla g^\gamma(x), y - x \rangle &= \int_0^1 \langle \nabla^2 g^\gamma(x + t(y - x))(y - x), y - x \rangle \mathrm{d}t \\
&= \int_0^{t_0} \langle \nabla^2 g^\gamma(x + t(y - x))(y - x), y - x \rangle \mathrm{d}t \\
&\quad + \int_{t_0}^1 \langle \nabla^2 g^\gamma(x + t(y - x))(y - x), y - x \rangle \mathrm{d}t \\
&\geq -2L_g t_0 \|y - x\|^2 + (1 - t_0)\mu \|y - x\|^2 \\
&\geq -L_g \|y - x\|^2 + \frac{\mu}{2} \|y - x\|^2,
\end{aligned}
$$

  and we deduce that

$$
\langle b^\gamma(x) - b^\gamma(y), x - y \rangle \geq \left(\frac{\mu}{2} - 2L_f - L_g\right) \|y - x\|^2.
$$

- $x, y \notin B(0, R_0)$, we look at the affine segment $t \in [0, 1] \mapsto x + t(y - x)$. If this segment does not intersect with $B(0, R_0)$, then we are in the first case. Thus we suppose that the

segment crosses the ball $B(0, R_0)$. We look at the norm of the vectors $u(\alpha)$ defined, for $\alpha \in \mathbb{R}$, along the affine line that passes through points $x$ and $y$, as

$$u(\alpha) = \|x + \alpha(y-x)\|^2 = \|x\|^2 + 2\alpha\langle x, y-x\rangle + \alpha^2 \|y-x\|^2.$$

Notice that $u$ is quadratic in $\alpha$. We assumed that $u(0), u(1) > R_0^2$ and there exists $t \in ]0, 1[$ such that $u(t) < R_0^2$, then there necessarily exist $0 < t_0 < t_1 < 1$, such that $u(t_0) = u(t_1) = R_0^2$. As a consequence, $\forall t \in (t_0, t_1)$, $u(t) < R_0^2$ and $\forall t \in [0, 1] \setminus [t_0, t_1]$, $u(t) > R_0^2$. We denote

$$z_0 = x + t_0(y-x)$$
$$z_1 = x + t_1(y-x).$$

By definition $z_0, z_1 \in B(0, R_0)$, therefore $\|z_0 - z_1\| \le 2R_0$. Then

$$t_1 - t_0 = \frac{\|z_1 - x\|}{\|y-x\|} - \frac{\|z_0 - x\|}{\|y-x\|} = \frac{\|z_1 - z_0\|}{\|y-x\|} \le \frac{2R_0}{\|y-x\|} \le \frac{1}{2}, \tag{83}$$

because $\|y - x\| \ge 4R_0$. We can now estimate the quantity of interest

$$
\begin{aligned}
\langle \nabla g^\gamma(y) - \nabla g^\gamma(x), y-x\rangle &= \int_0^1 \langle \nabla^2 g^\gamma(x + t(y-x))(y-x), y-x\rangle \mathrm{d}t \\
&= \int_0^{t_0} \langle \nabla^2 g^\gamma(x + t(y-x))(y-x), y-x\rangle \mathrm{d}t \\
&+ \int_{t_0}^{t_1} \langle \nabla^2 g^\gamma(x + t(y-x))(y-x), y-x\rangle \mathrm{d}t \\
&+ \int_{t_1}^1 \langle \nabla^2 g^\gamma(x + t(y-x))(y-x), y-x\rangle \mathrm{d}t \\
&\ge (1 + t_0 - t_1)\mu\|y-x\|^2 - 2(t_1 - t_0)L_g\|y-x\|^2 \\
&\ge (1 + t_0 - t_1)\mu\|y-x\|^2 - L_g\|y-x\|^2 \\
&\ge \frac{\mu}{2}\|y-x\|^2 - L_g\|y-x\|^2,
\end{aligned}
$$

where we used equation (83) in the last relation. So, we get

$$\langle b^\gamma(y) - b^\gamma(x), y-x\rangle \ge (\frac{\mu}{2} - 2L_f - L_g)\|y-x\|^2.$$

To conclude, in the three cases, for $\|y - x\| \ge 4R_0$ and by Assumption 3(ii) $8L_f + 4L_g \le \mu$, we get

$$\langle b(y) - b(x), y-x\rangle \ge (\frac{\mu}{2} - 2L_f - L_g)\|y-x\|^2.$$
$$\ge \frac{\mu}{4}\|y-x\|^2,$$

which proves the point (ii) of Lemma 21.

$\square$

### G.2   Proof of Theorem 3

First, we recall the statement of Theorem 3.

**Theorem 8.** *Under Assumptions 2-3, there exist $r \in (0, 1)$, $C_1, C_2 \in \mathbb{R}_+$ such that $\forall \gamma \in (0, \bar{\gamma}]$, with $\bar{\gamma} = \min\left(\frac{1}{2L_g}, \frac{\mu}{32(L_f + L_g)^2}, \frac{1}{2\rho}, \gamma_1\right)$, where $L_g, L_f, \rho, \gamma_1$ are defined in Assumptions 2-3, and $\forall k \in \mathbb{N}$, we have*

$$\mathbf{W}_p(p_{Y_k}, \mu_\gamma) \le C_1 r^{k\gamma} + C_2 \gamma^{\frac{1}{2p}}, \tag{84}$$

*with $p_{Y_k}$ the distribution of $Y_k$ and $\mu_\gamma \propto e^{-f-g^\gamma}$.*

*Moreover there exist $C_3, C_4 \in \mathbb{R}_+$ such that $\forall \gamma \in (0, \bar{\gamma}]$ and $\forall k \in \mathbb{N}$, we have*

$$\mathbf{W}_p(p_{X_k}, \nu_\gamma) \le C_3 r^{k\gamma} + C_4 \gamma^{\frac{1}{2p}}, \tag{85}$$

*with $p_{X_k}$ the distribution of $X_k$ and $\nu_\gamma \propto \mathsf{Prox}_{\gamma g}\#e^{-f-g^\gamma}$.*

*Proof.* Thanks to Lemma 21, under Assumptions 2-3, the drift of the Markov Chain $Y_k$ $b^\gamma$ verifies Assumption 1 with constants independent of $\gamma$. Therefore, using Theorem 5, we deduce that this Markov Chain $Y_k$ is geometrically ergodic, so there exist $C_1 \geq 0$, $r \in (0,1)$ and an invariant law $p_\infty$ such that

$$\mathbf{W}_1(p_{Y_k}, p_\infty) \leq C_1 r^{k\gamma}, \tag{86}$$

with $p_{Y_k}$ the distribution of $Y_k$.

Next we introduce the Markov Chain $\tilde{Y}_k$ such that $\tilde{Y}_0 = Y_0$ and

$$\tilde{Y}_{k+1} = \tilde{Y}_k + \gamma b_0(\tilde{Y}_k) + \sqrt{2\gamma}\tilde{Z}_{k+1},$$

with $b_0 = -\nabla g^\gamma - \nabla f$ and $\tilde{Z}_{k+1} \sim \mathcal{N}(0, I_d)$. Following the same proof than Lemma 21, we deduce that $b_0$ verifies Assumption 1 for $\gamma \leq \min\left(\frac{1}{\rho}, \frac{1}{2L_g}, \gamma_1\right)$ with the same parameters than $b^\gamma$, $L = 2L_f + 2L_g$, $R = 4R_0$ and $m = \frac{\mu}{4}$. Note that $L, R$ and $m$ are independent of $\gamma$. The parameters $L_f, L_g, R_0, \mu, \gamma_1, \rho$ are defined in Assumptions 2-3. Theorem 1 gives that for $\gamma_0 = \frac{\mu}{32(L_f+L_g)^2}$ and $\gamma \in (0, \gamma_0]$, $\tilde{Y}_k$ geometrically converges to its invariant law $p_\gamma$ and there exists $C_2 \geq 0$ such that

$$\mathbf{W}_p(p_\infty, p_\gamma) \leq C_2 \left(\mathbb{E}_{X\sim p_\infty}\left(\|b(X) - b_0(X)\|^2\right)\right)^{\frac{1}{2p}}.$$

We define $\bar{\gamma} = \min\left(\frac{1}{\rho}, \frac{1}{2L_g}, \gamma_1, \frac{\mu}{32(L_f+L_g)^2}\right)$.

Then, using the definition of $b$ in equation (11) and $b_0$, Assumption 1, Assumption 2(i), Assumption 3(i), the fact that $\nabla g^\gamma(x) = \nabla g(\mathsf{Prox}_{\gamma g}(x))$ (see Lemma 12 of Appendix D.6) and Lemma 11 of Appendix D.5, we get

$$\begin{aligned}
\mathbf{W}_p(p_\infty, p_\gamma) &\leq C_2 \left(\mathbb{E}_{X\sim p_\infty}\left(\|\nabla f(X - \gamma\nabla g^\gamma(X)) - \nabla f(X)\|^2\right)\right)^{\frac{1}{2p}} \\
&\leq C_2 L_f^{\frac{1}{p}} \gamma^{\frac{1}{p}} \left(\mathbb{E}_{X\sim p_\infty}\left(\|\nabla g^\gamma(X)\|^2\right)\right)^{\frac{1}{2p}} \\
&\leq C_2 L_f^{\frac{1}{p}} \gamma^{\frac{1}{p}} \left(\mathbb{E}_{X\sim p_\infty}\left(\|\nabla g(\mathsf{Prox}_{\gamma g}(x)\|^2\right)\right)^{\frac{1}{2p}} \\
&\leq C_2 L_f^{\frac{1}{p}} \gamma^{\frac{1}{p}} \left(2L_g^2 \mathbb{E}_{X\sim p_\infty}\left(\|\mathsf{Prox}_{\gamma g}(x) - \mathsf{Prox}_{\gamma g}(0)\|^2\right) + 2\|\mathsf{Prox}_{\gamma g}(x)\|^2\right)^{\frac{1}{2p}} \\
&\leq C_2 L_f^{\frac{1}{p}} \gamma^{\frac{1}{p}} \left(\frac{2L_g^2}{(1-\gamma\rho)^2}\mathbb{E}_{X\sim p_\infty}\left(\|X\|^2\right) + 2\|\mathsf{Prox}_{\gamma g}(0)\|^2\right)^{\frac{1}{2p}}.
\end{aligned}$$

By equation (53), we know that for $\gamma \leq \bar{\gamma}$, $\mathbb{E}_{X\sim p_\infty}\left(\|X\|^2\right)$ is bounded by a constant independent of $\gamma$. Then, for $\gamma \leq \bar{\gamma} \leq \frac{1}{2\rho}$, we have $\frac{1}{1-\gamma\rho} \leq 2$. Moreover $\mathsf{Prox}_{\gamma g}(0) \to 0$ when $\gamma \to 0$. So there exists a constant $C_3 \geq 0$, independent of $\gamma$, such that, for $\gamma \in (0, \bar{\gamma}]$

$$\mathbf{W}_1(p_\infty, p_\gamma) \leq C_3 \gamma^{\frac{1}{p}}. \tag{87}$$

Note that $b_0 = -\nabla(g^\gamma + f)$, so by Theorem 2, there exists $C_4 \geq 0$ such that $\forall \gamma \in (0, \bar{\gamma}]$ we have

$$\mathbf{W}_p(p_\gamma, \mu_\gamma) \leq C_4 \gamma^{\frac{1}{2p}}, \tag{88}$$

with $\mu_\gamma \propto e^{-f-g^\gamma}$.

Combining equations (86), (87) and (88) and the triangle inequality, we get

$$\begin{aligned}
\mathbf{W}_p(p_{Y_k}, \mu_\gamma) &\leq C_1 r^{k\gamma} + C_3 \gamma^{\frac{1}{p}} + C_4 \gamma^{\frac{1}{2p}} \\
&\leq C_1 r^{k\gamma} + C_5 \gamma^{\frac{1}{2p}},
\end{aligned}$$

with $C_5 = C_3 \bar{\gamma}^{\frac{1}{2p}} + C_4$. This proves the first part of Theorem 3.

Due to $X_k = \mathrm{Prox}_{\gamma g}(Y_k)$, the distribution $p_{X_k}$ of $X_k$ is $p_{X_k} = \mathrm{Prox}_{\gamma g} \# p_{Y_k}$, with $p_{Y_k}$ the distribution of $Y_k$ and "#" the push forward operator defined for a measurable set $A \subset \mathbb{R}^d$ by $p_{X_k}(A) = p_{Y_k}(\mathrm{Prox}_{\gamma g}^{-1}(A))$. With $\nu_\gamma = \mathrm{Prox}_{\gamma g} \# \mu_\gamma$, we get

$$\mathbf{W}_p(p_{X_k}, \nu_\gamma) = \min_{\beta \in \Lambda} \int_{\mathbb{R}^d \times \mathbb{R}^d} \|x_1 - x_2\|^p d\beta(x_1, x_2),$$

with $\Lambda$ the set of all possible transport plans $\beta$ between $p_{X_k}$ and $\nu_\gamma$. By the push-forward definition and Lemma 11 of Appendix D.5, we obtain

$$\mathbf{W}_p(p_{X_k}, \nu_\gamma) = \min_{\beta \in \Lambda} \int_{\mathbb{R}^d \times \mathbb{R}^d} \|\mathrm{Prox}_{\gamma g}(x_1) - \mathrm{Prox}_{\gamma g}(x_2)\|^p d\beta(x_1, x_2)$$

$$\leq \frac{1}{(1 - \gamma\rho)^{2p}} \min_{\beta \in \Lambda} \int_{\mathbb{R}^d \times \mathbb{R}^d} \|x_1 - x_2\| d\beta(x_1, x_2)$$

$$\leq \frac{1}{(1 - \bar{\gamma}\rho)^{2p}} \mathbf{W}_p(p_{Y_k}, \mu_\gamma).$$

The last equation demonstrates the second part of Theorem 3. $\qquad \square$

### G.3 On the exact law approximation with PSGLA

In this section, we provide a proof for Proposition 1. First, we recall this proposition

**Proposition 4.** *With* $\pi \propto e^{-f-g}$, $\mu_\gamma \propto e^{-f-g^\gamma}$ *and* $\nu_\gamma = \mathrm{Prox}_{\gamma g} \# \mu_\gamma$, *we have, for* $V : \mathbb{R}^d \to [1, +\infty)$ *and* $p \geq 1$

$$\lim_{\gamma \to 0} \|\mu_\gamma - \pi\|_V = 0 \tag{89}$$

$$\lim_{\gamma \to 0} \mathbf{W}_p(\mu_\gamma, \pi) = 0 \tag{90}$$

$$\lim_{\gamma \to 0} \mathbf{W}_p(\nu_\gamma, \pi) = 0 \tag{91}$$

*Moreover, if* $g$ *is* $L$-*Lipschitz, there exists* $E_p \in \mathbb{R}_+$ *such that* $\forall \gamma \in [0, \frac{2}{L^2}]$

$$\|\mu_\gamma - \pi\|_V \leq 2\pi(V)L\gamma \tag{92}$$

$$\mathbf{W}_p(\mu_\gamma, \pi) \leq E_p(L^2\gamma)^{\frac{1}{p}} \tag{93}$$

$$\mathbf{W}_p(\nu_\gamma, \pi) \leq E_p(L^2\gamma)^{\frac{1}{p}} + L\gamma. \tag{94}$$

*Proof.* This proof is a generalization of the strategy proposed in the proof of [28, Proposition 3.1]. By the definition of the $V$-norm in equation (43), we have

$$\|\mu_\gamma - \pi\|_V = \sup_{|\phi| \leq V} \int_{\mathbb{R}^d} \phi(x) \left(\mu_\gamma(x) - \pi(x)\right) dx,$$

with $\mu_\gamma(x) = \frac{e^{-f(x)-g^\gamma(x)}}{\int e^{-f-g^\gamma}}$ and $\pi(x) = \frac{e^{-f(x)-g(x)}}{\int e^{-f-g}}$. Then

$$\|\mu_\gamma - \pi\|_V \leq \sup_{|\phi| \leq V} \int_{\mathbb{R}^d} |\phi(x)| \, |\mu_\gamma(x) - \pi(x)| \, dx$$

$$\leq \int_{\mathbb{R}^d} V(x)\pi(x) \left| 1 - e^{g(x)-g^\gamma(x)} \frac{\int e^{-f-g}}{\int e^{-f-g^\gamma}} \right| dx$$

$$\leq \int_{\mathbb{R}^d} V(x)\pi(x) \left| 1 - e^{g(x)-g^\gamma(x)} \right| dx + \int_{\mathbb{R}^d} V(x)\pi(x)e^{g(x)-g^\gamma(x)} \left| 1 - \frac{\int e^{-f-g}}{\int e^{-f-g^\gamma}} \right| dx.$$

However, we know that $\forall x \in \mathbb{R}^d$, $g^\gamma(x) \leq g(x)$, so we get

$$\|\mu_\gamma - \pi\|_V$$

$$\leq \int_{\mathbb{R}^d} V(x)\pi(x) \left( e^{g(x)-g^\gamma(x)} - 1 \right) dx + \int_{\mathbb{R}^d} V(x)\pi(x)e^{g(x)-g^\gamma(x)} \left( 1 - \frac{\int e^{-f-g}}{\int e^{-f-g^\gamma}} \right) dx. \tag{95}$$

For $x \in \mathbb{R}^d$, $\lim_{\gamma \to 0} g^\gamma(x) = g(x)$ and the function $\gamma \to g^\gamma(x)$ is decreasing. So by the monotone convergence theorem, we have

$$\lim_{\gamma \to 0} \|\mu_\gamma - \pi\|_V = 0.$$

Lemma 18 proves that $\lim_{\gamma \to 0} \mathbf{W}_p(\mu_\gamma, \pi) = 0$. This concludes the first part of Proposition 1.

Next, by the triangle inequality, Lemma 11 of Appendix D.5 and taking the particular coupling $(I_d, \mathsf{Prox}_{\gamma g}) \# \pi$ between $\pi$ and $\mathsf{Prox}_{\gamma g} \# \pi$, we get

$$\mathbf{W}_p(\nu_\gamma, \pi) \leq \mathbf{W}_p(\mathsf{Prox}_{\gamma g} \# \mu_\gamma, \mathsf{Prox}_{\gamma g} \# \pi) + \mathbf{W}_p(\mathsf{Prox}_{\gamma g} \# \pi, \pi)$$

$$\leq \frac{1}{1 - \gamma \rho} \mathbf{W}_p(\mu_\gamma, \pi) + \left( \inf_\beta \int \|x_1 - x_2\|^p d\beta(x_1, x_2) \right)^{\frac{1}{p}}$$

$$\leq \frac{1}{1 - \gamma \rho} \mathbf{W}_p(\mu_\gamma, \pi) + \left( \int \|x - \mathsf{Prox}_{\gamma g}(x)\|^p d\pi(x) \right)^{\frac{1}{p}}.$$

By the definition of the Moreau envelope, we have $g(\mathsf{Prox}_{\gamma g}(x)) + \frac{1}{2\gamma} \|x - \mathsf{Prox}_{\gamma g}(x)\|^2 \leq g(x)$, so we get

$$\mathbf{W}_p(\nu_\gamma, \pi) \leq \frac{1}{1 - \gamma \rho} \mathbf{W}_p(\mu_\gamma, \pi) + \sqrt{2\gamma} \left( \int |g(x) - g(\mathsf{Prox}_{\gamma g}(x))|^{\frac{p}{2}} d\pi(x) \right)^{\frac{1}{p}}.$$

However, by Lemma 2 of Appendix D.2, we have that $|g(x) - g(\mathsf{Prox}_{\gamma g}(x))|$ has a monotone convergence to 0. By the monotone convergence theorem, we obtain

$$\lim_{\gamma \to 0} \mathbf{W}_p(\nu_\gamma, \pi) = 0.$$

If $g$ is $L$-Lipschitz, then, for $x \in \mathbb{R}^d$, we have

$$g(x) - g^\gamma(x) = g(x) - \inf_{y \in \mathbb{R}^d} \left( \frac{1}{2\gamma} \|x - y\|^2 + g(y) \right) = \sup_{y \in \mathbb{R}^d} g(x) - g(y) - \frac{1}{2\gamma} \|x - y\|^2$$

$$\leq \sup_{y \in \mathbb{R}^d} L\|x - y\| - \frac{1}{2\gamma} \|x - y\|^2 = \gamma \frac{L^2}{2}.$$

Since $\gamma(x) \geq g^\gamma(x)$ for all $x \in \mathbb{R}^d$, we get $\|g - g^\gamma\|_\infty \leq \gamma \frac{L^2}{2}$. By injecting this inequality into equation (95), we get

$$\|\mu_\gamma - \pi\|_V$$

$$\leq \int_{\mathbb{R}^d} V(x)\pi(x) \left( e^{\gamma \frac{L^2}{2}} - 1 \right) dx + \int_{\mathbb{R}^d} V(x)\pi(x) e^{\gamma \frac{L^2}{2}} \left( 1 - \frac{\int e^{-f-g}}{\int e^{-f-g^\gamma}} \right) dx.$$

Thanks to $\frac{\int e^{-f-g}}{\int e^{-f-g^\gamma}} \geq e^{-\gamma \frac{L^2}{2}}$, we get

$$\|\mu_\gamma - \pi\|_V \leq \pi(V) \left( e^{\gamma \frac{L^2}{2}} - 1 \right) + \pi(V) e^{\gamma \frac{L^2}{2}} \left( 1 - e^{-\gamma \frac{L^2}{2}} \right)$$

$$\leq 2\pi(V) \left( e^{\gamma \frac{L^2}{2}} - 1 \right).$$

For $u \in [0, 1]$, $e^u - 1 \leq 2u$, so for $\gamma \leq \frac{2}{L^2}$, we have

$$\|\mu_\gamma - \pi\|_V \leq 2\pi(V)\gamma L^2.$$

Using Lemma 18 of Appendix F.2, we obtain

$$\mathbf{W}_p(\mu_\gamma, \pi) \leq 2 \left( \pi(V_p)\gamma L^2 \right)^{\frac{1}{p}}.$$

which concludes for the inequality in $\mathbf{W}_p$ distance between $\mu_\gamma$ and $\pi$, with $E_p = 2 \left( \pi(V_p) \right)^{\frac{1}{p}}$.

Using Lemma 12, the fact that $\|\nabla g\|_\infty \leq L$ if $g$ is L-Lipschitz, the previous bound on $\mathbf{W}_p(\mu_\gamma, \pi)$ and the particular coupling $(I_d, \mathsf{Prox}_{\gamma g}) \# \mu_\gamma$ between $\mu_\gamma$ and $\nu_\gamma$, we get

$$\mathbf{W}_p(\nu_\gamma, \pi) \leq \mathbf{W}_p(\nu_\gamma, \mu_\gamma) + \mathbf{W}_p(\mu_\gamma, \pi) \leq \left( \int \|\gamma \nabla g(\mathsf{Prox}_{\gamma g}(x))\|^p d\mu_\gamma(x) \right)^{\frac{1}{p}} + \mathbf{W}_p(\mu_\gamma, \pi)$$

$$\leq \gamma L + 2 \left( \pi(V_p) \right)^{\frac{1}{p}} \left( L^2 \gamma \right)^{\frac{1}{p}}.$$

This demonstrates Proposition 1 with $E_p = 2(\pi(V_p))^{\frac{1}{p}}$. $\qquad \square$

## G.4 Proof of Theorem 4

The Inexact PSGLA is defined with $S_\gamma$ an inexact approximation of $\text{Prox}_{\gamma g}$ by

$$\hat{X}_{k+1} = S_\gamma\left(\hat{X}_k - \gamma\nabla f(\hat{X}_k) + \sqrt{2\gamma}\hat{Z}_{k+1}\right),$$

with $\hat{Z}_{k+1} \sim \mathcal{N}(0, I_d)$ i.i.d. We can write this algorithm as a two points algorithm

$$\hat{Y}_{k+1} = \hat{X}_k - \gamma\nabla f(\hat{X}_k) + \sqrt{2\gamma}\hat{Z}_{k+1}$$
$$\hat{X}_{k+1} = S_\gamma\left(\hat{Y}_{k+1}\right).$$

By introducing $G_\gamma = \gamma^{-1}(I_d - S_\gamma)$, we have that

$$\hat{Y}_{k+1} = S_\gamma\left(\hat{Y}_k\right) - \gamma\nabla f(S_\gamma\left(\hat{Y}_k\right)) + \sqrt{2\gamma}\hat{Z}_{k+1}$$
$$= \hat{Y}_k - \gamma G_\gamma(\hat{Y}_k) - \gamma\nabla f(\hat{Y}_k - \gamma G_\gamma(\hat{Y}_k)) + \sqrt{2\gamma}\hat{Z}_{k+1}$$
$$= \hat{Y}_k - \gamma\hat{b}^\gamma(\hat{Y}_k) + \sqrt{2\gamma}\hat{Z}_{k+1},$$

with the drift $\hat{b}^\gamma(y) = \nabla f(y - G_\gamma(y)) + G_\gamma(y)$.

We recall Theorem 4.

**Theorem 9.** *Under Assumption 2-3, if $\hat{b}^\gamma$ verifies Assumption 1, there exist $\hat{\gamma} > 0$ and $C_5, C_6, C_7 \in \mathbb{R}_+$ such that $\forall\gamma \in (0, \hat{\gamma}]$, $\hat{Y}_k$ of equation (14) has an invariant law $\hat{\mu}_\gamma$ and $\forall k \in \mathbb{N}$, we have*

$$\mathbf{W}_p(p_{\hat{X}_k}, \mu_\gamma) \le C_5 r^{k\gamma} + C_6\gamma^{\frac{1}{2p}} + \frac{C_7}{\gamma^{\frac{1}{p}}}\|S_\gamma - \text{Prox}_{\gamma g}\|_{\ell_2(\hat{\mu}_\gamma)}^{\frac{1}{2p}}.$$

Theorem 4 is a generalization of [33, Theorem 3.14] as it holds for non-convex potentials. Moreover, in our study, we do not assume *a priori* that the inexactness of the proximal operator is bounded.

*Proof.* We assume that $\hat{b}^\gamma$ verifies Assumption 1, so $\hat{Y}_k$ is geometrically ergodic with an invariant law $\hat{\mu}_\gamma$ and there exist $B_0 \ge 0$ and $r \in (0, 1)$, such that

$$\mathbf{W}_p(p_{\hat{Y}_k}, \hat{\mu}_\gamma) \le B_0 r^{k\gamma}, \tag{96}$$

with $p_{\hat{Y}_k}$ the distribution of $\hat{Y}_k$

We define the sequence $Y_{k+1} = Y_k - \gamma b^\gamma(Y_k) + \sqrt{2\gamma}Z_{k+1}$, with $b^\gamma(y) = \nabla f(y - \gamma\nabla g^\gamma(y)) + \nabla g^\gamma(y)$ defined in equation (11). Thanks to Theorem 3, under Assumption 2-3, $b^\gamma$ verifies Assumption 1 and $Y_k$ has $p_\infty$ as its invariant law. By Theorem 1, we get that there exist $\gamma_2 > 0$ and $B_1 \ge 0$ such that $\forall\gamma \in (0, \gamma_2]$

$$\mathbf{W}_p(\hat{\mu}_\gamma, p_\infty) \le B_1\left(\mathbb{E}_{Y\sim\hat{\mu}_\gamma}(\|\hat{b}^\gamma(Y) - b^\gamma(Y)\|^2)\right)^{\frac{1}{2p}}$$

$$\le 2^{\frac{1}{2p}}B_1\left(\mathbb{E}_{Y\sim\hat{\mu}_\gamma}(\|G_\gamma(Y) - \nabla g^\gamma(Y)\|^2 + \|\nabla f(Y - \gamma G_\gamma(Y)) - \nabla f(Y - \gamma\nabla g^\gamma(Y))\|^2)\right)^{\frac{1}{2p}}$$

$$\le 2^{\frac{1}{2p}}B_1\left(\mathbb{E}_{Y\sim\hat{\mu}_\gamma}(\|G_\gamma(Y) - \nabla g^\gamma(Y)\|^2 + L_f^2\gamma_2\|G_\gamma(Y) - \nabla g^\gamma(Y)\|^2)\right)^{\frac{1}{2p}}$$

$$\le 2^{\frac{1}{2p}}B_1(1 + L_f^2\gamma_2)^{\frac{1}{2p}}\left(\mathbb{E}_{Y\sim\hat{\mu}_\gamma}(\|\gamma^{-1}\left(\text{Prox}_{\gamma g}(Y) - S_\gamma(Y)\|^2)\right)^{\frac{1}{2p}}$$

$$\le 2^{\frac{1}{2p}}B_1(1 + L_f^2\gamma_2)^{\frac{1}{2p}}\gamma^{-\frac{1}{p}}\|\text{Prox}_{\gamma g} - S_\gamma\|_{\ell_2(\hat{\mu}_\gamma)}^{\frac{1}{2p}}. \tag{97}$$

By combining equation (87), equation (88), equation (96) and equation (97) and the triangle inequality, we get that

$$\mathbf{W}_p(p_{\hat{Y}_k}, \mu_\gamma) \le B_0 r^{k\gamma} + 2^{\frac{1}{2p}}B_1(1 + L_f^2\gamma_2)^{\frac{1}{2p}}\gamma^{-\frac{1}{p}}\|\text{Prox}_{\gamma g} - S_\gamma\|_{\ell_2(\hat{\mu}_\gamma)}^{\frac{1}{2p}} + C_3\gamma^{\frac{1}{p}} + C_4\gamma^{\frac{1}{2p}}$$

$$\le B_0 r^{k\gamma} + 2^{\frac{1}{2p}}B_1(1 + L_f^2\gamma_2)^{\frac{1}{2p}}\gamma^{-\frac{1}{p}}\|\text{Prox}_{\gamma g} - S_\gamma\|_{\ell_2(\hat{\mu}_\gamma)}^{\frac{1}{2p}} + C_3\gamma_2^{\frac{1}{2p}}\gamma^{\frac{1}{2p}} + C_4\gamma^{\frac{1}{2p}}$$

$$\le B_0 r^{k\gamma} + B_2\gamma^{-\frac{1}{p}}\|\text{Prox}_{\gamma g} - S_\gamma\|_{\ell_2(\hat{\mu}_\gamma)}^{\frac{1}{2p}} + B_3\gamma^{\frac{1}{2p}},$$

with $B_2 = 2^{\frac{1}{2p}} B_1 (1 + L_f^2 \gamma_2)^{\frac{1}{2p}}$ and $B_3 = C_3 \gamma_2^{\frac{1}{2p}} + C_4$.

Now, we prove that for $\gamma$ small enough, $G_\gamma$ is Lipschitz. By Assumption 1(i), $\hat{b}^\gamma$ is $L$-Lipschitz and by Assumption 2(i) $\nabla f$ is $L_f$-Lipschitz, so for $\gamma \in (0, \gamma_2]$ and $x, y \in \mathbb{R}^d$, we have

$$\|\hat{b}^\gamma(x) - \hat{b}^\gamma(y)\| \leq L\|x - y\|$$
$$\|G_\gamma(x) - G_\gamma(y) + \nabla f(x - \gamma G_\gamma(x)) - \nabla f(y - \gamma G_\gamma(y))\| \leq L\|x - y\|$$
$$\|G_\gamma(x) - G_\gamma(y)\| - \|\nabla f(x - \gamma G_\gamma(x)) - \nabla f(y - \gamma G_\gamma(y))\| \leq L\|x - y\|$$
$$\|G_\gamma(x) - G_\gamma(y)\| - L_f \left(\|x - y\| + \gamma\|G_\gamma(x) G_\gamma(y)\|\right) \leq L\|x - y\|$$
$$(1 - \gamma L_f)\|G_\gamma(x) - G_\gamma(y)\| \leq (L + L_f)\|x - y\|.$$

For $\gamma \leq \frac{1}{2L_f}$, we get

$$\|G_\gamma(x) - G_\gamma(y)\| \leq 2(L + L_f)\|x - y\|. \tag{98}$$

So $G_\gamma$ is $L_G = 2(L + L_f)$-Lipschitz. We denote $\gamma_3 = \min\left(\gamma_2, \frac{1}{2L_f}\right)$.

We have that $p_{\hat{X}_k} = S_\gamma \# p_{\hat{Y}_k}$ and $S_\gamma = I_d - \gamma G_\gamma$ is $(1 + \gamma_3 L_G)$ Lipschitz. Hence we obtain

$$\mathbf{W}_p(p_{\hat{X}_k}, \nu_\gamma) = \left(\min_{\beta \in \Lambda} \int_{\mathbb{R}^d \times \mathbb{R}^d} \|S_\gamma(x_1) - S_\gamma(x_2)\|^p d\beta(x_1, x_2)\right)^{\frac{1}{p}}$$
$$\leq (1 + \gamma_3 L_G) \left(\min_{\beta \in \Lambda} \int_{\mathbb{R}^d \times \mathbb{R}^d} \|x_1 - x_2\|^p d\beta(x_1, x_2)\right)^{\frac{1}{p}}$$
$$\leq (1 + \gamma_3 L_G) \mathbf{W}_p(p_{\hat{Y}_k}, \mu_\gamma),$$

with $\Lambda$ the set of distribution on $\mathbb{R}^d \times \mathbb{R}^d$ with marginals $p_{\hat{Y}_k}$ and $\mu_\gamma$. This proves the desired result with $\hat{\gamma} = \gamma_3$, $C_5 = (1 + \gamma_3 L_G) B_0$, $C_6 = (1 + \gamma_3 L_G) B_2$ and $C_6 = (1 + \gamma_3 L_G) B_3$. $\qquad\square$

