# OpenReview forum: "From stability of Langevin diffusion to convergence of proximal MCMC for non-log-concave sampling"
_NeurIPS.cc/2025/Conference — NeurIPS 2025 poster_

### Official Review · Reviewer_693y · 2025-06-26

**Clarity:** 4
**Significance:** 3
**Originality:** 3
**Rating:** 5
**Confidence:** 4

**Summary:**

This work studies the stability of the Unadjusted Langevin Algorithm to perturbations in the drift. These stability bounds allow them to prove a convergence result for an inexact proximal stochastic gradient Langevin algorithm in non-log-concave settings. Such a proximal algorithm can be used in cases where the potential is separable into a smooth part and a weakly convex part. The authors demonstrate an application of the proposed ideas in Bayesian denoising.

**Questions:**

- The paper is well-written and clear. However, I feel that it could be better structured to emphasize what the main idea and takeaway is. There are four theorems that say different things and each take parsing. It may be better to structure Theorems 3/4 as the main theorems, since they tie to the main practical application considered, and use Theorems 1/2 as supporting theory for these two. This would help the flow of the paper and allow for easier reading.
- It would be nice to see a setting where the proposed method is practically much better than existing methods. It is hard to see more than a marginal improvement in the existing experiments.
- Currently I am having a hard time of seeing how this is an analysis of a stochastic gradient Langevin algorithm. Where in the theory are you really using stochastic gradients? Under further assumptions on the stochastic gradients, can other techniques like variance reduction be used to result in faster convergence?
- Is there a possibility of applying this framework to study underdamped Proximal Stochastic Gradient Langevin? See
	Gurbuzbalaban, M., Hu, Y., & Zhu, L. (2024). Penalized Overdamped and Underdamped Langevin Monte Carlo Algorithms for Constrained Sampling. Journal of Machine Learning Research, 25(263), 1-67.

**Ethical Concerns:**

["NO or VERY MINOR ethics concerns only"]

**Final Justification:**

The authors have adequately answered my questions and I maintain that this is worthy to be accepted to neurips.

**Limitations:**

Yes, they comment on the lack of efficiency, but it is hard to see if the proposed routes towards increased efficiency will actually work.

**Quality:**

3

**Strengths And Weaknesses:**

Strengths:
- The authors develop useful theory on the stability of the inexact unadjusted Langevin algorithm.
- The theoretical developments allow them to prove a convergence bound for the inexact proximal stochastic gradient Langevin algorithm, which has applications in real scenarios where one uses plug-and-play estimators for proximal operators.
- The proof sketches, while short, are reasonably informative.

Weaknesses:
- The Gaussian mixture experiment is hard to read and not very informative.
- The denoising experiment exhibits a marginal improvement in denoising with increased cost over DiffPIR.
- In general, the framework does not seem to be computationall efficient compared to alternatives.

---

> ### Author Rebuttal · Authors · 2025-07-29
>
> Thank you for your positive feedback and careful reading of our paper. Please see below our detailed responses.
>
> - The Gaussian mixture experiment is hard to read and not very informative.
>
> In this experiment, we set a GMM prior in 2D and solve a denoising problem, this leads to a distribution to sample that is represented in black on Figure 1. PnP-ULA and PnP-PSGLA sampling outputs are plotted in green. We observe that both algorithms approximate the target distribution but PnP-PSGLA explores in few steps the distribution and the different modes. In order to show a quantitative metric, we plot the Wasserstein distance (on the right column) between the target distribution and the sampled distribution with respect to the number of iterations. We observe that both algorithms decrease this quantity and than PnP-PSGLA converges faster than PnP-ULA.
> We propose to reformulate the last sentence of the caption of Figure 1 with : "Note that PnP-PSGLA succeeds to sample the posterior distribution even if there are various modes and the convergence is faster than PnP-ULA".
>
> - The denoising experiment exhibits a marginal improvement in denoising with increased cost over DiffPIR.
>
> PnP-PSGLA and PnP-ULA sample the posterior distribution. DiffPIR, RED and PnP only estimate the maximum of the posterior distribution (MAP). Sampling provides a richer information with, for example, uncertainty information (see Figure 5 or Figure 6). Uncertainty quantification via sampling-based strategy is essential in decision-making processes that rely on restored images, notably in medical imaging applications. Therefore, it is natural than sampling methods such as PnP-ULA or PnP-PSGLA are slower than MAP methods as DiffPIR, RED and PnP. On Table 2, we show the capability of PnP-PSGLA to achieve state-of-the-art restoration quality, i.e. equivalent to DiffPIR, within sampling the posterior distribution and being significantly faster than PnP-ULA. This represents a clear step forward in empirical performance.
>
> - In general, the framework does not seem to be computational efficient compared to alternatives.
>
> As mentioned in the previous answer, sampling methods indeed have an increased computational cost compared to
> MAP methods such as DiffPIR, RED or PnP. However, PnP-PSGLA is significantly faster than PnP-ULA, the concurrent (state-of-the-art) method for sampling in real-world applications.
>
>
> - The paper is well-written and clear. However, I feel that it could be better structured to emphasize what the main idea and takeaway is. There are four theorems that say different things and each take parsing. It may be better to structure Theorems 3/4 as the main theorems, since they tie to the main practical application considered, and use Theorems 1/2 as supporting theory for these two. This would help the flow of the paper and allow for easier reading.
>
> We thank the reviewer for this suggestion. We choose to organize the paper in this way because we believe than Section 2 (Theorem 1/2) is  general and can be used for future works outside the context of proximal Langevin algorithms. Then we go deeper in our application on PSGLA in Section 3 (Theorem 3/4) based on the general results of Section 2 on Langevin algorithms. Finally, we apply PSGLA for posterior sampling. Our structure is going from the more general results to the more particular ones.
>
> - It would be nice to see a setting where the proposed method is practically much better than existing methods. It is hard to see more than a marginal improvement in the existing experiments.
>
> The proposed method PnP-PSGLA is significantly faster than the PnP-ULA method which is the state-of-the-art real-world sampling method. DiffPIR, RED and PnP do not sample the posterior distribution and in particular they do not compute uncertainty information.
>
> - Currently I am having a hard time of seeing how this is an analysis of a stochastic gradient Langevin algorithm. Where in the theory are you really using stochastic gradients ? Under further assumptions on the stochastic gradients, can other techniques like variance reduction be used to result in faster convergence ?
>
> In this paper, we choose to keep the original algorithm named "PSGLA" introduced in [80] which means Proximal Stochastic Gradient Langevin Algorithm. Unfortunately, we concede that this name can be confusing. In fact, the goal of PSGLA is to sample a target distribution and capture the variability of the distribution, e.g. its multi-modality. Therefore, PSGLA has not the same goal than stochastic gradient algorithms where the stochasticity is only used to optimize faster. Thus variance reduction methods are not appropriate in this context of sampling.
>
> - Is there a possibility of applying this framework to study underdamped Proximal Stochastic Gradient Langevin ? See Gurbuzbalaban, M., Hu, Y., \& Zhu, L. (2024). Penalized Overdamped and Underdamped Langevin Monte Carlo Algorithms for Constrained Sampling. Journal of Machine Learning Research, 25(263), 1-67.
>
> Thank for sharing with us this paper, we propose to include its reference in our related works in Appendix A. Our framework can be used to solve constraint sampling. If $g$ is the characteristic function of a convex set, PSGLA turns into a projected Langevin algorithm and our theory holds. Our theory about Langevin stability (Theorem 1) applies to  Penalized underdamped Langevin Monte Carlo (PULMC) as it is a Langevin type algorithm. On the other hand, the convergence result on PSGLA does not apply immediately to penalized underdamped Langevin Monte Carlo (PULMC) algorithm as PULMC is not a specific case of PSGLA. In this paper, the authors do not study a proximal Langevin algorithm as your question suggests. By any chance, would you have another reference in mind ?

---

### Official Review · Reviewer_op2G · 2025-06-30

**Clarity:** 3
**Significance:** 2
**Originality:** 3
**Rating:** 4
**Confidence:** 3

**Summary:**

The paper is interested in discretizations of the Langevin dynamics whose stationary measure has density is $\pi \propto e^{-V}$ where $V = f + g$ i.e., a composite potential. More specifically, the paper is focused on settings of $f$, $g$ which are both not necessarily convex and differentiable. The first contribution of the paper is to understand the effect of an inexact discretization of the Langevin dynamics, and give sharper results for the stability results of the discretizations in the Wasserstein and TV metrics. Somewhat orthogonal to this result, the authors then consider the proximal SGLA which mimics the forward-backward method in optimization but for sampling and provide guarantees for the method under what appear to be newer assumptions on $f$ and $g$ compared to prior work. Finally, they instantiate an approximate version of this method for a couple of empirical tasks where they demonstrate the efficacy of this method.

**Questions:**

1. Invariant law: $X_{\infty}$ instead of $X_{k}$?
2. What does data-fidelity mean? Is it another term for the negative log-likelihood of the data under some model?
3. Are there settings where the regularizer $g$ is chosen to the non-convex (which corresponds to a non-log-concave prior)?
4. Theorem 2 is a characterization of the bias limit of the unadjusted Langevin algorithm, for which there are analyses for when the distribution $\pi$ satisfies a log-Sobolev inequality which implies results of the form in Theorem 2 (for $p = 1, 2$ and TV). For drift $b = \nabla V$ that satisfies Assumption 1, can it be said that $pi$ satisfies a log-Sobolev inequality, and how would the bounds on $W_{p}(\pi_{\gamma}, \pi)$ compare in that case?
5. Why is there a dependence on the smoothness of $f$ in the properties assumed about the Moreau envelope of $g$? Is this in some sense necessary, or does it make the analysis easier?
6. The second part of theorem 3 is interesting, why is the push forward of the Moreau-corrected version of $\pi$ by the proximal operator an interesting distribution to have results for?
7. How does Proposition 1 compare to the composition of results from theorem 3 and Proposition 1 in https://arxiv.org/pdf/2405.15379? The setting in the latter proposition is constrained, so it might not be possible to directly use their result, in which case apologies.
8. How does this compare this to the work of https://arxiv.org/abs/2101.06369 generally?

**Ethical Concerns:**

["NO or VERY MINOR ethics concerns only"]

**Final Justification:**

I think this is an interesting paper, and would suggest the authors to make the recommended changes.

**Limitations:**

Yes.

**Quality:**

2

**Strengths And Weaknesses:**

The paper is decently written, although I feel like the paper could be written better -- there are redundancies in various parts of the text. For example:
- beginning of Section 3 (described previously in Lines 36-37),
- contributions (1) (described previously in Line 34).

As noted in the summary, the contents of Section 2 seem somewhat disconnected from the rest of the paper as framed, and it is not immediately clear how this is relevant to the rest of the work (although the results from Section 3 use a result from this section) and it would be useful to highlight the relevance apriori (or rearrange the sections appropriately). I still think that the results of that section are interesting, although it would be instructive to the reader to help clarify what were the (new, if any) elements in comparison to the previous analyses that led to the stronger guarantees in this work (I understand that there's a proof sketch, but I feel like this would be more valuable). In Section 3, the first two properties stated of the Moreau envelope are rather well known (see for instance the monograph by [Parikh and Boyd](https://dl.acm.org/doi/10.1561/2400000003)). The discussion about PSGLA is also mostly derivative of prior discussion in [81] and could be condensed. With regard to Assumption 3, it would be helpful to give some examples of potentials $g$ satisfy this together (since properties of $g^{\gamma}$ are generally harder to interpret.

Other miscellaneous remarks:

1. The notion of coupling in the definition of $W_{p}$ is a bit tedious, can be simply stated as collection of joint distributions whose marginals are $\mu$ and $\nu$.
2. Analogously, the pushforward can be simply stated as $\phi_{\\#}\mu(f) = \mu(\phi(f))$.
3. The notion of weakly convex is actually more commonly referred to as semi-convexity (see for instance: https://arxiv.org/tml/2502.03458v1).
4. Would recommend moving definitions about invariant law, geometric ergodicity into Sec 1.1.
5. Technically, only if $b$ is the gradient of a function is assumption 1(ii) a weaker version of $m$-strong convexity at infinity. This is just a growth condition that compliments the Lipschitz continuity assumption in 1(i) (Lipschitz continuity is not defined).
6. Line 64: A function twice-differentiable --> A twice-differentiable function
7. Line 103: following the same strategy than --> following the same strategy as

---

> ### Author Rebuttal · Authors · 2025-07-29
>
> Thank you for your positive feedback and careful reading of our paper. We thank in particular the reviewer for pointing us some typos. The typos that are not mentioned in our response will be corrected following the reviewer suggestions. Please see below our detailed responses.
>
> - I feel like the paper could be written better -- there are redundancies in various parts of the text.
>
> We thank the reviewer for spotting these redundancies. For the clarity of the paper, we choose to highlight our contributions in a separate paragraph and to explain our reasoning in the beginning of the sections. This choice can lead to some redundancies. We propose to reformulate line 34 into : "With a new proof strategy, we overcome this limitation in Theorem 1 in Section 2." We would also be happy to include other feedback regarding the writing of the paper.
>
> - As noted in the summary, the contents of Section 2 seem somewhat disconnected from the rest of the paper [...] this would be more valuable).
>
> We thank the reviewer for the feedback about the Section 2/3 articulation. Section 2 shows new general facts about Langevin stability in non-convex setting. Section 3 focuses on the convergence analysis of PSGLA, a specific proximal Langevin algorithm. The stability and discretization results proved in Section 2 are key for the PSGLA convergence analysis. Our structure is going from the more general results to the more particular ones. To make it clearer for the reader, we propose to add in line 71 the following sentence : "The sampling stability result (Theorem 1) and the discretization error bound (Theorem 2) demonstrated in the section will be the key ingredients for the PSGLA convergence analysis in Section 3."
>
> - In Section 3, the first two properties stated of the Moreau envelope are rather well known (see for instance the monograph by Parikh and Boyd).
>
> In fact the first three properties of Lemma 1 are well known for convex functions (as detailed in the monograph by Parikh and Boyd). In this paper, we generalize the properties for weakly convex functions. All the proofs and references for each property are given in Appendix D. The last property of Lemma 1 is totally new. We state these properties in the main paper because their are useful in Section 3.
>
> - The discussion about PSGLA is also mostly derivative of prior discussion in [81] and could be condensed.
>
> PSGLA have been introduced in [28] and studied in [81]. As advised by the reviewer, we checked [81] again. The interpretation of PSGLA as a different discretization of continuous Langevin dynamics than ULA is nevertheless not mentioned in [81] and the reformulation of PSGLA as a two-point algorithm to study the shadow sequence is not present in [81].
>
> - With regard to Assumption 3, it would be helpful to give some examples of potentials $g$ satisfy this together (since properties of $g^\gamma$ are generally harder to interpret).
>
> Lemma 16 in Appendix E gives an example of $g$ that satisfies Assumption 3. It shows that if $g$ is strongly convex at infinity and globally smooth, then Assumption 3 is verified. Prior with GMM potential are strongly convex at infinity and globally smooth.
> Please note that Assumption 3(ii) could be removed by slightly modifying PSGLA as detailed in Appendix B on the discussion about Assumption 3(ii).
> To make it more explicit in our comment, we propose to reformulate line 178-179 with : "Lemma 16 in Appendix E shows that if g is strongly convex at infinity and globally smooth, such as Gaussian mixture models, then Assumption 3(ii) is verified.".
>
> # Miscellaneous remarks:
> 1- We thank the reviewer for his suggestion and we propose to modify line 62-63 into "with $\Pi$ the set of probability law $\beta$ on $\mathbb{R}^d \times \mathbb{R}^d$ whose marginals are $\mu$ and $\nu$.".
>
> 2- In our paper, we need (for Theorem 3 and Proposition 1 in particular) to introduce the pushforward measure and not only the pushforward measure applied to a function $f$.
>
> 3- In the literature, both terminology are used : semiconvex in [Crandall, Michael G and Ishii, Hitoshi and Lions, Pierre-Louis, User’s guide to viscosity solutions of second order partial differential equations]; and weakly convex in [20, 39, 44, 47, 82]. We choose to use weakly-convex as it seems to be  common in the machine learning community. We propose to add in line 64, the sentence. "This notion is also called semi-convexity [Crandall, Michael G and Ishii, Hitoshi and Lions, Pierre-Louis, User’s guide to viscosity solutions of second order partial differential equations][The Performance Of The Unadjusted Langevin Algorithm Without Smoothness Assumptions, Tim Johnston, Iosif Lytras, Nikolaos Makras, and Sotirios Sabanis]."
>
> 4- We thank the reviewer for this recommendation and we propose to follow it.
>
> 5- We thank the reviewer for noticing this imprecision. We propose to reformulate line 78 as "$b$ is $L$-Lipschitz, i.e. $\forall x, y \in \mathbb{R}^d$, $\|b(x) - b(y)\| \le L \|x-y\|$". We propose to precise our comment in line 81-82 by "If $b = \nabla V$, Assumption 1(ii) is a relaxation of the strong convexity assumption on $V$, as it only holds for couple of points sufficiently far away".
>
> # Questions
> 1- Please correct us if we do not understand your suggestion. In the paper $p_{X_k}$ is the law of the iterate $X_k$ and $\mu$, $\nu$ and $\pi$ are used for invariant laws. Thus there is no conflict of notations.
>
> 2- In the image restoration community, the log-likelihood of the data is commonly called data-fidelity, see for instance [Plug-and-Play Methods for Integrating Physical and Learned Models in Computational Imaging, Kamilov, Bouman, Buzzard, Wohlberg 2022]. To avoid confusion, we propose to reformulate "data-fidelity" in line 36, line 128 and line 633 by "negative log-likelihood".
>
> 3- Yes, for GMM prior or modern image priors induced by deep neural networks, the regularizer $g$ is non-convex, this is the case in all our experiments.
>
> 4- From a first analysis, the connection between our Assumption 1 (strong convexity outside of a ball with certain radius) and a logarithmic Sobolev inequality does not seem to be immediate. Indeed, to the best of our knowledge the logarithmic Sobolev inequality is implied by a very strong version of the Foster-Lyapunov condition, see for instance [A link between the log-Sobolev inequality and Lyapunov condition, Lyapunov conditions for logarithmic Sobolev and Super Poincaré inequality]. However, while the target distribution might not satisfy a logarithmic Sobolev inequality it does satisfy a Poincar\'e inequality, since Poincar\'e inequalities are satisfied under a weaker Lyapunov condition [A simple proof of the Poincaré inequality for a large class of probability measures]. We then believe that using results from [Rapid Convergence of the Unadjusted Langevin Algorithm:
> Isoperimetry Suffices], we could derive the convergence of the ULA with respect to the R\'enyi divergence. It is unclear if those results would be comparable to the ones we obtained in Wasserstein distance. Overall, we believe that using functional analysis tools like the logarithmic Sobolev or the Poincar\'e inequality would constitute a complementary and interesting work avenue
>
> 5- The constant of strong convexity of $g^\gamma$ that appears in Assumption 3(ii) has to be larger than $8 L_f + 4 L_g$ in order to ensure that the drift defined in equation (11) that depends on $\nabla f$ and $\nabla g^\gamma$ is increasing enough at infinity, i.e. verifies Assumption 1(ii). Roughly, the strong convexity of $g^\gamma$ at infinity needs to compensate the weak convexity of $f$. This hard-to-verify Assumption 3(ii) could be remove by adding a projection mechanism in the PSGLA algorithm, see Appendix B for details.
>
> 6- The distribution $\nu_\gamma$ proportional to $Prox_{\gamma g}$ \# $e^{-f - g^\gamma}$ is not the target distribution, which is $\pi \propto e^{-f-g}$. We were not able to prove the convergence of PSGLA to $\pi$ without additional assumption, but only to the shifted distribution $\nu_{\gamma}$, which arises naturally in the proof. However, Proposition 1 shows that $\nu_{\gamma}$ is close to $\pi$ when $\gamma \to 0$ and quantifies the difference between these two distribution if $g$ is Lipschitz.
> From a practical point of view in image restoration, finishing by the application of the proximal operator "projects" the iterates on the clean-image manifold and improves the samples' quality.
>
> 7- We thank the reviewer for pointing us these results. They are derived in a more restrictive setting but they require less assumptions. On the one hand this paper is lore restrictive because $g$ is supposed to be the characteristic function of some convex set. Proposition 1 of this paper then proposes a result that is dependent of the dimension of the problem ($p$ in their notation). On the other hand, they do not assume the Lipschitzness of $g$ but some geometrical properties on the convex set and the projection to this set. Therefore, these results could not be applied directly in our setting.
>
> 8- First, the paper that you mention focuses on ULA and does not analyse proximal Langevin algorithm such as PSGLA. Secondly, this paper focuses on the ULA convergence whereas we study in our paper the ULA stability to drift shifts. Finally, we derive our bound in total-variation and Wasserstein distances but not in Kullback-Leibler (KL) divergence, which is included in this paper. As mentioned in Appendix F.4 after Theorem 6, line 1321-1324, our proof strategy does not generalize easily to KL because it relies massively on the triangle inequality that is not verified by KL. Based on the mentioned paper, we could try to derive a bound in KL. Thanks for sharing with us this paper, we propose to include this reference in line 1321.

---

> > ### Comment · Reviewer_op2G · 2025-08-02
> > **Some minor clarifications**
> >
> > Thank you for your rebuttal which I've read thoroughly.
> >
> > 1. [In reference to your rebuttal to Q1]: When you mention "invariant law of the MCMC $X_{k}$, I think you intended to state $X_{\infty}$. This doesn't coincide with the invariant law of the SDE in Eq. (1) unless $h \to 0$.
> >
> > 2. [In reference to your rebuttal to Q5]: I think a reference to Appendix B to avoid Assumption 3(ii) in the main text would be useful.

---

> > > ### Author Response · Authors · 2025-08-05
> > >
> > > We thanks the reviewer for these two additional suggestions.
> > >
> > > We understand the ambiguity of the formulation of line 31, to avoid confusion we suggest to reformulate "the shift between the invariant law of the MCMC and the target distribution $\pi$". There is no need to name this invariant law in the introduction of the paper (named $\pi_{\gamma}$ or $\hat{\pi_{\gamma}}$ depending of the MCMC in Section 2).
> > >
> > > There is a reference to Appendix B in line 177-178 in the comment on Assumption 3(ii). We propose to create a new paragraph in line 177 to separate the comment on Assumption(i) and the comment on Assumption(ii) and stress the reference to Appendix B.

---

### Official Review · Reviewer_zNf2 · 2025-07-02

**Clarity:** 3
**Significance:** 3
**Originality:** 3
**Rating:** 5
**Confidence:** 3

**Summary:**

This is a theoretical paper addressing the problem of sampling distributions with non-convex potentials. The authors focus on analyzing the Unadjusted Langevin Algorithm (ULA) and its proximal variant, the Proximal Stochastic Gradient Langevin Algorithm (PSGLA). They contribute to proving the stability guarantee and discretization error of inexact ULA, as well as the convergence of PSGLA. Finally, an application to posterior sampling is included to show the superiority of PSGLA over ULA.

**Questions:**

I don't have further questions but I will continue to refer to the opinions of other reviewers.

**Ethical Concerns:**

["NO or VERY MINOR ethics concerns only"]

**Final Justification:**

I thank the authors for the response. After reading the other valuable reviews, I believe that the quality and potential impact of this paper are worthy of further recognition. I am willing to increase my rating.

**Limitations:**

Yes.

**Paper Formatting Concerns:**

I don't find any major formatting issues in this paper.

**Quality:**

3

**Strengths And Weaknesses:**

Strengths:
- The theoretical part of this paper is well-organized with clear statements. Several contributions are made for theoretical guarantee of non-log-concave sampling.
- Detailed proofs as well as comprehensive discussions about the made assumptions are included.

Weaknesses:
- The link between the theoretical and experiment parts is not very tight. While the theoretical part mainly focuses on proving the stability results of ULA and the convergence of PSGLA, the experiment part focuses on showing the superiority of PSGLA over ULA.

I am familiar with diffusion models but not particularly familiar with this specific field. I will continue to refer to the opinions of other reviewers for a more comprehensive evaluation.

---

> ### Author Rebuttal · Authors · 2025-07-29
>
> Thank you for your positive feedback and careful reading of our paper. If the reviewer has any more concerns regarding our paper, we are ready to answer them.
>
> - The link between the theoretical and experiment parts is not very tight. While the theoretical part mainly focuses on proving the stability results of ULA and the convergence of PSGLA, the experiment part focuses on showing the superiority of PSGLA over ULA.
>
> The theoretical part focuses on proving a refined stability result on Langevin algorithms and use it to derive the first proof of convergence of the PSGLA in non-convex setting. In the experimental part, we showcase the PSGLA convergence and performance in different non-convex settings including GMM sampling and image restoration (see Figure 1 right row for convergence for GMM). Those theoretical results motivate our empirical study. In practice, we found that PSGLA converges faster than ULA to acceptable solutions.

---

### Official Review · Reviewer_GF73 · 2025-07-03

**Clarity:** 4
**Significance:** 3
**Originality:** 3
**Rating:** 5
**Confidence:** 3

**Summary:**

The paper studies the stability of unadjusted Langevin algorithm with drift approximations, under the assumption that the potential is strongly convex at infinity. This assumption allows for nonconvex potentials. The resulting bound of the distance between (two biased) invariant distributions involve only the drift approximation error and no additional discretization error as seen in previous bounds. The distance between the biased invariant distribution and the target distribution is also provided. These bounds are in both $W_p$ and TV. The paper then applies these bounds to a specific algorithm with an approximate drift, namely, the proximal stochastic gradient Langevin algorithm (PSGLA). The convergence and bias bounds of PSGLA are provided. Numerical experiments compare several methods in posterior sampling, including PSGLA and its variants, and show competitive performance.

**Questions:**

Questions:
- The key contraction results are referenced from previous works. I am curious whether they are based on reflection coupling since it is a common technique in nonconvex settings. Also, how good are those referenced bounds?
- Regarding the tightness of the bounds in terms of nonconvexity, I am wondering whether the dependence on $R$ is hidden in the constants in the proofs. For example, in the proof of Theorem 5, page 41 eq (52), Corollary 2 in reference [22] is referred, but as I see in that paper, it should be $(1 + \|x-y\|/R)^{\frac12}$. Maybe exposing $R$ dependence could be helpful to see the behavior of the bounds in nonconvex cases.
- In the proof of Theorem 5, how important is it to choose $m \gamma \sim 1$? Do other choices of $m$ give similar bounds?
- page 49 eq(80)-(81): for $\mathbb{E}\|\nabla V(y_{u,j})\|^2$, I understand that $y_{u,j}$ is not at equilibrium. It is known that $\mathbb{E}_\pi\|\nabla V\|^2 \leq Ld$ if $\nabla^2 V \preceq L I_d$, and I am wondering whether utilizing this fact could improve the bound a bit.


Typos/minor questions:
- page 3 line 81: "In particular, the $L$-Lipschitzness involves that the drift is $L$-weakly convex". Also page 16 line 623-624, "Assumption (i) is equivalent to $-LI_d \preceq \nabla^2 b \preceq L I_d$. I don't understand this claim. Since the Lipschitz constant is essentially a first-order information, how could it imply the second order derivatives of $b$?
- page 8 Figure 1 caption: prior distributions in green -> they are in blue. The samples are green.
- page 16 line 620: "Assumption 2(i)" -> Assumption 1(i).
- page 38 Lemma 17, for TV, $V = 1$ and the current statement will be off by a factor of 2 (since TV is used as in line 1151 on page 39).
- page 39 the equation after line 1156: there should be an expectation sign for the integral part.
- page 41 line 1218, "$R \pi = \pi$" -> the LHS should be left multiplied by the $\pi$.
- page 45 line 1295: the inequalities after line 1295 and line 1297 have some inconsistent notations: eg. C_2 vs A_2 and W_4.
- The statement in Proposition 1 and its restatement as Proposition 4 on page 52 are not consistent: Proposition 4 has $L^2$ in eqs (92) and (93) but they are $L$ in Proposition 1.
- page 47 line 1339, "$\rho_p$ is independent of $d$" -- does $\rho_p$ depend on $\gamma$?
- page 57 the equation after line 1524 misses $1/p$ powers on the right.

**Ethical Concerns:**

["NO or VERY MINOR ethics concerns only"]

**Final Justification:**

The authors have provided detailed and clear response to all my questions, and I think this is a solid paper to publish in NeurIPS.

**Limitations:**

There are some discussions of limitations in Section 7, but limitations in terms of the theoretical analysis are not discussed.

**Quality:**

4

**Strengths And Weaknesses:**

Strengths:
1. The paper is extremely well-written. The related works are clearly covered. The organization of the paper is thoughtful: Section 2 provides stability results of ULA which could be helpful for future research; and Section 3 provides intuitions and properties of proximal samplers, and then applies theories in Section 2 to the proximal sampler. This organization separates two theoretically heavy parts: stability results and proximal sampler properties, into two sections, which help readers to follow the paper. Within each theoretical part, the authors clearly state the main theorems and provide discussions of these results, without confusing readers with technical details. The appendix is also well-written. It has great background information. For example, it summarizes properties of the Moreau envelope, which is very friendly for people first knowing this important concept for proximal samplers. The proofs of the theorems have clear frameworks and I find them easy to follow. The authors show care for many writing details. Overall, the authors did a great job of presenting the results which could otherwise be very dense.
2. The question of convergence and bias of proximal samplers in non-log-concave sampling is interesting.
3. The stability results of unadjusted Langevin algorithm involve only the difference of the drifts, and bounds in both $W_p$ and TV metrics are provided. I think the results are interesting and can be useful for future research.
4. Besides the theoretical contributions, the paper also has abundant numerical experiments on posterior sampling, comparing variants of PSGLA with other image restoration algorithms. The authors draw clear connections between the theoretical setup and the application scenario.

Weaknesses:
1. I am not sure how good/tight these bounds are, especially since most constants are not explicit (although I understand that the dependence could be complicated). I see discussions in Appendix F.5 on constants in Theorem 1. Currently, there are no discussions on the tightness of these bounds, for example, on the dimension $d$, nonconvexity (the radius $R$), and constants $m$ and $L$. In simple cases such as the Gaussian mixture, are the bounds good enough (especially when the nonconvexity is severe)?
2. There is no theoretical explanation about the phenomenon that PnP-PSGLA with DnCNN outperforms PnP-ULA and the latter is slower in the numerical experiments. In the Gaussian mixture example, page 8 line 265, the authors mention the intuition that "PnP-PSGLA rigidly project the iterates on the 'clean data' manifold". I am curious whether the advantage of PnP-PSGLA over PnP-ULA can be understood within the framework of Section 2. (Although Section 2 is about distance between invariant distributions, the proof involves convergence rates along iterations.)

---

> ### Author Rebuttal · Authors · 2025-07-29
>
> Thank you for your positive feedback and careful reading of our paper. We thank the reviewer for sharing some typos, especially in the Appendix. The typos that are not mentioned in our response will be corrected following the reviewer suggestions. Please see below our detailed responses.
>
> - I am not sure how good/tight these bounds are, especially since most constants are not explicit (although I understand that the dependence could be complicated). I see discussions in Appendix F.5 on constants in Theorem 1. Currently, there are no discussions on the tightness of these bounds, for example, on the dimension $d$, nonconvexity (the radius $R$), and constants $m$ and $L$. In simple cases such as the Gaussian mixture, are the bounds good enough (especially when the nonconvexity is severe)?
>
> We thank the reviewer for raising questions regarding the tightness of the derived bounds. We acknowledge that this problem is mostly untractable. Indeed, in order to evaluate the tightness of the bound, we need for two drift $b^1$ and $b^2$ to be able to compute the associated invariant laws $\pi_{\gamma}^1$ and $\pi_{\gamma}^2$. Even if the drifts $b^1, b^2$ derive from Gaussian Mixture potentials, the invariant laws $\pi_{\gamma}^1, \pi_{\gamma}^2$ do not seem to be explicit. We find that if the drift $b^1, b^2$ derive from isotropic Gaussian potentials, the invariant distributions $\pi_{\gamma}^1, \pi_{\gamma}^2$ can be computed explicitly and the constant of Theorem 1 does not depend of the dimension. However this simple case does not capture the non-convex phenomena that we aim to address in this work. In order to highlight this limitation of our work, we propose to add at line 309 : "From a theoretical point of view, we identify two main limitations of this work that might motivate future research. The first one is to obtain theoretical insights, such as accelerated convergence rate, of the superiority of PSGLA over ULA. The second one is to analyse the tightness of the constants involved in Theorem 1-2 that are partially implicit in the current work."
>
> - There is no theoretical explanation about the phenomenon that PnP-PSGLA with DnCNN outperforms PnP-ULA and the latter is slower in the numerical experiments. In the Gaussian mixture example, page 8 line 265, the authors mention the intuition that "PnP-PSGLA rigidly project the iterates on the 'clean data' manifold". I am curious whether the advantage of PnP-PSGLA over PnP-ULA can be understood within the framework of Section 2. (Although Section 2 is about distance between invariant distributions, the proof involves convergence rates along iterations.)
>
> The superiority of PnP-PSGLA over PnP-ULA is only supported by empirical evidence in this paper. We highlight it to motivate the interest for more theoretical insights on PSGLA. In fact, the mixing time of PnP-ULA is its main drawback and PnP-PSGLA seems to be a good practical candidate to sample with a more reasonable mixing time. In this paper, we give the first proof of convergence for PSGLA in a non-convex setting.
> Both PSGLA and ULA geometrically converge in our setting, studying the convergence rates of each method might be a way to compare theoretically these two algorithms. A fine analysis of the rates provided by [22] should be a first step for this analysis. We leave that work for future work.
> We propose to add in the conclusion the following sentence at line 309: "From a theoretical point of view, we identify two main limitations of this work that might motivate future research. The first one is to obtain theoretical insights, such as accelerated convergence rate, of the superiority of PSGLA over ULA. The second one is to analyse the tightness of the constants involve in Theorem 1-2 that are partially implicit in the current work."
>
> - The key contraction results are referenced from previous works. I am curious whether they are based on reflection coupling since it is a common technique in nonconvex settings. Also, how good are those referenced bounds?
>
> Indeed the main contraction rates we exploit are proved in  [22], where the authors based their analysis on a projected version of the discrete reflection coupling. To our knowledge, these bounds are state-of-the-art in the context of Langevin sampling for non-convex potential. Moreover, the rate proved in [22] (see equation (10)-(11) in [22]) is closed to the optimal convergence rate (from a small factor $(1-e^{2mR^2})^{-1}$) in the small step-size limit, i.e, those rates match the continuous-time limite rates obtained in [A Eberle, A Guillin, R Zimmer, Couplings and quantitative contraction rates for Langevin dynamics] which are optimal.
>
> - Regarding the tightness of the bounds in terms of nonconvexity, I am wondering whether the dependence on $R$ is hidden in the constants in the proofs. For example, in the proof of Theorem 5, page 41 eq (52), Corollary 2 in reference [22] is referred, but as I see in that paper, it should be $(1+|x-y|/R)^{\frac{1}{2}}$. Maybe exposing $R$ dependence could be helpful to see the behavior of the bounds in nonconvex cases.
>
> As the reviewer noticed, $C_p$ indeed explodes when $R \to 0$. This dependency is hidden in the constants $E_p$ and $A_2(x)$ in the paper. We obtain $C_p \sim \left(\frac{1}{R}\right)^{\frac{1}{2p}}$ when $R \to 0$. However, in the case of $R = 0$, the potential is strongly convex and other tools could be used. The challenging and albeit more interesting case is when $R \to + \infty$. We obtain that the dependency is exponential in $R$ with $C_p = \mathcal{O}(e^{\frac{LR^2}{4}})$ when $R \to +\infty$.
> We propose to make these dependencies explicit and detail them in Section F.5.
>
> - In the proof of Theorem 5, how important is it to choose $m \gamma \approx 1$ ? Do other choices of $m$ give similar bounds?
>
> The choice of $m \gamma \approx 1$ is arbitrary in the proof and can be changed. The important point for the proof to hold is to choose $m \gamma$ close to a fixed quantity. If we chose a different quantity for $m \gamma$, the proof can be derived in the same way and, at the end, the constant will be multiplied by a factor $\frac{\sqrt{m \gamma}}{1 - \rho_p^{m \gamma}}$. In theory, the factor $m \gamma$ could be optimized.  However, to our knowledge there is no closed-form formula to find the minimum of $m \gamma \mapsto \frac{\sqrt{m \gamma}}{1 - \rho_p^{m \gamma}}$. Therefore, we choose to keep $m \gamma \approx 1$ for simplicity.
>
> - page 49 eq(80)-(81): for $\mathbb{E}|\nabla V(y_{u,j})|^2$, I understand that $y_{u,j}$ is not at equilibrium. It is known that $\mathbb{E}|\nabla V|^2 \le L d$ if $\nabla^2 V \preceq L I_d$, and I am wondering whether utilizing this fact could improve the bound a bit.
>
> We thank the reviewer for this suggestion to improve the proof. Please correct us if we do not understand your suggestion. If $V(x) = \|x\|^2$ for $x \in \mathbb{R}^d$, then $\nabla V(x) = 2x$ and $\nabla^2 V = 2 I_d$, so, for $L = 2$, $\nabla^2 V \preceq L I_d$ but $\mathbb{E}|\nabla V(y_{u,j})|^2 = 4 \mathbb{E}|y_{u,j}|^2$ could be larger than $L d$ if $y_{u,j}$ has a large variance. This seems to be a counter-example to this suggestion.
>
> - page 3 line 81: "In particular, the $L$-Lipschitzness involves that the drift is $L$-weakly convex". Also page 16 line 623-624, "Assumption 1(i) is equivalent to $-L I_d \preceq \nabla^2 b \preceq L I_d$. I don't understand this claim. Since the Lipschitz constant is essentially a first-order information, how could it imply the second order derivatives of $b$?
>
> We thank the reviewer to point us this typo, it should be $\nabla b$ instead of $\nabla^2 b$. In fact the Lipschitness is linked to a first order information. We propose to reformulate page 3 line 81 into : "In particular if $b = \nabla V$, the $L$-Lipschitzness of $b$ implies that the potential $V$ is $L$-weakly convex". And to reformulate page 16 line 623-624 into "Moreover, if $b = \nabla V$, the potential $V$ is $L$-weakly convex. Assumption 1(i) is equivalent to $-L I_d \preceq \nabla b \preceq L I_d$.".
>
> - There are some discussions of limitations in Section 7, but limitations in terms of the theoretical analysis are not discussed.
>
> We propose to add in line 309 the following theoretical limitations : "From a theoretical point of view, we identify two main limitations of this work that might motivate future research. The first one is to obtain theoretical insights, such as accelerated convergence rate, of the superiority of PSGLA over ULA. The second one is to analysis the tightness of the constants involve in Theorem 1-2 that are partially implicit in the current work.".
>
> - page 47 line 1339, "$\rho_p$ is independent of $d$" -- does $\rho_p$ depend on $\gamma$ ?
>
> The constant $\rho_p$ is independent of $\gamma$ for $\gamma \in (0, \gamma_0]$, This is detailed in Appendix F.5 line 1338. We highlight here that the main dependence of $\rho_p$ is with respect to the radius of non-convexity $R$.
>
> - page 38 Lemma 17, for TV, $V = 1$ and the current statement will be off by a factor of 2 (since TV is used as in line 1151 on page 39).
>
> We thank the reviewer for stressing this imprecision. The correct formula is $|\mu - \nu|= 2 \inf_{\beta \in \Pi} E_{(X, Y) \sim \beta}{1_{X \neq Y}}$, with$|\cdot|$ the total-variation norm, (see for instance Theorem 5.2 in [Torgny Lindvall. Lectures on the coupling method, 2002]).
> We will correct the error in line 1151, lemma 19 and the constant in the proof (it only changes the constant $D_p$ to be $D_p = 4 \sqrt{2} E_p \sqrt{M_{2p}}$ in line 1236).

---

> > ### Comment · Reviewer_GF73 · 2025-08-04
> >
> > I want to thank the authors for their detailed and clear response. Some follow-up thoughts:
> >
> > 1. This point was not raised in my original review. I am wondering the order in the step size $\gamma$ in the bias bound in Theorem 2. For unadjusted Langevin, the $W_2$ bias is on the order of $\mathcal{O}(\sqrt{\gamma})$ (for example, in Theorem 1 of [Arnak S. Dalalyan, Further and stronger analogy between sampling and optimization: Langevin Monte Carlo and gradient descent]). I am wondering why the bound in Theorem 2 says $\gamma^{1/(2p)} = \gamma^{1/4}$ for the $W_2$ bias.
> >
> > 2. The response about the tightness of bounds is very helpful. For $C_p = \mathcal{O}(\exp(L R^2/4))$ when $R\to \infty$ (mentioned in the rebuttal), I am wondering whether this kind of exponential dependence on $R$ is essential.
> >
> > 3. Regarding my comments on page 49 eq(80)-(81): $\mathbb{E}_{\pi}|\nabla V|_2^2 \leq Ld$ and in the Gaussian case $V = 1/2 |x|_2^2$ it holds as an equality, because the expectation is taken wrt $\pi = \exp(-V)$. And I think that if another distribution is close to $\pi$ then a similar bound might hold.

---

> > > ### Author Response · Authors · 2025-08-05
> > >
> > > We thank the reviewer for the interest and these new questions/suggestions.
> > >
> > > 1/ We do not proved that the bound in $\gamma^{\frac{1}{p}}$ of Theorem 2 is optimal. Theorem 1 of [Arnak S. Dalalyan, Further and stronger analogy between sampling and optimization: Langevin Monte Carlo and gradient descent] proves a bound in $\gamma^{\frac{1}{2}}$ in the strongly convex case which is more restrictive than our. Thus, this result does not show that our bound is not optimal. Do you have other references in mind about the discretization bound under the weakly dissipative assumption ?
> > >
> > > 2/ It is natural that $C_p \to +\infty$ when $R \to +\infty$ because the potential is more and more non-convex. However, we do not have intuitions to share about the speed of growth.
> > >
> > > 3/ In page 49, the expectation is not taken on $\pi$ but on the random variable $y_{u,j}$ on which we do not have control of the variance.
> > >
> > > We will be happy to discuss or answer to new questions !

---

> > > > ### Comment · Reviewer_GF73 · 2025-08-05
> > > >
> > > > I want to thank the authors for their additional response. I do not have further questions.

---

### Decision · Program_Chairs · 2025-09-17

**Decision:**

Accept (poster)

**Comment:**

This work established the convergence of proximal, stochastic gradient MCMC algorithms under inexact gradients in the non-log concave setting. This is primarily achieved by studying the stability of unadjusted langevin algorithm to perturbations in the drift function (i.e, inexactness). These algorithms are widely used in practice and this work provides important theoretical justifications for their use in the relevant inexact, stochastic, non-log-concave settings. The reviewers agree that the contributions are important and the paper is technically sound. I recommend acceptance.